# Mode-Shell correspondence, a unifying phase space theory in topological physics – part II: Higher-dimensional spectral invariants

Lucien Jezequel[1,2] and Pierre Delplace[1]

[1]CNRS, ENS de Lyon, LPENSL, UMR5672, 69342, Lyon cedex 07, France
[2]Department of Physics, KTH Royal Institute of Technology, Stockholm 106 91, Sweden

January 24, 2025

### Abstract

The mode-shell correspondence relates the number $\mathcal{I}_\mathrm{M}$ of gapless modes in phase space to a topological *shell invariant* $\mathcal{I}_\mathrm{S}$ defined on a close surface – the shell – surrounding those modes, namely $\mathcal{I}_\mathrm{M} = \mathcal{I}_\mathrm{S}$. In part I [1], we introduced the mode-shell correspondence for zero-modes of chiral symmetric Hamiltonians (class AIII). In this part II, we extend the correspondence to arbitrary dimension and to both symmetry classes A and AIII. This allows us to include, in particular, $1D$-unidirectional edge modes of Chern insulators, $2D$ massless Dirac and $3D$-Weyl cones, within the same formalism. We provide an expression of $\mathcal{I}_\mathrm{M}$ that only depends on the dimension of the dispersion relation of the gapless mode, and does not require a translation invariance. Then, we show that the topology of the shell (a circle, a sphere, a torus), that must account for the spreading of the gapless mode in phase space, yields specific expressions of the shell index. Semi-classical expressions of those shell indices are also derived and reduce to either Chern or winding numbers depending on the parity of the mode's dimension. In that way, the mode-shell correspondence provides a unified and systematic topological description of both bulk and boundary gapless modes in any dimension, and in particular includes the bulk-boundary correspondence. We illustrate the generality of the theory by analyzing several models of semimetals and insulators, both on lattices and in the continuum, and also discuss weak and higher-order topological phases within this framework. Although this paper is a continuation of Part I, the content remains sufficiently independent to be mostly read separately.

# Contents

# 1   Motivation and brief summary

In the Part I of this work [1], we derived an explicit relation between a spectral invariant that counts the chiral number of zero-modes, called *mode index* $\mathcal{I}_M$, and a *shell index* $\mathcal{I}_S$ that evaluates this number over a shell that surrounds the zero-modes in phase space. We called this result the *mode-shell correspondence.* We then showed that in a semi-classical limit, where the radius of the shell $\Gamma \to \infty$, the shell index coincides with a (higher) winding number defined on the shell. Those results dealt with chiral symmetric Hamiltonians $\hat{H}(x, \partial_x)$ whose Wigner-Weyl symbols $H(x, k_x)$ are gapped in phase space where the shell is defined, that is, away from the Wigner representation of the zero-mode. Here, the meaning of "zero" in zero-mode is two-fold: it refers both to the zero-energy of the mode, owing to chiral symmetry, and to the number of directions in which this mode disperses in $k$-space. We shall refer to the latter as the "dimension of the mode" $D_M$ below. In other words, we focused in Part I on $D_M = 0$ zero-energy modes, meaning that those modes have no dispersion relation, a basic example of which being the boundary-mode of an SSH chain [2, 3]. For this reason, we will say, for short, that $\mathcal{I}_M$ is a zero-dimensional mode index in that case.

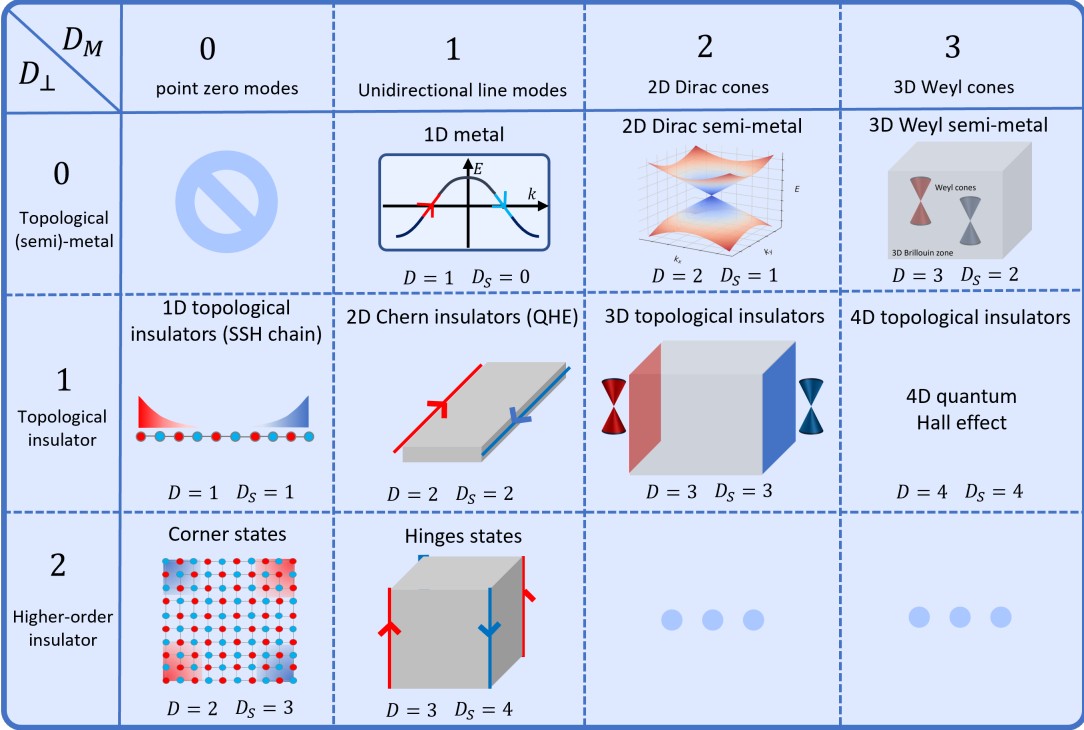

Figure 1: Classification of gapless states according to $D_M$ and $D_\perp$ within the mode-shell correspondence picture, with illustrative examples discussed in Part I and Part II of this study. The dimension $D$ of the system and the dimension $D_S$ of the shell are also specified. The mode-shell formalism provides explicit expressions for the mode and shell invariants for each of these cases.

In this Part II, we introduce higher dimensional mode indices $\mathcal{I}_M$ for $D_M > 0$ and discuss their mode-shell correspondence. This generalization is motivated by cornerstone examples in the literature. In particular, edge states of Chern insulators display a dispersion relation with respect to one parameter, the momentum (or wavenumber) parallel to the boundary of the system [4–9]. We thus would like to extend the definition of $\mathcal{I}_M$ such that it incorporates those unidirectional edge modes for $D_M = 1$. Similarly, the dispersion relation of Weyl fermions in dimension $D = 2$ and $D = 3$ consist of a Dirac/Weyl cone that lies respectively along 2 and 3 directions in momentum space [10–20]. Accordingly, we would like to assign them a mode index $\mathcal{I}_M$ for $D_M = 2$ and $D_M = 3$ respectively. We will therefore introduce, in this paper, a general expressions of $\mathcal{I}_M$ for arbitrary $D_M$.

In the examples above, the attentive reader may have noticed that massless Dirac fermions with $D_M = 2$ can be described by a chiral symmetric Hamiltonian, similarly to zero-modes int $D_M = 0$ of Part I, while in contrast, the two other examples, corresponding to $D_M = 1$ and $D_M = 3$, belong to a different symmetry class. Actually, our formalism consistently reproduces the well-known staggered structure in the classification of topological insulating phases between the symmetry classes A and AIII with respect to the dimension $D$ of the system [21–23]. In the particular case where $\mathcal{I}_M$ describes the existence of boundary states of a (strong) topological insulator, the total dimension of the system verifies $D = D_M + 1$ and one recovers consistently the shifted structure of the standard classification for those two symmetry

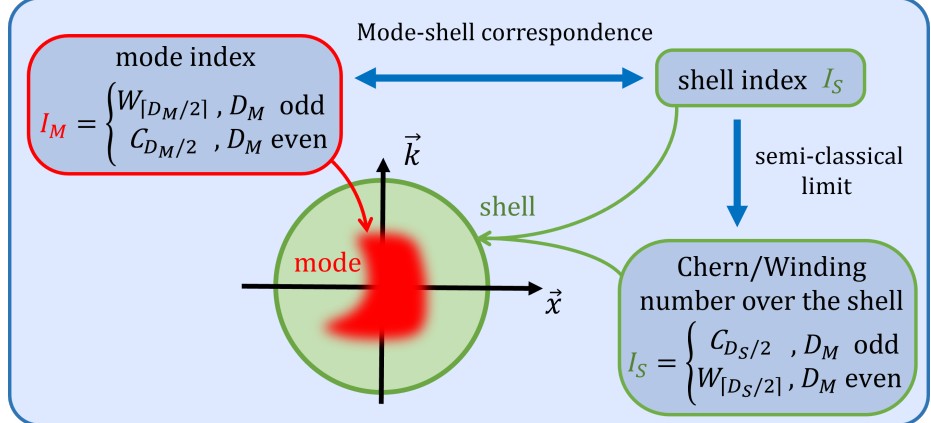

Figure 2: Brief summary of the mode-shell correspondence. A spectral index $\mathcal{I}_M$, characterizing the existence of gapless modes of a parameterised operator $\hat{H}(\boldsymbol{\lambda})$ and that disperse along $D_M$ directions, is introduced. This index is then re-expressed as an index $\mathcal{I}_S$ that only takes value on a shell (green) that surrounds the isolated gapless mode (sketched in red) in phase space where $\hat{H}(\boldsymbol{\lambda})$ is gapped. This shell index can be approximated, through a semi-classical expansion, as either a $D_S/2$-Chern number or a $\lceil D_S/2 \rceil$-winding number depending on the parity of $D_M$, and accordingly, on that of $D_S$ too. Those indices are properties of the Wigner-Weyl symbol $H(\boldsymbol{\lambda}, \boldsymbol{x}, \boldsymbol{k})$ of $\hat{H}(\boldsymbol{\lambda})$ and characterize topologically the gapless modes of $\hat{H}(\boldsymbol{\lambda})$ in phase space.

classes[1].

The strength of our formalism is that, although $\mathcal{I}_M$ can be seen as an edge index in the case where the gapless mode is a boundary mode, it is not restricted to this interpretation. Actually, $\mathcal{I}_M$ more generally counts gapless modes that disperse along $D_M$ directions *in phase space*, and that are confined in a specific region of phase space, i.e. position space, wavenumber space or a mix of both. For instance, for $D_M = 2$, $\mathcal{I}_M$ accounts for massless Dirac states which are either *bulk* modes in graphene, and thus confined in *wavenumber space*, or *surface* modes of 3D chiral symmetric topological insulators, which are confined in *position space*. So, the same mode index should capture the topological modes of both $2D$-semi-metals and $3D$-topological insulators (see the column $D_M = 2$ of figure 1).

Similarly to the case $D_M = 0$ discussed in Part I, we show how to re-express the mode indices into shell indices $\mathcal{I}_S$ for $D_M > 0$. The advantage is that shell indices are prone to a semi-classical expansion that simplifies their expression into more familiar integral formulas over the shell, that is a surface of dimension $D_S$ in phase space. Those semi-classical expressions are more likely computable analytically than their $\mathcal{I}_M$ counterpart. For $D_M = 0$, we showed in Part I that the semi-classical shell invariant associated to chiral zero-modes is a winding number $W$ in phase space. In this Part II, we show that the semi-classical shell invariant is also a higher dimensional winding number $W_{\lceil D_S/2 \rceil}$ when $D_M$ is even, and a Chern number $C_{D_S/2}$ when $D_M$ is odd. Here the subscripts $\lceil D_S/2 \rceil$ and $D_S/2$ label the rank of the higher winding or Chern numbers. More specifically, $\lceil D_S/2 \rceil$ designates the round up integer above $D_S/2$ when $D_S$ is odd. The possible winding numbers $W_{\lceil D_S/2 \rceil} = W_1, W_2, W_3 \ldots etc$ are respectively given by

---

[1]Note that we shall not discuss other symmetry classes (real classes) in this paper, such as time-reversal symmetric Hamiltonians in class AII that also display massless Dirac fermions as surface states, and instead focus only on A and AIII classes (complex classes).

an integral in $D_S = 1, 3, 5 \ldots etc$ dimensions in phase space. Similarly, $D_S$ is always even when $D_M$ is odd. In that case, the semi-classical invariants $C_{D_S/2} = C_1, C_2, C_3 \ldots etc$ are then the first, second, third $\ldots etc$ Chern numbers obtained from an integral over $D_S = 2, 4, 6 \ldots etc$ dimensions in phase space. A brief summary of our theory is sketched in figure 2, and detailed in section 4.

As we have just recalled, it is worth stressing here again that gapless modes of dimension $D_M$ all have the same general mode index expression $\mathcal{I}_M$, but may differ by their shell index $\mathcal{I}_S$. The reason is that the shell that surrounds them can span a different part of phase space, depending on the nature of the mode (bulk mode, boundary mode, corner mode...). This results into winding numbers and Chern numbers given by integrals over wavenumber space, position space, or a combination of both. In particular, the bulk-boundary correspondence [7,8,24–32] is recovered when the gapless modes are localized on a boundary, such that the shell then corresponds to the Brillouin zone. In that case, the semi-classical shell index is exclusively given by an integral over wavenumber space and coincides with the usual *bulk* topological invariant. In other words, the usual bulk-boundary correspondence is a particular case of the mode-shell correspondence. The interest of our formalism is that it also captures the topological nature of gapless modes confined in phase space in general, and not only those confined in one direction of position space. A standard situation where this generalization is for instance needed is that of continuous waves systems where the topology of gapless modes localized at a domain wall is related to a shell invariant defined on a sphere in phase space [9,28–30,33–38] instead of the usual Brillouin zone which is missing in the absence of a lattice (see respectively subfigures c and b of figure 5).

Actually, if the topology of the shell (Brillouin torus, sphere...) is not fixed once $D_M$ is given, neither is its dimension $D_S$. In the case of *strong* topological insulators that we mentioned above, $D_M = D - 1$ is the dimension of the boundary which is one less than the dimension of the bulk invariant, so that $D_S = D_M + 1 = D$. As an example, for a Chern insulator, the system is $D = 2$-dimensional, the chiral edge states correspond to $D_M = 1$ modes, and the shell corresponds to the Brillouin zone, so $D_S = 2$. But in general, the gapless mode may not not be a boundary mode, e.g. in topological semi-metals or higher-order topological insulators. The relation between $D$, $D_M$ and $D_S$ is more generally given by

$$D_M + D_S = 2D - 1 . \tag{1}$$

This relation can be seen as a kind of Kleman-Toulouse formula $d' + r = d - 1$ originally used to classify topological defects in ordered phases of matter [39]. In the original formula, $d \leftrightarrow 2D$ is the dimension of the system which is here is the dimension of phase space, $d' \leftrightarrow D_M$ is the dimension of the defect, and $r \leftrightarrow D_S$ is the dimension of the *cage* that surrounds the defect. Our shell can thus be seen as the cage of a topological defect in phase space, while the defects are the gapless modes, separated in phase space from the other modes by a gapped region.

The relation (1) provides us with a way to classify systems hosting topological modes, as it allows us to account for other interesting physical situations than that of strong topological insulating phases. In particular, it accounts for weak and higher order topological insulators as already discussed in Part I for $D_M = 0$. In different ways, those two kinds of topological phases share the property to host boundary modes whose dimension $D_M$ is such that $D_M + 1 < D$. In comparison with the strong topological insulating phases, the dimension $D_S$ of the shell must increase to satisfy (1). More precisely, the relation $D_M + n = D_S - n + 1 = D$ defines $n^{\text{th}}$ order topological insulating phases in phase space. Strong topological insulating phases are recovered for $n = 1$, while weak [40–49] and higher-order topological insulators [50–52] both correspond to $n > 1$ (see figure 1). Meanwhile, topological semi-metals correspond

to $n = 0$, i.e. $D_M = D_S + 1 = D$. For instance, a Weyl node in $D = 3$ disperses along $D_M = 3$ directions and is characterized by the first Chern number given by the integral of the Berry curvature over a $D_S = 2$ surface that encloses the Weyl node in reciprocal space [10, 11]. It is the strength of our formalism that all these situations can be understood in a unified way.

In the rest of the paper, the increment of dimension $n$ introduced above to distinguish the different phases will be noted $D_\perp$, as it is the complement to $D_M$ to recover the dimension of the system, i.e. $D_M + D_\perp = D$. In many situations, $D_\perp$ can be interpreted as the number of spatial directions $x_i$ along which the gapless modes do not disperse, but are, on the contrary, localized. For example, $D_\perp = 0$ for topological bulk excitations, $D_\perp = 1$ for topological modes localised at edges and $D_\perp \geq 2$ for corner or hinges modes of higher-order insulators, as summarized in figure 1. The mode dimension $D_M$ remains important as it intervenes in the dimension on the shell $D_S = D_M + 2 D_\perp$ and so will affect the semi-classical limit of the shell invariant.

The generalization to arbitrary $D_M$ of the mode-shell analysis requires some precision about the Hamiltonian operator $\hat{H}$, since the system is now almost systematically multidimensional. It thus depends a priori on $D$ pairs of non-commutative canonical conjugate variables $\boldsymbol{x} = (x_1, \cdots, x_D)$ and $\boldsymbol{\partial_x} = (\partial_{x_1}, \cdots, \partial_{x_D})$, and we should mean $\hat{H}(\boldsymbol{x}, \boldsymbol{\partial_x})$ when referring to the operator Hamiltonian $\hat{H}$. Using the Wigner-Weyl transform (see e.g. appendix B [1] for an introduction), we can also define the symbol Hamiltonian

$$\text{Continuous case: } H(\boldsymbol{x}, \boldsymbol{k}) = \int dx'^d \left\langle \boldsymbol{x} + \boldsymbol{x}'/2 \right| \hat{H} \left| \boldsymbol{x} - \boldsymbol{x}'/2 \right\rangle e^{-i\boldsymbol{k}\boldsymbol{x}'}$$
$$\text{Discrete case: } H(\boldsymbol{x}, \boldsymbol{k}) = \sum_{\boldsymbol{x}'} \left\langle \boldsymbol{x} + \boldsymbol{x}' \right| \hat{H} \left| \boldsymbol{x} \right\rangle e^{-i\boldsymbol{k}\boldsymbol{x}'} \tag{2}$$

where position $\boldsymbol{x} = (x_1, \cdots, x_D)$ and wavenumber $\boldsymbol{k} = (k_1, \cdots, k_D)$ coordinates now commute. This picture will be particularly useful to define the shell invariant. To define the mode index, we will use an intermediary picture using *partial* Wigner-Weyl transform in only $D_M \leqslant D$ directions, such that we are left with a *partial* operator $\hat{H}(\{x_i, k_i\}, \{x_j, \partial_{x_j}\})$, where $i$ runs from 1 to $D_M$ and $j$ runs from 1 to $D_\perp$. This partial operator constitutes actually the starting point of our analysis in the main text, and will therefore be simply referred to as the operator in the main text. We shall use the simplified notation $\hat{H}(\boldsymbol{\lambda})$ where $\boldsymbol{\lambda} = (\lambda_1, \cdots, \lambda_{D_M})$ denotes the parameters along which the dispersion relation is defined, and we shall not specify the $D_\perp$ other directions in that notation. In a strict sense, $\lambda_i = k_i$, as it will be in most examples, but in full generality, the energy/frequency states may vary with respect to other parameters, such as classical coordinates $x_i$ or external parameters. Thus, to summarize, the mode indices $\mathcal{I}_M$ introduced in the main text are associated to $\hat{H}(\boldsymbol{\lambda})$, and the semi-classical analysis will be carried out over the $D_\perp$ dimensions left. The symbol Hamiltonian, indifferently noted $H(\boldsymbol{\lambda}, \boldsymbol{x}_\perp, \boldsymbol{k}_\perp)$ or $H(\boldsymbol{x}, \boldsymbol{k})$ is less ambiguous than its Weyl operator counter-part as all the variables are classical and commute.

Of course, it is possible to define the mode indices at the full operator level, that is from $\hat{H}(\boldsymbol{x}, \boldsymbol{\partial_x})$, before any Wigner transform in a given direction is performed. Such an approach is particularly useful to tackle topological modes in disordered systems where the semi-classical analysis is not suited, and so are neither the semi-classical shell invariants. We thus also provide such a *non-commutative* expression of the mode invariants for arbitrary $D_M$. Since those expressions are more involved as that associated to the partial operator Hamiltonian $\hat{H}(\boldsymbol{\lambda})$, their discussion is postponed to appendix C.

This paper is organized as follows: In section 2, we introduce the mode invariant for the case $D_M = 1$,

that is the spectral flow, and derive the mode-shell correspondence. We then show that the semiclassical shell index is the $D_S/2$-th Chern number. In section 3, we discuss various examples hosting spectral flows, from 1D metals to 2D Chern insulators and 3D higher-order and weak topological insulators. Next, we generalise the approach to arbitrary $D_M$ in section 4, where we discuss the general structure that relates the mode index to the semiclassical shell index, according to $D_M$, $D_S$ and $D$, and the symmetry class (A or AIII). We finally apply this general theory in section 5 to models hosting $D_M = 2$ and 3 gapless modes. The examples cover lattice and continuous models, and illustrate the mode-shell correspondence on strong, weak and higher-order topological phases as well as in topological semi-metals.

## 2  $D_M = 1$ : Spectral flow and mode-shell correspondence

This section is dedicated to gapless modes in the case $D_M = 1$. Those modes sometimes refer to the spectral flow in the literature [9, 35, 36]. We will thus refer to $\mathcal{I}_M$ as the *spectral flow index* all over this section and write indifferently $\mathcal{I}_M = \mathcal{I}_{\text{s-f}}$ .

### 2.1  Spectral flow index

The spectral flow is defined as a property of the spectrum of a Hamiltonian operator $\hat{H}(\lambda)$ which depends on one parameter $\lambda$. Physically, such a parameter can either have an external origin, as in a pumping process, or designate a coordinate in phase space. In this case, the most common example is the wave number coordinate $\lambda = k$. This situation is encountered both in continuous wave systems, for which $k \in \mathbb{R}$, and in lattice models where $k$ is the Bloch quasi-momentum $k \in \mathbb{S}^1$ and is instead bounded. [2].

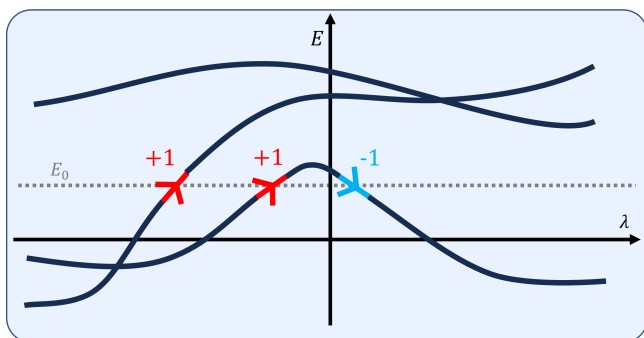

Figure 3: Example of evolution of the energies of the modes of a Hamiltonian depending on $\lambda$. The positive crossing of the energy level $E_0$ from below to above are denoted in red and the negative crossing from above to below are denoted in blue. The overall spectral-flow index of such system is therefore $\mathcal{I}_{\text{s-f}} = 2 - 1 = 1$

The spectral flow is then defined as the number of eigenenergies $E(\lambda)$ of $\hat{H}(\lambda)$ that algebraically cross a reference energy $E_0$ when varying $\lambda$, that is

$$\mathcal{I}_{\text{s-f}}(E = E_0) = (\#\text{crossings from } E < E_0 \text{ to } E > E_0) - (\#\text{crossings from } E > E_0 \text{ to } E < E_0). \quad (3)$$

---

[2]In fact, translation invariance is not strictly required to invoke $\lambda = k_i$, as such a continuous parameter can in principle follow more generally from a valid Wigner-Weyl transform when a semi-classical limit makes sense in the conjugate direction $x_i$.

A simple illustration is sketched in Figure 3. By construction, $\mathcal{I}_{\text{s-f}}$ is obviously an integer. However it is not obvious that it is a continuous function of the Hamiltonian $\hat{H}(\lambda)$ that is robust to deformations. So, similarly to the chiral number of zero-modes discussed in the part I, we want to formulate an equivalent but smooth formulation of the spectral index.

## 2.2 Smooth formulation of the index

To define a smooth version of the spectral index, we introduce, in the same fashion as in Part I [1], a smoothly flatten version of the Hamiltonian, $\hat{H}_F(\lambda) = f(\hat{H}(\lambda))$ that has the same eigenmodes as $\hat{H}$ but with a smooth flattening of the energies to $\pm 1$ above some threshold $|E - E_0| < \Delta$, and smoothly interpolating in between (see figure 4, where $E_0 = 0$). Typically, we choose the gap threshold $\Delta$ such that there is only a finite number of modes of $\hat{H}(\lambda)$ which are gapless. Next, we introduce the unitary $\hat{U}(\lambda) \equiv -e^{i\pi\hat{H}_F(\lambda)}$ to re-express the spectral flow as the winding number of $\hat{U}(\lambda)$ when $\lambda$ spans either $\mathbb{R}$ or $\mathbb{S}^1$. Indeed, if a mode $n$ participates positively to the spectral flow, it means that its energy $E_n(\lambda)$ bridges continuously the gap that separates states of energy $E < E_0 - \Delta$ to the states of energy $E > E_0 + \Delta$, when varying $\lambda$. Its rescaled energy $f(E_n(\lambda))$ then varies continuously from $-1$ to $+1$, so that the unitary eigenvalue $u_n \equiv -e^{i\pi f(E_n(\lambda))}$ winds counterclockwise once around the unit circle, yielding a winding number of $+1$ (see figure 4). Similarly, a mode that contributes negatively to the spectral flow has an energy that bridges the gap in the opposite way when sweeping $\lambda$, so that it contributes to a winding of $-1$. In contrast, gapped modes with an energy above the gap $E > E_0 + \Delta$ have a rescaled energy that maps to $+1$ on the unit circle, and therefore do not yield any winding contribution. In the same way, gapped modes with an energy below the gap $E < E_0 - \Delta$ have a rescaled energy that also maps to $+1$ on the unit circle, and do not contribute neither. The net spectral flow thus corresponds to the sum of the winding numbers $\frac{1}{2i\pi}\int_\Lambda u_n^\dagger \partial_\lambda u_n$ of all the rescaled energies of all the nodes $n$, that reads

$$\mathcal{I}_{\text{s-f}} = \frac{1}{2i\pi}\int_\Lambda d\lambda\, \text{Tr}\Big(\hat{U}^\dagger(\lambda)\partial_\lambda\hat{U}(\lambda)\Big) \equiv \mathcal{W}_1(\hat{U}) \tag{4}$$

with $\Lambda = \mathbb{R}$ or $\Lambda = \mathbb{S}^1$. Since $\hat{U} = -e^{i\pi f(\hat{H})}$ is a smooth function of $\hat{H}$, the index $\mathcal{I}_{\text{s-f}}$ is a continuous function of the Hamiltonian. It is therefore a topological integer stable to smooth deformations of $\hat{H}$.

However, in finite systems, similarly to what happens for zero-modes, the total winding number is often zero, due to several contributions – e.g. from opposite edges when $\mathcal{I}_M$ counts edge states – that cancel out. Therefore, one would like to select the crossings in a sub-region of phase space only (confined in position at an edge or an hinge, or confined in wavenumber). This can be done in a very similar way by adding a cut-off $\hat{\theta}_\Gamma$

$$\mathcal{I}_{\text{s-f}} = \mathcal{W}_1(\hat{U}) = \frac{1}{2i\pi}\int d\lambda\, \text{Tr}\Big(\hat{U}^\dagger(\lambda)\partial_\lambda\hat{U}(\lambda)\hat{\theta}_\Gamma\Big) \tag{5}$$

where $\hat{\theta}_\Gamma$ is the identity in the sub-region we want to select, and vanishes in the other gapless regions of $\hat{H}$ we wish to disregard. When the gapless regions are separated enough from each other in phase space by a region where $\hat{H}$ is gapped, and when $\hat{H}$ has a short-range behavior in the direction of separation, we expect the addition of the cut-off to not alter the quantisation of the winding number which thus remains an integer up to exponentially small deviations. This reasonable expectation is motivated by a similar result rigorously demonstrated in $1D$ chiral systems [53]. Similarly to what we discussed in Part I [1], the shape of the cut-off depends on the selected gapless modes confined in one (topological insulator) or several directions (higher-order insulator) in position, in wavenumber or in $\lambda$. These different choices of cut-offs implies different topologies for the shell which is defined as the intermediate region where

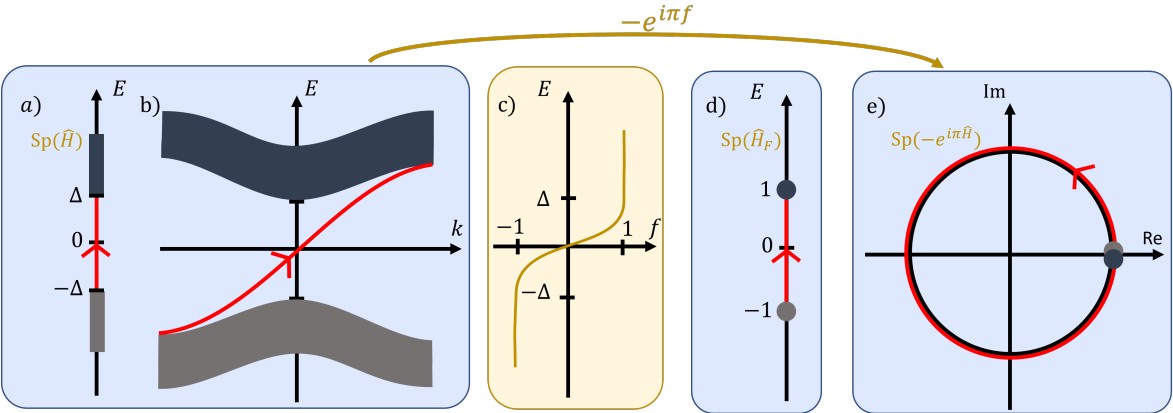

Figure 4: a) Projected energy spectrum $\mathrm{Sp}(\hat{H})$ of a typical topological Hamiltonian $\hat{H}(\lambda)$ and b) its full spectrum with respect to the parameter $\lambda = k$. United stripes denote the gapped bulk bands, the red line denotes a gapless mode confined at the edge with positive spectral flow. c) Sketch of a possible smooth flattening function. d) Spectrum of the operator $\hat{H}_F = f(\hat{H})$ where the bulk bands are flattened. e) Spectrum of $-e^{i\pi\hat{H}_F}$, where all gapped modes are mapped to 1 and the gapless mode of positive spectral flow is now mapped to the circle with positive winding number.

the cut-off goes from identity to zero. Wigner-Weyl representations of various spectral flow modes with different confinements in phase space and their associated shells are sketched in figure 5.

## 2.3 Relation with chiral symmetric systems.

It is interesting to observe that the formulation of the spectral index $\mathcal{I}_\mathrm{M}$ when $\mathrm{D}_\mathrm{M} = 1$, namely the spectral flow, that captures a gapless topological property, can be formulated as a winding number, while similarly winding numbers are also obtained in the context of semiclassical bulk invariants $\mathcal{I}_\mathrm{S}$, for gapped systems with chiral symmetry, as discussed in Part I. This is actually the manifestation of a quite general link between invariants capturing gapless topology and invariants describing gapped topology but in a different symmetry class that we will observe later in this article. This link can be made more explicit by defining the operator $\hat{H}'$ such that

$$\hat{H}' = \begin{pmatrix} 0 & -e^{-i\pi\hat{H}_F} \\ -e^{i\pi\hat{H}_F} & 0 \end{pmatrix} = -\sigma_x e^{-i\pi\sigma_z \otimes \hat{H}_F} \tag{6}$$

which is chiral symmetric ($\{\sigma_z, \hat{H}'\} = 0$) and gapped because $\hat{H}'^2 = \mathbb{1}$. Moreover its topology is linked to that of $\hat{H}$ as we have

$$\mathcal{I}_\text{s-f} = \frac{1}{4i\pi} \int d\lambda \, \mathrm{Tr}'(\sigma_z \hat{H}'(\lambda) \partial_\lambda \hat{H}'(\lambda) \hat{\theta}_\Gamma) \tag{7}$$

where $\mathrm{Tr}'$ denotes the trace on the new Hilbert space which is now twice bigger. Such an expression is similar to the expression of the bulk-index in 1D chiral systems encountered in Part I [1]. This relation between Hamiltonians in symmetry classes A and AIII will be used in the generalisation of our theory in section 4.

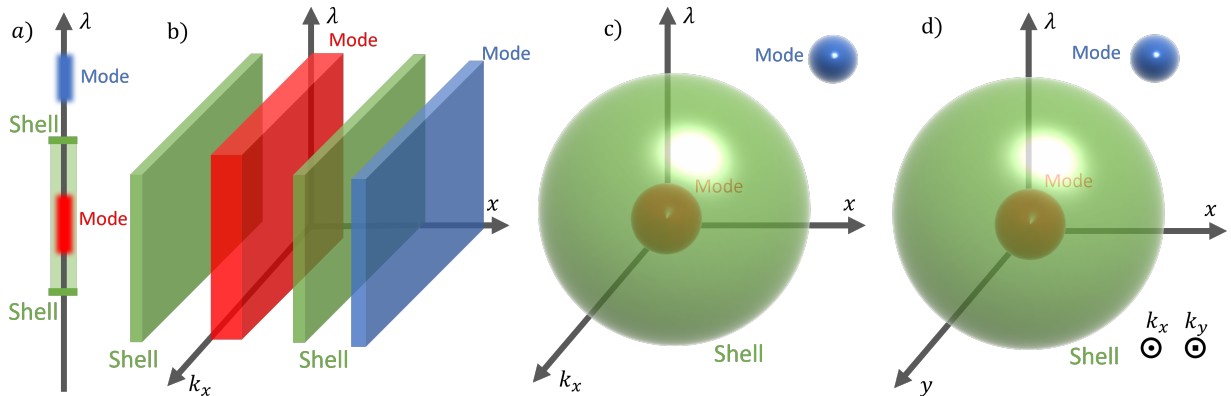

Figure 5: Sketches of shells (green) with different topologies and dimensions depending on the choice of the cut-off for $D_M = 1$. The shell encloses a selected gapless mode in phase space (red) and disregard possible other gapless modes (blue). a) For a cut-off acting in the direction of the spectral flow parameter $\lambda$ only (e.g. for $1D$ quantum channels, section 3.1)), the shell reduces to points. b) For a cut-off acting in position $x$ only (e.g. lattice models of Chern insulators, section 3.2), the shell consists in the Brillouin zone $(k_x, \lambda = k_y) \in [0, 2\pi]^2$. c) For a cut-off acring both in position $x$ and wavenumber $k_x, \lambda$ spaces (e.g. 2D continuous interface or the valley Quantum Hall effect, section 3.3), the shell is a sphere. d) For a cut-off acting in $x$ and $y$ directions (e.g. higher-order topological insulators, section 3.5), the shell is a $4D$-sphere.

## 2.4 Mode-shell correspondence for spectral flow modes

Similarly to the chiral number of zero-modes discussed in details in Part I [1], the spectral flow index also verifies a mode-shell correspondence. This means that it can be expressed, up to a rearranging of the terms, as an index which is defined on a shell surrounding the Wigner representation of the spectral flow mode in phase space, where $\hat{H}$ is gapped.

To make this apparent, we introduce the path of unitaries $\hat{U}_t = -e^{it\hat{H}_F}$ which interpolates between a trivial state $U_0 = \mathbb{1}$ of zero winding number and our target unitary $\hat{U}_1 = \hat{U}$. The introduction of this homotopy comes along with the chiral symmetric operator $\hat{H}'_t = \begin{pmatrix} 0 & \hat{U}^\dagger_t \\ \hat{U}_t & 0 \end{pmatrix}$. Then, if we differentiate the spectral flow-index (5) with respect to $t$ and integrate by part in $\partial_\lambda$, we obtain

$$
\begin{aligned}
\mathcal{I}_{\text{s-f}} &= \frac{1}{4i\pi} \int_0^\pi dt \int d\lambda \, \text{Tr}'(\hat{\sigma}_z \partial_t \left( \hat{H}'_t \partial_\lambda \hat{H}'_t \right) \hat{\theta}_\Gamma) \\
&= \frac{1}{4i\pi} \int_0^\pi dt \int d\lambda \, \text{Tr}'(\hat{\sigma}_z \left( \partial_t \hat{H}'_t \partial_\lambda \hat{H}'_t + \hat{H}'_t \partial_{t,\lambda} \hat{H}'_t \right) \hat{\theta}_\Gamma) \\
&= \frac{1}{4i\pi} \int_0^\pi dt \int d\lambda \, \text{Tr}'(\hat{\sigma}_z \left( \partial_t \hat{H}'_t \partial_\lambda \hat{H}'_t - \partial_\lambda \hat{H}'_t \partial_t \hat{H}'_t \right) \hat{\theta}_\Gamma) - \text{Tr}'(\hat{\sigma}_z \hat{H}'_t \partial_t \hat{H}'_t \partial_\lambda \hat{\theta}_\Gamma) .
\end{aligned}
\tag{8}
$$

Next, by using the identity $(\hat{H}'_t)^2 = \mathbb{1}$ and its differentiated version $\{\hat{H}'_t, \partial_{t/\lambda} \hat{H}'_t\} = 0$ we find the expression

$$
\mathcal{I}_{\text{s-f}} = \frac{-1}{4i\pi} \int_0^\pi dt \int d\lambda \frac{1}{2} \text{Tr}'(\hat{\sigma}_z \hat{H}'_t \partial_t \hat{H}'_t \left( \partial_\lambda \hat{H}'_t [\hat{\theta}_\Gamma, \hat{H}'_t] - [\hat{\theta}_\Gamma, \hat{H}'_t] \partial_\lambda \hat{H}'_t \right)) + \text{Tr}'(\hat{\sigma}_z \hat{H}'_t \partial_t \hat{H}'_t \partial_\lambda \hat{\theta}_\Gamma)
\tag{9}
$$

that takes values on the shell because all its terms contain either a commutator of the cut-off $[\hat{\theta}_\Gamma, \hat{H}_F]$

or a derivative $\partial_\lambda \hat{\theta}_\Gamma$. Since $\hat{H}_F^2 = \mathbb{1}$ in the shell, it follows that $-e^{it\hat{H}_F} = -\cos(t) - i\sin(t)\hat{H}_F$ and so $\hat{H}' = -\sigma_x \cos(t) - \sigma_y \sin(t)\hat{H}_F$, which allows us to express the spectral flow as a shell invariant $\mathcal{I}_S$ as

$$
\begin{aligned}
\mathcal{I}_{\text{s-f}} &= \frac{-1}{4i\pi} \int_0^\pi dt \int d\lambda\, i\sin(t)^2 \operatorname{Tr}\Big(\hat{H}_F\big(\partial_\lambda \hat{H}_F[\hat{\theta}_\Gamma, \hat{H}_F] - [\hat{\theta}_\Gamma, \hat{H}_F]\partial_\lambda \hat{H}_F\big)\Big) + 2i\operatorname{Tr}\Big(\hat{H}_F \partial_\lambda \hat{\theta}_\Gamma\Big) \\
&= -\int d\lambda \frac{1}{2}\operatorname{Tr}\Big(\hat{H}_F \partial_\lambda \hat{\theta}_\Gamma\Big) + \frac{1}{4}\operatorname{Tr}\Big(\hat{H}_F \partial_\lambda \hat{H}_F[\hat{\theta}_\Gamma, \hat{H}_F]\Big) \equiv \mathcal{I}_S
\end{aligned}
\tag{10}
$$

The shell invariant is particularly prone to a semi-classical expansion. When this approximation is performed, the shell index reduces to a (higher)-Chern number

$$
\mathcal{I}_{\text{s-f}} = \frac{1}{2^{2D+1}D!(-2i\pi)^D} \int_{\text{shell}} \operatorname{Tr}\big(H_F(dH_F)^{2D}\big) = \mathcal{C}_{D_S/2}
\tag{11}
$$

where $2D = D_S$ is the dimension of the shell which is in general even (otherwise, the Chern number is zero), and where $H_F$ is the Wigner-Weyl symbol of $\hat{H}_F$. When $D_S = 2$, $\mathcal{C}_1$ is the usual first Chern number. When $D_S \geq 4$, the shell invariant is a higher-Chern number and when $D_S = 0$, i.e. it consists in a set of points on which $\operatorname{Tr}(H_F)$ is just the number of positive energies minus the number of negative energies of $H$ at those points.

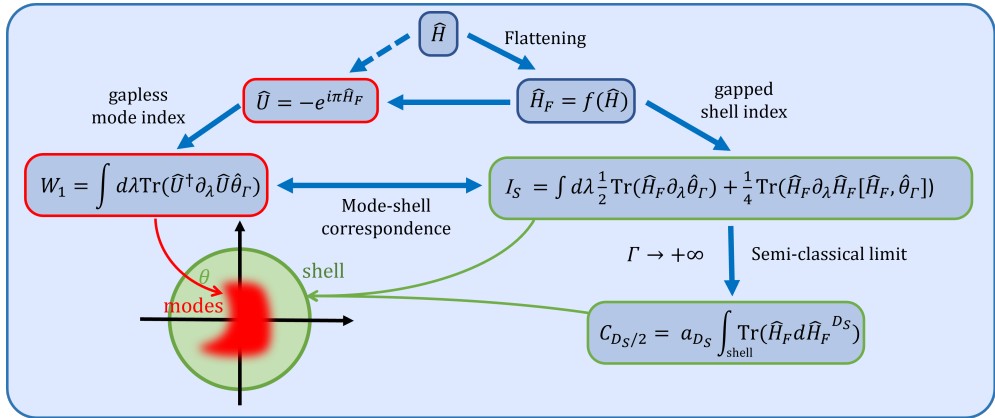

Figure 6: Summary diagram of the mode-shell correspondence when the mode invariant $\mathcal{I}_M$ is the spectral flow $\mathcal{I}_{\text{s-f}}$. We use a smoothly flatten version $\hat{H}_F$ of the Hamiltonian $\hat{H}$ and introduce a unitary $\hat{U} = -e^{i\pi\hat{H}_F}$ to define two indices: the winding number $\mathcal{W}_1$ associated to $\hat{U}$, that counts the spectral flow of $\hat{H}$, a gapless property, and $\mathcal{I}_S$ measuring a topological property of the gapped system on the boundary of dimension $D_S$ enclosing the gapless mode (namely the shell) in phase space. This shell index reduces, in a semi-classical limit, to a (higher) Chern number. The prefactor $a_{D_S}$ of $\mathcal{C}_{D_S/2}$ is given in (11). More specific examples of shells encircling the spectral flow modes are given in figure 5.

Instead of working with the flatten Hamiltonian $H_F$, one can also work with the Fermi projector $P$ on the negative bands of $H$. Since the Hamiltonian is gapped, we have the relation $H_F = 1 - 2P$ with $P = \sum_i |\psi_i\rangle \langle\psi_i|$ where $|\psi_i\rangle$ are the bands selected by the projector, and it follows that

$$
\mathcal{I}_{\text{s-f}} = \frac{-1}{D!(-2i\pi)^D} \int_{\text{shell}} \operatorname{Tr}\big(P(dP)^{D_S}\big) = \mathcal{C}_{D_S/2}
\tag{12}
$$

which is a usual expression of the Chern numbers when working with the Berry curvature formalism [11, 54–56].

# 3 Systems hosting a spectral flow and their mode-shell correspondence

In this section, we illustrate the mode-shell correspondence on various examples exhibiting a spectral flow ($D_M = 1$). Among the most common ones are the edge states of a Chern insulator (3.2) and its somehow continuous version consisting of a Dirac Hamiltonian with a space-varying mass term (3.3). Those two examples concern two-dimensional systems. We then elaborate from these examples to construct higher dimensional systems exhibiting a spectral flow, that consist of either weak (3.4) or higher order (3.5) topological insulators. But first, let us discuss a most simple example in $D = 1$, namely a quantum channel, that we interpret as a spectral flow and revisit through the mode-shell correspondence.

## 3.1 Spectral flows as 1D quantum channels ($D_M = 1, D_\perp = 0$)

Let us start with the most simple example which exhibits a spectral flow, the bulk of a 1D metal, which typically, hosts a spectral flow at each point of the Fermi surface. To illustrate this point, consider the simplest 1D Hamiltonian with only hopping terms $t$ between neighbouring sites

$$\hat{H}_{\text{QC}} = \sum_n t/2(|n\rangle \langle n+1| + |n+1\rangle \langle n|) . \tag{13}$$

This Hamiltonian is invariant by translation, so the Wigner-Weyl transform coincides with the Bloch transform here

$$\hat{H}_{\text{QC}}(k) = t\cos(k) . \tag{14}$$

Note that, after this transform, there is no operator left in this simple example ($D_\perp = 0$), so that the operator Hamiltonian to investigate the spectral flow coincides with its symbol. If we assume that the Fermi energy $E_F$, playing the role of $E_0$, is zero $E_F = 0$, we see that the spectrum of the Bloch Hamiltonian crosses twice the Fermi energy, at $k = -\pi/2$ and at $k = \pi/2$ (see figure 7). This Hamiltonian is therefore gapless and the associated material is a metal. Focusing our attention on those individual

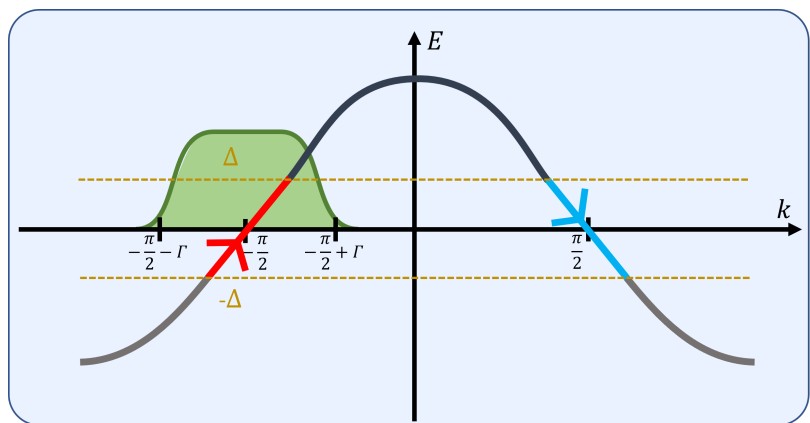

Figure 7: Schematic band dispersion of the 1D chain (13). The energy band crosses the zero energy line at $k = -\pi/2$ and $k = \pi/2$ with respectively a positive and negative spectral flow. A given spectral flow can be selected with a cut-off such as that displayed in green.

crossings, we see that the one at $k = -\pi/2$ is a crossing of positive spectral flow $\mathcal{I}_{\text{s-f}} = 1$ (with flow parameter $\lambda = k$) while the one at $k = \pi/2$ is associated to a negative spectral flow $\mathcal{I}_{\text{s-f}} = -1$. In that

case the cut-off $\hat{\theta}_\Gamma$ should act in wavenumber space (or quasi-momentum space) to separate one crossing from the other (see figure 7).

In the mean time, in the semi-classical limit, the shell invariant becomes

$$\mathcal{I}_S = -\int d\lambda \frac{1}{2} \operatorname{Tr}\left(\hat{H}_F \partial_\lambda \hat{\theta}_\Gamma\right) + \frac{1}{4} \operatorname{Tr}\left(\hat{H}_F \partial_\lambda \hat{H}_F [\hat{\theta}_\Gamma, \hat{H}_F]\right) \xrightarrow{S-C} -\int d\lambda \frac{1}{2} \operatorname{Tr}(H_F \partial_\lambda \theta_\Gamma) . \qquad (15)$$

If we introduce $\hat{P}(k) = (\mathbb{1} - \hat{H}_F(k))/2$, the Fermi projector on eigenspaces of energy below the Fermi energy $E_F = 0$, and take a sufficiently sharp cut-off $\hat{\theta}_\Gamma \to \mathbb{1}_{[k_0-\Gamma, k_0+\Gamma]}$ (where $k_0$ is the position of the crossing and $\Gamma$ is the radius of the shell) the above expression reduces to $\mathcal{I}_S = \operatorname{Tr}(P(k_0 - \Gamma)) - \operatorname{Tr}(P(k_0 + \Gamma))$. The shell invariant is therefore the number of negative band eigenvalues on the left minus the number of negative eigenvalues on the right. It can be check easily that we indeed have that $\mathcal{I}_S = 1$ around $k_0 = -\pi/2$ and $\mathcal{I}_S = -1$ around $k_0 = \pi/2$, and therefore the mode-shell correspondence is verified.

Both $\mathcal{I}_{\text{s-f}}$ and $\mathcal{I}_S$ remain quantised and topologically protected as long as there is a gap which separates the two Fermi points in phase space, but also as long as the Hamiltonian is short range in wavenumber. This is here insured by the fact that the Hamiltonian is invariant by translation and hence diagonal in wavenumber. In general, we do not need perfect invariance by translation and could allow the coefficients of the Hamiltonian to be slowly varying in position space (compared to the inter site distance). For example, phonons of large wavelength (compared to the lattice wavelength) induce only small deformations of the Hamiltonian, so the scattering they generate between the two valleys would be exponentially small. 1D ballistic conductors are examples of materials where the mean free-path associated to the scattering is small compared to the length of the material. In those materials, quantisation of the conductance, as predicted by Landaueur [57], was indeed observed [58]. The main problem in condensed-matter applications is that defects in the lattice structure (vacancies, impurities, ...) induce perturbations which do not vary slowly compared to the inter-site distance. This can therefore induce scattering between the two valleys and could therefore destroy the topological protection.

We discuss, in the next section, the case of Chern insulators where spectral flow modes are separated in position space. This provides an enhanced topological protection compared to that associated with the separation in wavenumber space discussed here.

## 3.2 Spectral flows as edge states of a 2D Chern insulator ($\mathrm{D_M} = 1, \mathrm{D_\perp} = 1$)

**Semiclassical invariant as a bulk Chern number in the Brillouin zone.** In this section, we show how the mode-shell correspondence coincides with the bulk-edge correspondence in the case of Chern insulators.

Consider a Hamiltonian operator $\hat{H}$ of a Chern insulator defined on a lattice, and let us call $x$ and $y$ the two spatial directions. For simplicity, we assume that the lattice is invariant by translation in the $y$ direction so that $k_y$ can serve as a spectral flow parameter $\lambda$. We then consider that the lattice has an edge/interface at $x = 0$ where the Hamiltonian, that becomes $\hat{H}(k_y)$, may have gapless edge/interface states and is gapped far away from it. The mode index $\mathcal{I}_{\text{s-f}}$ thus counts the number of such chiral edge/interface states with chirality at a given edge/interface, and in a given gap. We call $\hat{H}_\pm$ the bulk Hamiltonians respectively far to the right/left of the interface. Since we work on a lattice, wavenumbers must be understood as quasi-momenta which are bounded, and so one can choose a cut-off which acts only in position space, such as $\hat{\theta}_\Gamma = \exp(-x^2/\Gamma^2)$ or $\hat{\theta}_\Gamma = (1 + \exp(x^2 - \Gamma^2))^{-1}$ for concreteness. The transition region where the cut-off drops from one to zero consists of a domain in $(x, k_x, k_y)$ space such that $x \sim \Gamma$.

As a result, the shell is a 2D torus spanned by $k_x$ and $k_y$ and located far from the edge when $\Gamma \to \infty$, as depicted in figure 5 (b). In other words, the shell is nothing but the 2D Brillouin zone, and the semi-classical Hamiltonian $H(k_x, k_y)$ obtained in that limit describes the bulk of the system, and coincides with the Bloch Hamiltonian. In the case of an interface, one needs to specify two such bulk Hamiltonians $H_\pm(k_x, k_y)$ to designate the right/left bulks from the interface.

Since the cut-off does not depend on $\lambda$, the term $\text{Tr} \hat{H}_F \partial_\lambda \hat{\theta}_\Gamma$ in the expression of the shell invariant (10) vanishes, in contrast with the previous example. A semi-classical expansion of the other term is performed by replacing the commutator by a Poisson bracket, and the shell invariant becomes

$$\mathcal{I}_S = \frac{-1}{8i\pi} \int_0^{2\pi} dk_y \int_0^{2\pi} dk_x \sum_{n_x} \text{Tr}\big(H_F \partial_{k_x} H_F \partial_{k_y} H_F \delta_{n_x} \theta_\Gamma\big) . \tag{16}$$

Summing over the discrete lattice coordinates $n_x$ in the direction $x$, and using $\sum_{n_x > 0} \delta_{n_x} \theta_\Gamma = \theta_\Gamma(+\infty) - \theta_\Gamma(0) = -1$ and $\sum_{n_x < 0} \delta_{n_x} \theta_\Gamma = \theta_\Gamma(0) - \theta_\Gamma(-\infty) = 1$, this expression reduces to the difference of Chern numbers, obtained by an integration over the 2D-Brillouin zone, far to the right of the interface/edge, where $H_F = H_{F,+}$, and to the left where $H_F = H_{F,-}$

$$\begin{aligned}
\mathcal{I}_S &= \frac{-1}{8i\pi} \int_{[0,2\pi]^2} dk_x dk_y \, \text{Tr}\big(H_{F,+} \partial_{k_x} H_{F,+} \partial_{k_y} H_{F,+}\big) - \text{Tr}\big(H_{F,-} \partial_{k_x} H_{F,-} \partial_{k_y} H_{F,-}\big) \\
&= \frac{1}{i\pi} \int_{[0,2\pi]^2} dk_y dk_x \, \text{Tr}\big(P_+ \partial_{k_x} P_+ \partial_{k_y} P_+\big) - \text{Tr}\big(P_- \partial_{k_x} P_- \partial_{k_y} P_-\big) \\
&\equiv \mathcal{C}_+ - \mathcal{C}_-
\end{aligned} \tag{17}$$

where $P$ is the Fermi projector satisfying $H_F = 1 - 2P$. Through theses explicit computations, we therefore recover the expected bulk-edge correspondence [7, 8, 27], where the usual bulk invariant (here the Chern number) is obtained as the semi-classical limit of the shell index, that only involves an integral in reciprocal space in that case.

**Example 1: The Qi-Wu-Zhang model** We now more concretely discuss a well-known model of a Chern insulator that exhibits a spectral flow, namely the Qi-Wu-Zang (QWZ) lattice model [59] which is a 2D model with 2 pseudo-spin internal degrees of freedom. In a cylindrical geometry, it is given by the operator Hamiltonian

$$\begin{aligned}
\hat{H}(k_y) &= (\sigma_z \sin(k_y) + \sigma_x(M + \cos(k_y))) \sum_{n_x} |n_x\rangle \langle n_x| + \sigma_+ \sum_{n_x} |n_x + 1\rangle \langle n_x| + \sigma_- \sum_{n_x} |n_x\rangle \langle n_x + 1| \\
&= \begin{pmatrix} \sin(k_y)\mathbb{1} & \sum_{n_x} |n_x + 1\rangle \langle n_x| + (M + \cos(k_y))\mathbb{1} \\ \sum_{n_x} |n_x\rangle \langle n_x + 1| + (M + \cos(k_y))\mathbb{1} & -\sin(k_y)\mathbb{1} \end{pmatrix}
\end{aligned} \tag{18}$$

where $k_y$ is the quasi-momentum associated to the $y$ direction and $n_x$ is a lattice coordinate associated to the $x$ direction.

Spectral flow occurs in this model when the lattice has an edge with an open boundary condition. It manifests as a chiral edge state that bridges the two bands of this model, as shown numerically in figure 8. The simplicity of this model allows us to also evaluate analytically the mode index $\mathcal{I}_{\text{s-f}}$ by noticing that the Hamiltonian can be decomposed as a sum of a SSH Hamiltonian with a $k_y$-dependent term that breaks the chiral symmetry of the SSH model

$$\hat{H}(k_y) = \sigma_z \sin(k_y) + \hat{H}_{\text{SSH}}(k_y) \tag{19}$$

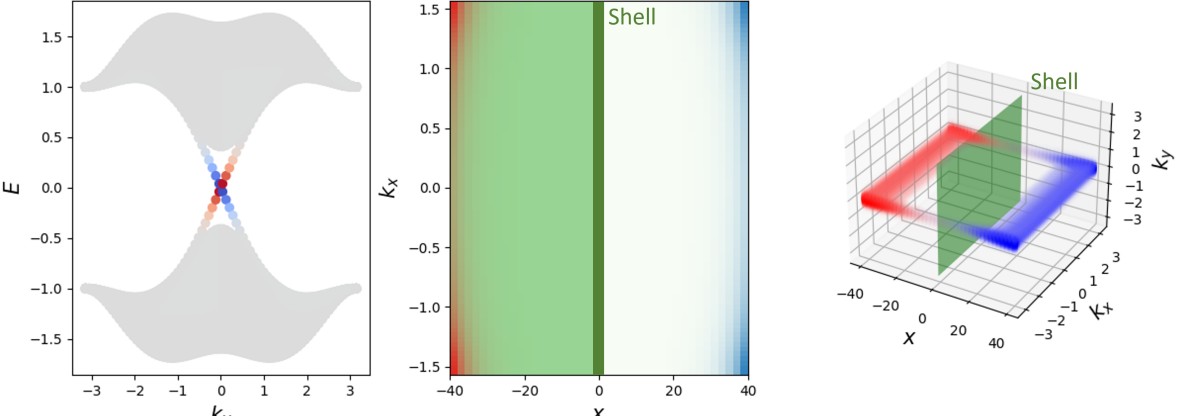

Figure 8: Numerical computation of the QWZ model (18) with $M = -1$. (Left) Dispersion relation along $k_y$ of $\hat{H}(k_y)$, where the associated states are colored in red/blue if they are both close to the Fermi energy $E = 0$ and located near the left/right edge, in grey otherwise. (Center) Wigner-Weyl transform of the two gapless modes of $\hat{H}(k_y)$ for $k_y = 0$. The region selected by the cut-off is denoted in light green, the intersection of the shell for $k_y = 0$ is shown in dark green. (Right) Wigner-Weyl transform of the modes of $\hat{H}(k_y)$ of energy close to the Fermi-energy. The shell is denoted in green and consists in a 2D Brillouin zone.

where $\hat{H}_{\mathrm{SSH}}(k_y)$ is the SSH Hamiltonian as discussed in Part I [1], with hopping amplitudes $t' = 1$ and $t = M + \cos(k_y)$, while $\sigma_z$ is the chiral operator associated to this SSH model $\{\sigma_z, \hat{H}_{\mathrm{SSH}}(k_y)\} = 0$. We therefore have

$$\hat{H}^2(k_y) = \sin(k_y)^2 + \hat{H}_{\mathrm{SSH}}(k_y)^2 \tag{20}$$

so we can only have modes $|\psi(k_y)\rangle$ crossing the zero-energy either if $k_y = 0$ or if $k_y = \pi$. Moreover, those modes must be zero-modes of the SSH model, i.e. $\hat{H}_{\mathrm{SSH}}(k_y)|\psi\rangle = 0$. With our knowledge of the SSH model (see e.g. Part I [1]), we know that zero-modes only appear when $|t'| = 1 > |M + \cos(k_y)| = |t|$ and are confined on the edges. We moreover know that, on the left edge, the zero-mode is of positive chirality $|\psi(k_y)\rangle = \begin{pmatrix} |\psi(k_y)\rangle_+ \\ 0 \end{pmatrix}$. It follows that, at first order in $k_y$ around the crossing point, we have

$$\hat{H}(k_y)|\psi(k_y)\rangle = \left[ \begin{pmatrix} \sin(k_y) & 0 \\ 0 & -\sin(k_y) \end{pmatrix} + \hat{H}_{\mathrm{SSH}}(k_y) \right] \begin{pmatrix} |\psi(k_y)\rangle_+ \\ 0 \end{pmatrix} \approx \pm k_y |\psi(k_y)\rangle \tag{21}$$

where $\pm 1 = +1$ when the crossing point is located at $k_y = 0$ and $-1$ when it is located at $k_y = \pi$. Therefore, the crossing point is associated respectively to a positive/negative spectral flow. It follows that the overall spectral flow depends on whether $\hat{H}_{\mathrm{SSH}}(k_y)$ is topological or not at $k_y = 0$ and at $k_y = \pi$. If it is topological at neither $k_y$, then there is clearly no spectral flow, $\mathcal{I}_{\mathrm{s\text{-}f}} = 0$. If it is topological at $k_y = 0$ but not at $k_y = \pi$, then there is a positive spectral flow, $\mathcal{I}_{\mathrm{s\text{-}f}} = +1$. Conversely, if it is topological at $k_y = \pi$ but not at $k_y = 0$, then there is a negative spectral flow, $\mathcal{I}_{\mathrm{s\text{-}f}} = -1$. Finally, if it is topological at both $k_y = 0$ and $\pi$, then the two local spectral flows cancel each other and the net spectral flow vanishes, $\mathcal{I}_{\mathrm{s\text{-}f}} = +1 - 1 = 0$. Doing such an analysis using that $\hat{H}_{\mathrm{SSH}}(k_y)$ is topological if and only if $|t'| > |t|$, we obtain the following phase diagram 9 depending on the $M$ parameter.

The semiclassical shell/bulk invariant is then derived from the symbol/Bloch transform $H = H(k_x, k_y)$

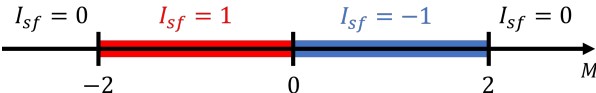

Figure 9: Values of the total spectral flow index with respect to the parameter $M$. When $|M| > 2$, $\hat{H}_{\text{SSH}}(k_y)$ is topologically trivial for both $k_y = 0$ and $k_y = \pi$ so there is no spectral flow. When $-2 < M < 0$, $\hat{H}_{\text{SSH}}(k_y)$ is topological at $k_y = 0$ and trivial at $k_y = \pi$ leading to a positive spectral flow. When $0 < M < 2$, $\hat{H}_{\text{SSH}}(k_y)$ becomes trivial at $k_y = 0$ while becoming topological at $k_y = \pi$ leading to a negative spectral flow.

of the operator Hamiltonian $\hat{H}(k_y)$ that reads

$$H(k_x, k_y) = \begin{pmatrix} \sin(k_y) & e^{-ik_x} + M + \cos(k_y) \\ e^{ik_x} + M + \cos(k_y) & -\sin(k_y) \end{pmatrix} , \tag{22}$$

from (16). This Chern number can be computed either using the degree formula [60] or numerical methods to evaluate it [61]. With both methods, one consistently recovers the same diagram as in figure 9.

### 3.3 Spectral flows as domain wall interface states in 2D continuous inhomogeneous systems $(\mathrm{D_M} = 1, \mathrm{D_\perp} = 1)$

**Semiclassical shell invariant as a Chern monopole in phase space.** Another well-known situation where a spectral flow appears is that of anisotropic two-dimensional continuous systems, where a "mass term", driving the amplitude of a gap of the symbol Hamiltonian $H(x, k_x, k_y)$, varies in space and changes sign [35,36,38,62–70]. We apply the mode-shell correspondence in such a situation, where the semiclassical shell invariant becomes a Chern number over a sphere in phase space, thus providing an alternative demonstration of the relation between this Chern monopole and the spectral flow [9, 28–30, 33–38, 63, 69, 70].

We consider a continuous model with a mass term varying in the $x$ direction and changing sign once, thus forming a gap closure domain wall in the $y$ direction. The spectral flow parameter $\lambda$ is given by the wavenumber $k_y \in \mathbb{R}$, which is unbounded in continuous models. Therefore, one needs to consider a cut-off which not only accounts for the confinement of the interface states in the transverse direction $x$, but that also selects the spectral flow mode in wavenumbers $k_x$ and $\lambda = k_y$. A concrete choice could be for example $\hat{\theta}_\Gamma = (1 + \exp(x^2 - \partial_x^2 + \lambda^2 - \Gamma^2))^{-1}$, whose symbol is $\theta_\Gamma = (1 + \exp(x^2 + k_x^2 + \lambda^2 - \Gamma^2))^{-1}$. The shell is located at the transition region where the cut-off varies from 1 to 0, and that spreads over $x^2 + k_x^2 + \lambda^2 \approx \Gamma^2$. The shell is therefore now a sphere in phase space $(x, k_x, \lambda)$ of radius $\Gamma$ (see figure 5.c). Note that this is different from the lattice case discussed above where the shell is a Brillouin zone in $k$-space localized at bulk position in $x$.

The semi-classical limit of the shell index (10) gives [3]

$$\mathcal{I}_{\text{S}} = \frac{-1}{i16\pi} \int_{x^2 + k_x + \lambda^2 = \Gamma^2} \text{Tr}\big(H_F(dH_F)^2\big) \tag{23}$$

with differential $dH_F = \partial_x H_F dx + \partial_{k_x} H_F dk_x + \partial_\lambda H_F d\lambda$.

---

[3]Such semi-classical limit is more difficult to derive than its discrete counter-part of the previous section. This is due to the fact that in the expression of the shell index (10), the terms in $\text{Tr}\big(\hat{H}_F \partial_\lambda \hat{\theta}_\Gamma\big)$ no longer vanishes and its semi-classical limit needs additional treatment. See section 4.3 for more detail.

**Example 2: Generalised Jackiw-Rebbi model/2D Dirac fermion with varying mass.** A canonical and simple two-band continuous model that exhibits a spectral flow is given by the operator

$$\hat{H}(k_y) = \begin{pmatrix} k_y & x + \partial_x \\ x - \partial_x & -k_y \end{pmatrix} \tag{24}$$

sometimes dubbed "normal form", whose spectrum is recalled in figure 10. This minimal model has been discussed in details and used in several studies, see e.g. [35,36,62,64]. It has several interpretations such as, for instance, the Hamiltonian of a two-dimensional spin-1/2 Dirac fermion with a varying mass term $m(x) \sim x$. The spectral flow then corresponds to a unidirectional mode that propagates along the $y$ direction where $m(x)$ changes sign, and the spectral flow parameter is then given by $\lambda = k_y$ in that case. This model is simple enough for both mode and shell indices to be analytically computed.

Let us start with the mode index. Similarly to the spectral flow of the QWZ lattice model that we inferred from a construction implying the lower dimensional SSH model in section 3.2, we can also deduce analytically the spectral flow in this 2D continuous model from a similar construction built on the simplest, chiral symmetric, 1D Jackiw-Rebbi model discussed in Part I

$$\hat{H}_{\mathrm{JR}} = \begin{pmatrix} 0 & x + \partial_x \\ x - \partial_x & 0 \end{pmatrix} . \tag{25}$$

Indeed, the Hamiltonian (24) is nothing but a generalized Jackiw-Rebbi Hamiltonian (25), where a term $k_y$ proportional to the chiral symmetric operator $\sigma_z$ of $\hat{H}_{\mathrm{JR}}$ has been added. Since $k_y\sigma_z$ anticommutes with $\hat{H}_{\mathrm{JR}}$, it follows that if a mode $|\psi(k_y)\rangle$ crosses zero-energy, then this crossing must occur at $k_y = 0$, and moreover, $|\psi(k_y = 0)\rangle$ must be a chiral zero-mode of $\hat{H}_{\mathrm{JR}}$ captured by the mode index for $\mathrm{D_M} = 0$. Then, since the Jackiw-Rebbi model has exactly one zero-mode $(e^{-x^2/2}/\sqrt{2\pi}, 0)^t$ of positive

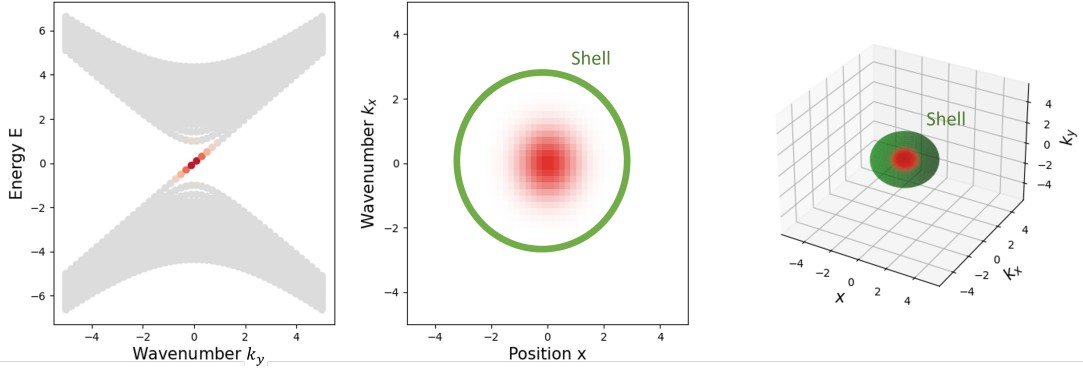

Figure 10: Numerical computation of the modified Jackiw-Rebbi model (25). (Left) Dispersion relation in $k_y$ of the eigenstates of $\hat{H}(k_y)$. Modes are colored in red if they are both close to the Fermi energy $E = 0$ and located close to $x = 0$, in grey otherwise. (Center) Wigner-Weyl transform of the gapless mode of $\hat{H}(k_y)$ in $k_y = 0$, the intersection of the shell for $k_y = 0$ is denoted in green. (Right) Wigner-Weyl transform of the modes of $\hat{H}(k_y)$ of energy close to the Fermi-energy. The shell is a sphere and is displayed in green.

chirality, there is therefore exactly one crossing. Such a crossing is thus associated to a positive spectral flow when increasing $\lambda$, and thus $\mathcal{I}_{\text{s-f}} = +1$, in agreement with a brut force calculation (see figure 10).

To compute the shell index analytically, we first need to compute the symbol Hamiltonian which simply reads

$$H = \begin{pmatrix} k_y & x + ik_x \\ x - ik_x & -k_y \end{pmatrix}. \tag{26}$$

The two band eigenvalues satisfying $E^2 = k_x^2 + k_y^2 + x^2$, we can write the flatten Hamiltonian as

$$H_F = \frac{1}{\sqrt{k_x^2 + k_y^2 + x^2}} \begin{pmatrix} k_y & x + ik_x \\ x - ik_x & -\lambda \end{pmatrix}. \tag{27}$$

The shell being the 2-sphere here, it is convenient to introduce spherical coordinates $k_y = \Gamma \cos(\theta), x = \Gamma \sin(\theta) \cos(\phi), k_x = \Gamma \sin(\theta), \cos(\phi)$ so that

$$H_F = \begin{pmatrix} \cos(\theta) & \sin(\theta)e^{i\phi} \\ \sin(\theta)e^{-i\phi} & -\cos(\theta) \end{pmatrix}. \tag{28}$$

One can then compute $\mathrm{Tr}(H_F dH_F \wedge dH_F) = -4i \sin(\theta) d\theta \wedge d\phi$ which, once integrated on the 2-sphere, gives $\mathcal{I}_S = \frac{-1}{16i\pi} \int_0^\pi d\theta \int_0^{2\pi} d\phi 4\sin(\theta) = +1$, in agreement with the mode index. The mode-shell correspondence is thus satisfied.

## 3.4 Weak topological insulators and their stacked topology ($D_M = 1, D_\perp = 2$)

Weak topological insulators can be obtained by stacking many copies of the same topological system. By doing so, one can construct topological systems with a large number of boundary modes, that are also captured by the mode-shell correspondence. In this formalism, the stacking procedure consists in increasing $D_\perp$ that becomes larger than 1, while keeping $D_M$ constant. This consistently increases the dimension $D$ of the system since $D_M + D_\perp = D$.

In this section, we provide two examples to illustrate the mode-shell correspondence for weak topological insulators with $D_M = 1$ and $D = 3$, i.e. $D_\perp = 2$. The two models are mathematically constructed following a so-called multiplicative tensor product construction that we discussed in Part I [1]. This procedure provides a simple and systematic way to construct models for weak topology and that can be analyzed from their lower dimensional sub-blocks, here 1D and 2D models exhibiting a spectral flow that we already discussed above.

**Example 3: Stack of 2D Chern insulators** A simple model with such a phenomenology is made of $N$ uncoupled stacked layers of some QWZ Chern insulator which reads

$$\hat{H}(k_y) = H_{QWZ}(k_y) \otimes \mathbb{1}_N$$
$$= \begin{pmatrix} \sin(k_y)\mathbb{1} & \sum_{n_x} |n_x + 1\rangle \langle n_x| + (M + \cos(k_y))\mathbb{1} \\ \sum_{n_x} |n_x\rangle \langle n_x + 1| + (M + \cos(k_y))\mathbb{1} & -\sin(k_y)\mathbb{1} \end{pmatrix} \otimes \sum_{n_z=1}^{N} |n_z\rangle\langle n_z| \tag{29}$$

where $m$ is the index of the layer and $N$ is the number of layers. This model has a spectral flow index $\mathcal{I}_{\text{s-f}} = N$. This result is straightforward for uncoupled layers, and can be extended to coupled layers as long as the inter-layer couplings are not large enough to close the bulk gap. Indeed, the (macroscopic) spectral flow, being a topological invariant, it is preserved by such perturbations.

**Example 4: Stack of 1D metallic chains forming a 3D metal with disjoint Fermi-surface**
One can also consider the stacking of uncoupled 1D quantum channel of section 3.1, which can be done in two transverse directions

$$\hat{H} = \hat{H}_{\mathrm{QC}} \otimes \mathbb{1}_N \otimes \mathbb{1}_N$$
$$= \sum_n t/2 \left( |n\rangle \langle n+1| + |n+1\rangle \langle n| \right) \otimes \sum_{m,m'=1}^{N} |m,m'\rangle\!\langle m,m'| \tag{30}$$

where $m$ and $m'$ are the indices of the layers in the two transverse directions. This model has therefore a spectral flow index of $\mathcal{I}_{\text{s-f}} = N^2$. Similarly to the previous example, considering uncoupled chains is just for the sake of having quick-to-solve models. One can add inter-layer couplings without breaking the topological protection. The only constraint is that the coupling between the layers must not be too large to close the gap in the region separating the modes of opposite group velocity.[4] In the case of the stack of 1D chains, the Fermi surface must keep the property of being split into two distinct connected components, one of positive group velocity, and one of negative group velocity.

In those two previous examples, the spectral flow is of order $N$ or $N^2$, so that the associated conductance is proportional to $N$ or $N^2$ in units of the quantum conductance $\hbar/e^2$. It is thus much larger than the conductance of a single layer which can be interesting for electronic transport purposes. The main challenge to the use of those dissipation-less currents is still the conditions required to obtain these unidirectional zero modes with topologically suppressed scattering (induced by the separation in phase space). In the case of the quantum hall effect or of Chern insulators, this requires quite low temperature or/and high magnetic field [71–73]. In the case of the 1D chain, the topological protection provided by the separation in wavenumber is known to be weaker [58]: It may suppress the scattering by perturbation (like phonon) of low wavenumber but is still sensible to the scattering of abrupt perturbations (like defects or short range interaction) which would generate dissipation. Therefore, the material would need to be particularly pure to obtain good transport properties.

## 3.5 Higher-order topological insulators $(\mathrm{D_M} = 1, \mathrm{D_\perp} = 2)$

Higher-order topological modes are confined in more than one direction. Although they are physically two disctinct phases, weak topological insulating phases and certain higher order topological insulating phases, actually correspond to the same $\mathrm{D_\perp} > 1$ and $\mathrm{D_M}$ in the framework of the mode-shell correspondence. Their topology is therefore very similar.

Also, similarly to gapless modes of weak topological insulators, higher-order topological modes can also be constructed from spectral flow modes. Indeed, there exists a procedure to generate simple-to-analyze models that display higher-order topological properties. We dubbed *additive tensor product construction* this method in Part I for chiral symmetric systems $(\mathrm{D_M} = 0)$ and re-discuss it here for the case $\mathrm{D_M} = 1$. The reader who wishes to maintain focus on the mode-shell correspondence may skip this section and proceed directly to Example 5.

---

[4]This means that the coupling between the atoms must be quite anisotropic, strong inside the layer, and weaker between them.

**Additive tensor product construction ⊞**

Let us consider two Hamiltonians $\hat{H}_A$ and $\hat{H}_B$ where $H_A$ is chiral symmetric with chiral symmetry operator $\hat{C}_A$. Then we introduce the additive tensor product construction ⊞ as

$$\hat{H}_⊞ = \hat{H}_A \otimes \mathrm{Id} + \hat{C}_A \otimes \hat{H}_B \tag{31}$$

Unlike the additive construction of Part I, $\hat{H}_⊞$ is not chiral symmetric. However, one preserves the useful relation

$$\hat{H}_⊞^2 = \hat{H}_A^2 \otimes \mathrm{Id} + \mathrm{Id} \otimes \hat{H}_B^2 + \{\hat{H}_1, \hat{C}_1\} \otimes \hat{H}_2 = \hat{H}_A^2 \otimes \mathrm{Id} + \mathrm{Id} \otimes \hat{H}_B^2 \tag{32}$$

so that the spectrum of $\hat{H}_⊞$ is still of the form $\pm\sqrt{(E_n^A)^2 + (E_m^B)^2}$ with $E_n^{A/B}$ eigenenergies of $\hat{H}_{A/B}$. In particular, the zeros modes of $\hat{H}_⊞$ are a product $|\psi_n^A\rangle \otimes |\psi_m^B\rangle$ of zero modes of $\hat{H}_{A/B}$ which is confined in both the gapless region of $\hat{H}_A$ and that of $\hat{H}_B$.

One important use of this construction is to take a chiral higher-order insulator and generate a higher-order insulator with spectral flow. In this case we want to take an Hamiltonian $\hat{H}$ with chiral symmetry $\hat{C}$ that has topological zero mode and transform it into an Hamiltonian with topological spectral flow. We use the additive construction using $\hat{H}_A = \hat{H}$ and $\hat{H}_B = \lambda$ which generate the Hamiltonian $\hat{H}'$.

$$\hat{H}_⊞(\lambda) = \lambda \boxplus \hat{H} \equiv \lambda\hat{C} + \hat{H} \tag{33}$$

In the section 3.3, we already encounter a model of this type and argued that each zero-mode of positive chirality $\mathcal{I}_M = 1$ of $\hat{H}$ is associated to a mode of positive spectral flow $\mathcal{I}_S = 1$ of $\hat{H}_⊞$ and vice-versa for zero-modes of negative chirality $\mathcal{I}_M = -1$ which are associated to a negative spectral flow $\mathcal{I}_S = -1$.

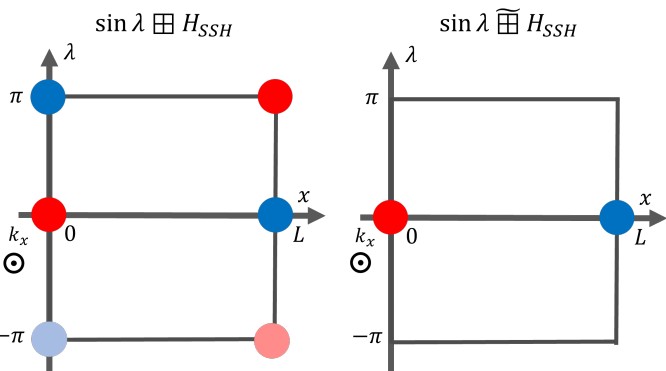

Figure 11: Localisation of the zero modes in phase space depending on the construction used. Modes zero-modes with positive/negative spectral flow are denoted in red/blue. (left) regular additive construction of an SSH model using 34 (right) modified additive construction of an SSH model using 35 .

In those constructions, the spectral flow dimension is unbounded. If we want a bounded spectral flow parameter $\lambda \in \mathcal{S}^1$ as for wavenumber in discrete model, we need to tweak such an expression. One way is to do replace $k$ by $\sin(k)$ as

$$\hat{H}_⊞(\lambda) = \sin(\lambda) \boxplus \hat{H} = \sin(\lambda)\hat{C} + \hat{H}. \tag{34}$$

This construction creates another zero mode of opposite spectral flow at $k = \pi$ and which is therefore separated only in wavenumber (see left of figure 11). If one wants to have a model where the zero-modes

are only separated in position, it is possible to use a slightly more involved construction that we designate by $\tilde{\boxplus}$. Fixing $C = \sigma_z$ for simplicity, the new Hamiltonian is defined as

$$\hat{H}_{\tilde{\boxplus}}(\lambda) = \sin(\lambda)\sigma_z + \sigma_x + (\hat{H} - \sigma_x)(1 + \cos(\lambda))/2 . \tag{35}$$

It exhibits the same spectral flow mode at $\lambda = 0$ as $\hat{H}_{\boxplus}(\lambda) \underset{\lambda \sim 0}{\approx} \lambda\sigma_z + \hat{H}$. Moreover it is gapped for $\lambda \neq 0$ as $\sin(\lambda) \neq 0$ and has no spectral flow modes at $k = \pi$ because $\hat{H}_{\tilde{\boxplus}}(\pi) = \sigma_x$ so that the Hamiltonian is now gapped (see right of figure 11). This is similar to the QWZ model we encountered in section 3.2. We use the notation $\hat{H}_{\tilde{\boxplus}} = \sin(k)\tilde{\boxplus}\hat{H}$ to refer to this useful construction.

If we instead take for $\hat{H}$ a chiral higher-order insulator, we obtain a higher-order insulator with spectral flow.

**Example 5: Continuous model of a 3D higher-order insulator**  A continuous model of a 3D higher-order topological insulator can be obtained from the 2D Jackiw-Rossy Hamiltonian $\hat{H}_{2D-JR}$ (see example 1 of section 4.3 of part I [1]) to which we add a $\lambda\sigma_z \otimes \sigma_z$ term ($\sigma_z \otimes \sigma_z$ is the chiral operator of the Jackiw-Rossy model) to transform it into a model with positive spectral flow $\mathcal{I}_M = 1$. This gives the Hamiltonian

$$\hat{H}(k_z) = k_z \boxplus \hat{H}_{2D-JR} = \begin{pmatrix} k_z & y - \partial_y & x - \partial_x & 0 \\ y + \partial_y & -k_z & 0 & x - \partial_x \\ x + \partial_x & 0 & -k_z & -(y - \partial_y) \\ 0 & x + \partial_x & -(y + \partial_y) & k_z \end{pmatrix} . \tag{36}$$

The Jackiw-Rossy Hamiltonian can itself be decomposed into two blocks of Jackiw-Rebbi Hamiltonians $\hat{H}_{2D-JR} = \hat{H}_{JR} \boxplus \hat{H}_{JR}$. Therefore, the full Hamiltonian can be decomposed as the sum of 3 simple building blocks

$$\hat{H} = k_z\sigma_z \otimes \sigma_z + \hat{H}_{jr}(x, \partial_x) \otimes \text{Id} + \sigma_z \otimes \hat{H}_{jr}(y, \partial_y) \equiv k_z \boxplus \hat{H}_{JR} \boxplus \hat{H}_{JR} . \tag{37}$$

The symbol of this operator Hamiltonian simply reads

$$H = \begin{pmatrix} k_z & y - ik_y & x - ik_x & 0 \\ y + ik_y & -k_z & 0 & x - ik_x \\ x + ik_x & 0 & -k_z & -(y - ik_y) \\ 0 & x + ik_x & -(y + ik_y) & k_z \end{pmatrix} . \tag{38}$$

The shell is here the 4-sphere $k_z^2 + x^2 + y^2 + k_x^2 + k_y^2 = \Gamma^2$ (see figure 5.d), so it makes sense to introduce spherical coordinates in four dimensions. In those coordinates, the symbol of the flatten Hamiltonian is

$$H_F = \cos(\theta_2) \left( \cos(\theta_1) \begin{pmatrix} 0 & e^{-i\phi_1} \\ e^{i\phi_1} & 0 \end{pmatrix} \otimes \text{Id} + \sigma_z \otimes \sin(\theta_1) \begin{pmatrix} 0 & e^{-i\phi_2} \\ e^{i\phi_2} & 0 \end{pmatrix} \right) + \sin(\theta_2)\sigma_z \otimes \sigma_z . \tag{39}$$

The shell index in that case is the second Chern number, that is obtained from the integration, over the 4-sphere, of the 4-form curvature $\text{Tr}^{\text{int}}(H_F dH_F^4)$. The calculation can be carried out analytically and one finds $\mathcal{I}_S = -1/(256\pi^2) \int_{\mathcal{S}^4} \text{Tr}^{\text{int}}(H_F dH_F^4) = 1$. The mode-shell correspondence is thus verified.

**Example 6: Lattice model of 3D higher-order insulator**  Let us construct a lattice model of a higher-order topological insulator in $D = 3$ with spectral flow modes propagating along the hinges, by a similar procedure, where we now start with the BBH model discussed in the example 2 of section 4.3 of part I [1]

$$\hat{H}(k_z) = \sin(k_z) \,\tilde{\boxplus}\, \hat{H}_{\text{BBH}}. \tag{40}$$

Since the BBH model is itself a sum of SSH Hamiltonians $\hat{H}_{\text{BBH}} = \hat{H}_{\text{SSH},x} \boxplus \hat{H}_{\text{SSH},y}$, the Hamiltonian (40) can be decomposed into three elementary blocks

$$
\begin{aligned}
\hat{H}(k_z) =\,& \sin(k_z)\sigma_z \otimes \sigma_z + \sigma_z \otimes \sigma_x (1 - \cos(k_z))/2 \\
\equiv\,& \sin(k_z) \,\tilde{\boxplus}\, \hat{H}_{\text{SSH},x} \,\boxplus\, \hat{H}_{\text{SSH},y} \\
& + (1 + \cos(k_z))/2 \left( \begin{pmatrix} 0 & t + t'T_x^\dagger \\ t + t'T_x & 0 \end{pmatrix} \otimes \text{Id} + \sigma_z \otimes \begin{pmatrix} 0 & t + t'T_y^\dagger \\ t + t'T_y & 0 \end{pmatrix} \right)
\end{aligned}
\tag{41}
$$

where $T_x = \sum_{n_x} |n_x + 1\rangle \langle n_x|$ is the translation operator by one lattice length in the $x$ direction. Because of the property of the additive construction, we therefore know that $\mathcal{I}_{\text{M}} = 1$ when $|t'| > |t|$.

It can also be decomposed as a sum of a QWZ Hamiltonian with an SSH Hamiltonian

$$\hat{H}(k_z) = (\sin(k_z) \,\tilde{\boxplus}\, \hat{H}_{\text{SSH},x}) \,\boxplus\, \hat{H}_{\text{SSH},y} = \hat{H}_{\text{QWZ,xz}} \,\boxplus\, \hat{H}_{\text{SSH,y}}. \tag{42}$$

Similarly to the BBH or the QWZ model which are its building blocks, the shell invariant is difficult to obtain explicitly, without using the mode-shell correspondence or using numerical methods. Evaluating the shell invariant $\mathcal{I}_{\text{S}}$ numerically, one can find that $\mathcal{I}_{\text{S}} = 1$ when $|t'| > |t|$, recovering therefore the mode-shell correspondence. However the important message is that tensor constructions are powerful tools to create, easy to analyse, model of non trivial topology $\mathcal{I}_{\text{s-f}} \neq 0$.

## 4  Mode-shell correspondence for arbitrary $D_{\text{M}}$

In this section, we generalize the construction of the mode index $\mathcal{I}_{\text{M}}$ for arbitrary mode dimension $D_{\text{M}}$. This construction depends on the parity of $D_{\text{M}}$ which is here in a one-to-one correspondence with the symmetry class of $\hat{H}$, either A or AIII. We then discuss how the mode invariants are related to their shell invariants by providing explicit formulas, and finally show their semi-classical expression in phase space. A summary of our theory is sketched in figure 12 and the derivation of the expressions of the invariants is provided in appendices A and B.

### 4.1  General construction of the mode invariant

We now discuss how to construct the mode indices $\mathcal{I}_{\text{M}}$ for arbitrary $D_{\text{M}}$. Let us recall that the starting points to address the cases $D_{\text{M}} = 0$ and $D_{\text{M}} = 1$ treated in details in Part I and in section 2 of Part II, were the operator Hamiltonians $\hat{H}$ and $\hat{H}(\lambda)$ respectively. Our task is now to construct $\mathcal{I}_{\text{M}}$ from an operator $\hat{H}(\boldsymbol{\lambda})$ parameterised by $D_{\text{M}}$ parameters $\boldsymbol{\lambda} = (\lambda_1, \cdots, \lambda_{D_{\text{M}}})$. Similarly to the spectral flow, those parameters can either have an external origin, as for a pump, or designate a coordinate in phase space. In particular, we are motivated by the cases $D_{\text{M}} = 2$ and $D_{\text{M}} = 3$ where the parameters are wavenumbers $\lambda_i = k_i$, coming from a Fourier/Bloch/Wigner transform of the operator $\hat{H}$ along $D_{\text{M}} \leqslant D$ directions. We will see specifically in section 5 that the mode indices for $D_{\text{M}} = 2$ and $D_{\text{M}} = 3$ capture the number of $2D$-Dirac cones and $3D$-Weyl cones in the spectrum of the Hamiltonian.

Although our original motivation is to provide a mode index that accounts for $2D$ Dirac and $3D$ Weyl cones as topological modes on the same footing as chiral zero-modes and spectral flows such as chiral edge states, it turns out that the generalization of the expression of $\mathcal{I}_M$ to arbitrary $D_M$ does not yield additional significative difficulties. Actually, the two cases $D_M = 0$ and $D_M = 1$ previously investigated are very instructive as they already unveil a pattern for the definition of $\mathcal{I}_M$ that will only depend on the parity of $D_M$ and will thus repeat straightforwardly to higher mode dimension $D_M$, that is

- when $D_M$ is even, as for zero-modes and $2D$ Dirac cones, we shall consider Hamiltonian operators with a chiral symmetry $\hat{C}$ i.e. such that $\{\hat{H}, \hat{C}\} = 0$ (class AIII)[5]

- when $D_M$ is odd, as for spectral flows and $3D$-Weyl cones, we shall consider Hamiltonian operators with no particular symmetry (class A).

In both cases, we aim at describing gapless modes that are surrounded by a gapped region of $\hat{H}(\boldsymbol{\lambda})$ in phase space.

The strategy we followed to define a mode index for $D_M = 1$ in section 2.1 was to introduce a unitary operator whose associated homotopy invariant (5) gave us the required spectral index (after an appropriate cut-off was added). We build on this strategy for odd $D_M$ by introducing an auxiliary chiral symmetric Hamiltonian $\hat{H}' = \hat{H}'(\boldsymbol{\lambda})$ whose off-diagonal blocks are those unitaries (and Hermitian conjugate). This systematic construction allows us to use both the winding properties of the unitaries, as we did to tackle the spectral flow, and express the mode invariant in terms of a chiral symmetric Hamiltonian, as we did for the zero-modes. In a complementary manner, when $D_M$ is even, the operator Hamiltonian we consider to start with is already chiral symmetric. Still, we shall nonetheless also introduce an auxiliary Hamiltonian $\hat{H}'$ which this time is not chiral symmetric. The purpose of this manipulation is to establish a uniform formalism with a clear pattern between the two symmetry classes and with the dimension. The two constructions are given by

$$
\begin{aligned}
\hat{H} \text{ non-chiral symmetric (class A):} \quad &\hat{H}' \equiv \begin{pmatrix} 0 & -e^{-i\pi\hat{H}_F} \\ -e^{i\pi\hat{H}_F} & 0 \end{pmatrix} = -\sigma_x e^{-i\pi\sigma_z \otimes \hat{H}_F} \\
\hat{H} \text{ chiral symmetric (class AIII):} \quad &\hat{H}' \equiv \sin\left(\pi\hat{H}_F\right) - \hat{C}\cos\left(\pi\hat{H}_F\right) = -\hat{C}e^{-\pi\hat{C}\hat{H}_F}.
\end{aligned}
\tag{43}
$$

Note that the construction $\hat{H} \to \hat{H}'$ switches the symmetry classes A and AIII. This trick will allow us to express the mode invariant for both symmetry classes A and AIII in an almost similar form in terms of the auxiliary Hamiltonian, for any $D_M$.

Let us now establish a useful property of the auxiliary Hamiltonians $\hat{H}'$. Since beyond the gapless domain, i.e. where $\hat{H}$ is gapped, we have by construction $\hat{H}_F^2 = \mathbb{1}$, then on can infer that in this domain $-e^{\pm i\pi\hat{H}_F} = \mathbb{1}$. It follows that, in this gapped region, $\hat{H}' = \begin{pmatrix} 0 & 1 \\ 1 & 0 \end{pmatrix}$ for $\hat{H}$ in class A, and $\hat{H}' = \hat{C}$ for $\hat{H}$ in class AIII. As a consequence, $\hat{H}'$ is topologically trivial in the gapped region of $\hat{H}$, or equivalently, the topologically non-trivial behavior of $\hat{H}'$ is concentrated in the selected gapless region of $\hat{H}$.

Besides, in both cases, $\hat{H}'$ is built in such a way that $\hat{H}'^2 = \mathbb{1}$, so that it is a gapped operator. We can thus apply the known theory of topological invariants on $\hat{H}'$ in order to express the mode index $\mathcal{I}_M$ associated to $\hat{H}$ as a (higher) winding number $\mathcal{W}$ or Chern number $\mathcal{C}$ as

---

[5]Dirac cones can also be protected by other symmetries [74–78], but we do not consider those here.

$$\mathcal{I}_\mathrm{M} = \begin{cases} \mathcal{W}_{\lceil \mathrm{D_M}/2 \rceil}(\hat{H}') & = & a_{\mathrm{D_M}} \displaystyle\int \mathrm{Tr}\Big( \hat{\sigma}_z \hat{H}' (d\hat{H}')^{\mathrm{D_M}} \hat{\theta}_\Gamma \Big) & \hat{H} \text{ in class A, } \mathrm{D_M} \text{ odd} \\[2ex] \mathcal{C}_{\mathrm{D_M}/2}(\hat{H}') & = & b_{\mathrm{D_M}} \displaystyle\int \mathrm{Tr}\Big( \hat{H}' (d\hat{H}')^{\mathrm{D_M}} \hat{\theta}_\Gamma \Big) & \hat{H} \text{ in class AIII, } \mathrm{D_M} \text{ even} \end{cases} \tag{44}$$

where the integral runs over the $\mathrm{D_M}$ parameters $\lambda_i$ with typically $\lambda_i \in \mathbb{S}^1$ or $\lambda_i \in \mathbb{R}$. $\lceil \mathrm{D_M}/2 \rceil$ denotes the round up integer above $\mathrm{D_M}/2$ when $\mathrm{D_M}$ is odd and is thus equal to $(\mathrm{D_M}+1)/2$. $d\hat{H}' = \sum_i \partial_{\lambda_i} \hat{H}' d\lambda^i$ is a differential 1-form, and $(d\hat{H})^{\mathrm{D_M}}$ is a $\mathrm{D_M}$-differential form that yields an antisymmetrized sum of all possible products of derivatives $\partial_{\lambda_i}$. For example, in $\mathrm{D_M} = 2$, $(d\hat{H})^2$ yields the term $\partial_{\lambda_1}\hat{H}\partial_{\lambda_2}\hat{H} - \partial_{\lambda_2}\hat{H}\partial_{\lambda_1}\hat{H}$. The operator $\hat{\theta}_\Gamma$ is a cutoff operator that selects a particular gapless region of phase space of size $\Gamma$, similarly to the spectral flow case. Finally, the pre-factor coefficients read

$$a_{\mathrm{D_M}} = \frac{-((\mathrm{D_M}+1)/2)!}{(\mathrm{D_M}+1)!(-2i\pi)^{(\mathrm{D_M}+1)/2}} \tag{45}$$

$$b_{\mathrm{D_M}} = \frac{-1}{2^{\mathrm{D_M}+1}(\mathrm{D_M}/2)!(-2i\pi)^{\mathrm{D_M}/2}} \ . \tag{46}$$

With this convention, $\mathcal{W}_1$ and $\mathcal{C}_1$ are, respectively, the usual winding number and the first Chern number, while $\mathcal{W}_\alpha$ and $\mathcal{C}_\alpha$ for $\alpha > 1$ are their higher-dimensional generalizations.

**Remark about the mode index for $\mathrm{D_M} = 0$.** We would like to comment about the consistency between the expressions of the mode index given by (44) for $\mathrm{D_M} = 0$ and that introduced in part I [1] that was dedicated to this case. The readers interested in the expressions of the shell invariant in the semi-classical limit in higher dimension can skip this paragraph.

According to (44), the mode index for $\mathrm{D_M} = 0$ in the AIII symmetry class should read $\mathcal{I}_\mathrm{M} = -\frac{1}{2}\mathrm{Tr}\,\hat{H}'\hat{\theta}_\Gamma$. Actually, to match the definition of Part I, it is convenient to slightly change this definition by subtratcting the chiral operator as

$$\mathcal{I}_\mathrm{M} = -\frac{1}{2}\mathrm{Tr}\Big( (\hat{H}' - \hat{C})\hat{\theta}_\Gamma \Big) \ , \tag{47}$$

the difference between the two expressions being the term $\frac{1}{2}\mathrm{Tr}\Big( \hat{C}\hat{\theta}_\Gamma \Big)$. This term, proportional to the chiral polarization [1, 79, 80], vanishes when the Hilbert space has a balanced chirality. It is specific to $\mathrm{D_M} = 0$, and was therefore not included in our general formalism for arbitrary $\mathrm{D_M}$ in this Part II. In fact, the mode indices for higher $\mathrm{D_M}$'s in the AIII class could be unharmly redefined with this subtraction. But this modification would make the expressions unnecessarily complicated since this term is only relevant for $\mathrm{D_M} = 0$. Thus, we prefer to simply slightly change the definition for $\mathrm{D_M} = 0$ only.

Let us now check that the definition (47) is consistent with that of the mode index given in Part I, that is $\mathcal{I}_\mathrm{M} = \mathrm{Tr}\Big( \hat{C}(1 - \hat{H}_F^2)\hat{\theta}_\Gamma \Big)$. Substituting $\hat{H}'$ by its expression (43) leads to

$$\mathcal{I}_\mathrm{M} = \mathrm{Tr}\Big( \hat{C}\frac{1}{2}(\cos\Big(\pi\hat{H}_F\Big) + 1)\hat{\theta}_\Gamma \Big) \tag{48}$$

which has a similar form as the mode index introduced in part I, up to the substitution of $1 - \hat{H}_F^2$ by $\frac{1}{2}(\cos\Big(\pi\hat{H}_F\Big) + 1)$. Actually, both expressions capture the zero-modes because, in the common eigenbasis

of $\hat{H}^2$ and $\hat{C}$ (which commute thanks to chiral symmetry), both expressions are of the form

$$\mathcal{I}_{\mathrm{M}} = \sum_\lambda C_\lambda g(E_\lambda) \langle \psi_\lambda | \hat{\theta}_\Gamma | \psi_\lambda \rangle \tag{49}$$

where $g(E_\lambda)$ is an even function of $E_\lambda$, is equal to one for zero eigenvalues, and vanishes for eigenvalues outside the gap $|E| \geq \Delta$ Because of this last property, we are thus left with the zero modes we would like to keep, and *a priori* other in-gap but non-zero modes. However, the latest come by pairs of opposite chirality, due to chiral symmetry. Indeed $|\psi\rangle$ is an eigenmode of both $\hat{H}^2$ and $\hat{C}$ with eigenvalues $E_\lambda^2$ and $C_\lambda$ then $H|\psi\rangle$ is also an eigenmode with eigenvalues $E_\lambda^2$ and $-C_\lambda$. Moreover, as $\hat{H}$ is short range, $H|\psi\rangle$ is in the same region of phase space as $|\psi\rangle$, so with the same value of the cut-off. Therefore, both contributions of $|\psi\rangle_\lambda$ and $H|\psi\rangle_\lambda$ cancel out two by two in the sum. The only contributions that remain are those of the zero-energy modes $\hat{H}|\psi_\lambda\rangle = 0$ that do not allow a valid way to construct a symmetric partner of opposite chirality. So we end up with $\mathcal{I}_{\mathrm{M}} = \sum_{\lambda, E_\lambda=0} C_\lambda g(0) \langle \psi_\lambda | \hat{\theta}_\Gamma | \psi_\lambda \rangle$. As, $g(0) = 1$, this is exactly the chirality of the zero-modes located inside the cut-off. Therefore, even if the functions $g(E_\lambda)$ differ, both expressions evaluate the same topological quantity and are therefore equivalent. In the following, we shall refer to the mode index for $\mathrm{D_M} = 0$ as $\mathcal{C}_0$.

## 4.2  Mode-shell correspondence and its semi-classical limit

The mode indices (44) satisfy also a mode-shell correspondence

$$\mathcal{I}_{\mathrm{M}} = \mathcal{I}_{\mathrm{S}} \tag{50}$$

for arbitrary $\mathrm{D_M}$, where $\mathcal{I}_{\mathrm{S}}$ is a topological invariant defined on a shell of dimension $\mathrm{D_S}$ in phase space where $\hat{H}$ is gapped. We already proved the case $\mathrm{D_M} = 1$ of this correspondence in section 2.4, the general derivation of this correspondence is provided in appendix A, and leads to the following expression for the shell invariant

$$\mathcal{I}_{\mathrm{S}} = \begin{cases} -b_{\mathrm{D_M}-1} \int \left( \mathrm{Tr}\left( \hat{H}_F (d\hat{H}_F)^{\mathrm{D_M}-1} d\hat{\theta}_\Gamma \right) + \mathrm{Tr}\left( \hat{H}_F (d\hat{H}_F)^{\mathrm{D_M}} [\hat{\theta}_\Gamma, \hat{H}_F] \right) \right) & \text{class A, } \mathrm{D_M} \text{ odd} \\ -a_{\mathrm{D_M}-1} \int \left( \mathrm{Tr}\left( \hat{C}\hat{H}_F (d\hat{H}_F)^{\mathrm{D_M}-1} d\hat{\theta}_\Gamma \right) + \mathrm{Tr}\left( \hat{C}\hat{H}_F (d\hat{H}_F)^{\mathrm{D_M}} [\hat{\theta}_\Gamma, \hat{H}_F] \right) \right) & \text{class AIII, } \mathrm{D_M} \text{ even} \end{cases} \tag{51}$$

This mode-shell correspondence is depicted by a horizontal double arrow in the summary of figure 12. When those shell invariants admit a semi-classical limit (in a sense defined in Part I), an important result is that they reduce to a Chern number or a winding number which are given by an integral over the shell as

$$\mathcal{I}_{\mathrm{S}} = \begin{cases} b_{\mathrm{D_S}} \int_{\mathrm{shell}} \mathrm{Tr}\left( H_F (dH_F)^{\mathrm{D_S}} \right) & = & \mathcal{C}_{\mathrm{D_S}/2}(H_F) & \text{class A, } \mathrm{D_M} \text{ odd} \\ a_{\mathrm{D_S}} \int_{\mathrm{shell}} \mathrm{Tr}\left( C H_F (dH_F)^{\mathrm{D_S}} \right) & = & \mathcal{W}_{\lceil \mathrm{D_S}/2 \rceil}(H_F) & \text{class AIII, } \mathrm{D_M} \text{ even} \end{cases} . \tag{52}$$

Those expressions also correspond to Chern and winding numbers, but the integral now runs over the shell and the operator they are associated to is the semi-classical Wigner-Weyl symbol of $\hat{H}$. Thus, in that semi-classical limit, the number of gapless modes of $\hat{H}$ that disperse along an odd number of directions, is given by the $\mathrm{D_S}/2$-Chern number, while the number of gapless modes that disperse along

an even number of directions is given by a $(D_S+1)/2$-winding number. The specific order of the Chern and winding numbers depends on the dimension of the shell, which is itself fixed by the set dimension of the system $D$ and that of the mode $D_M$. Indeed, according to the relation (1), a $D_S/2$-Chern number is also a $(D-(D_M+1)/2)$-Chern number (with $D_M$ odd), and a $(D_S+1)/2$-winding number is equivalently a $(D-D_M/2)$-winding number (with $D_M$ even).

Our theory, that again applies generally in phase space, gives in particular an alternative constructive and systematic demonstration of the bulk-boundary correspondence for symmetry classes A and AIII in arbitrary dimension $D$ without the assumption of a crystalline structure, and provides several explicit expressions of the topological invariants, given by (44), (51) and (52). For instance, the edge states $(D_M = 1)$ of a Chern insulator $(D = 2)$ are indeed given by a $D - (D_M+1)/2 = 1$ (i.e. first) Chern number that implies an integral over a close surface of $D_S = 2$ dimensions, and the surface states $(D_M = 2)$ of a chiral symmetric topological insulator $(D = 3)$ are given by a $(D - D_M/2) = 2$ (second) winding number that implies an integral over a close surface of $D_S = 3$ dimensions, as expected from the bulk-boundary correspondence. Since the dimension $D$ of the system and that $D_M < D$ of the topological mode are independent, our theory also provides for higher order topological phases. As an exotic example, the semi-classical invariant that constrains the existence of $D_M = 3$-dimensional topological gapless modes in a $D = 5$-dimensional system (say in synthetic dimensions) is given by the third Chern number, and the Hamiltonian should be in class A. However, to show that the semi-classical invariants are indeed those in the higher-order case $(D_\perp \geqslant 2)$, it is preferable to first establish a semi-classical expression of the mode index and then show the mode-shell correspondence. This is the meaning of the left downward arrow in the summaries of figure 12, and we call *index semiclassical expansion* the formal rule that allows this development.

## 4.3 Index semi-classical expansion in higher dimension

The equations (44) and (52) establish an equality between winding numbers and Chern numbers associated to the semi-classical Hamiltoninan and the auxiliary operator Hamiltonian in different symmetry classes as

$$
\begin{array}{llll}
\hat{H}' \text{ in class AIII} & \mathcal{W}_{\lceil D_M/2 \rceil}(\hat{H}') & = & \mathcal{C}_{D_S/2}(H_F) & \hat{H} \text{ in class A} \\
\hat{H}' \text{ in class A} & \mathcal{C}_{D_M/2}(\hat{H}') & = & \mathcal{W}_{\lceil D_S/2 \rceil}(H_F) & \hat{H} \text{ in class AIII}
\end{array}
\tag{53}
$$

This result can be understood relatively easily from the equations above for topological semi-metals and strong topological insulators, for which we have $(D_\perp = 0, D_S = D_M - 1)$ and $(D_\perp = 1, D_S = D_M + 1)$ respectively. In those cases, the semi-classical index (52) basically appears as the first order in the semi-classical expansion of respectively the first and second term of (51), where we replace every product of operators by the product of their Wigner-Weyl symbol, except for the commutator that is replaced by the differential of the symbol. However, for higher-order topological insulators, where $D_\perp \geq 2$, such a first order approximation does not work anymore. To deal with these higher-order cases, we step back to the mode index and introduce a new result that we call index semiclassical expansion.

Suppose we have an operator Hamiltonian $\hat{H}(\lambda_1, \ldots, \lambda_{D_M}; \partial_{x_\perp}, x_\perp)$ with $D_M$ mode parameters and one orthogonal dimension $x_\perp$ along which $\hat{H}$ acts as a differential operator. In phase space, this orthogonal direction corresponds to two additional coordinates $x_\perp$ and $k_\perp$ and we can also define the Wigner symbol $H(\lambda_1, \ldots, \lambda_{D_M}, k_\perp, x_\perp)$ of this operator Hamiltonian. The question is then the following: Does the mode index (44) associated to the gapless modes of $\hat{H}(\lambda_1, \ldots, \lambda_{D_M})$, admit a semi-classical expression on its own in terms of $H(\lambda_1, \ldots, \lambda_{D_M}, k_\perp, x_\perp)$, that is, without first expressing the index on the shell ? The answer to that question is yes, and we find that this semi-classical expression takes the form of a higher

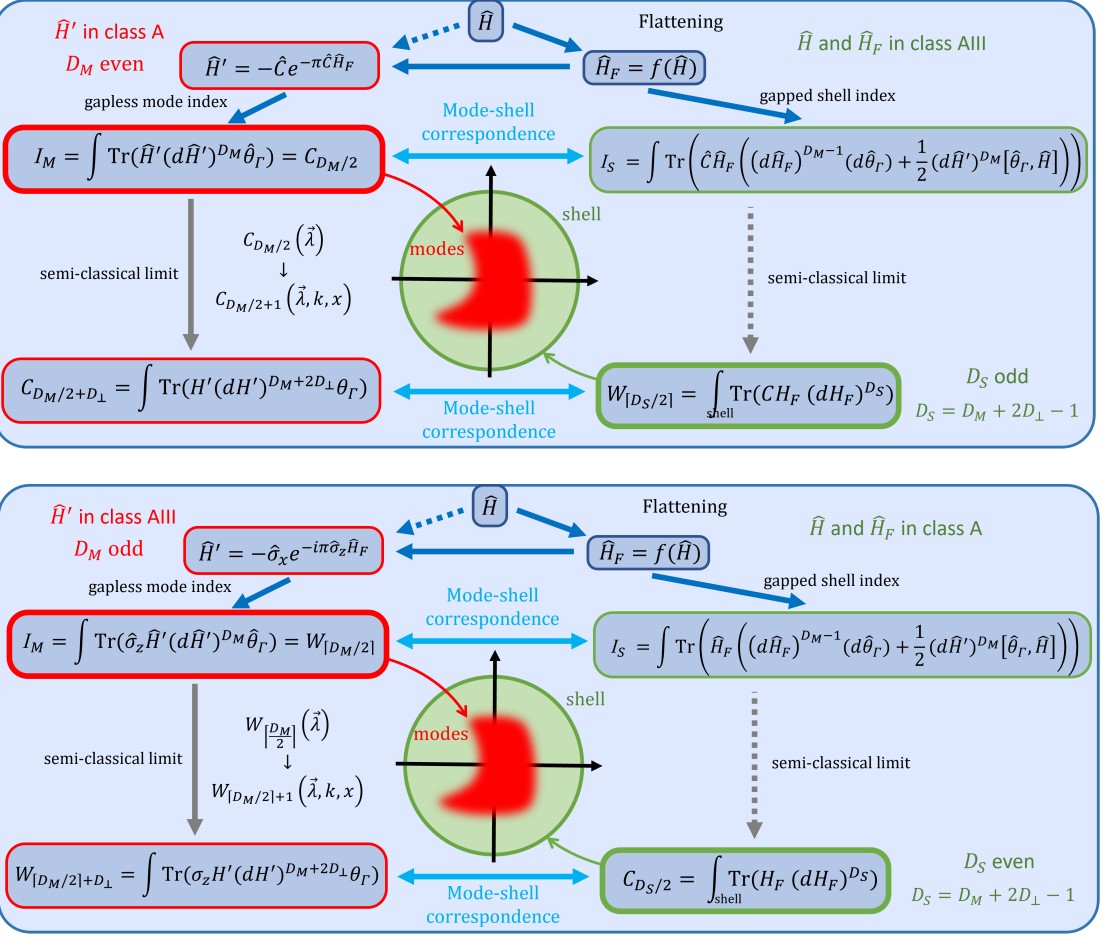

Figure 12: Summary diagram of the mode-shell correspondence in even (top) and odd (bottom) mode dimension $D_M$. The left red part shows the mode indices, while the right green part shows the shell indices. The top part of each diagram gives the expressions of mode and shell indices in terms of the operator Hamiltonian $\hat{H}$ and auxiliary Hamiltonian $\hat{H}'$, and the bottom part gives their semi-classical expressions. The explicit derivation provided in the appendix follows the solid arrows.

Chern/winding number in $D_M + 2$ dimensions $(\lambda_1, \ldots, \lambda_{D_M}, k_\perp, x_\perp)$ as

$$
\mathcal{I}_M = \begin{cases}
\mathcal{W}_{\lceil D_M/2 \rceil}(\hat{H}'_{\lambda_1,\ldots,\lambda_{D_M}}) &=& \mathcal{W}_{\lceil D_M/2 \rceil+1}(H'_{\lambda_1,\ldots,\lambda_{D_M},k_\perp,x_\perp}) & \hat{H} \text{ in class A/}D_M \text{ odd} \\
\mathcal{C}_{D_M/2}(\hat{H}'_{\lambda_1,\ldots,\lambda_{D_M}}) &=& \mathcal{C}_{D_M/2+1}(H'_{\lambda_1,\ldots,\lambda_{D_M},k_\perp,x_\perp}) & \hat{H} \text{ in class AIII/}D_M \text{ even}
\end{cases}
.
$$

(54)

This important result is not obvious because the second index, which involves the Wigner symbol $H'$ of the operator $\hat{H}'$, is not just a first-order approximation of the first index. Below we show a proof of the above statement in the simplest case $\mathcal{C}_0(\hat{H}') = \mathcal{C}_1(H'_{k,x})$. This case will contain all the important ideas which are then used in the general case proved in the appendix B.

**Proof of $\mathcal{C}_0(\hat{H}') = \mathcal{C}_1(H'_{k,x})$:** We start by introducing the identity $i[\partial_x, -ix] = \mathbb{1}$ in the definition of $\mathcal{C}_0(\hat{H}')$. For simplicity, we will remove the cut-off in the expression, as it does not play any role in the

derivation if we keep in mind that the non-zero contributions of the mode index are only confined in the gapless region of phase space. We have

$$\mathcal{C}_0(\hat{H}') \equiv -\frac{1}{2}\operatorname{Tr}\left(\hat{H}' - \hat{C}\right) = -\frac{i}{2}\operatorname{Tr}\left([\partial_x, -ix](\hat{H}' - \hat{C})\right) \tag{55}$$

and then use the "integration by part" identity for commutators $\operatorname{Tr}([A, B]C) = \operatorname{Tr}(\cancel{[A, BC]}) - \operatorname{Tr}(B[A, C]) = -\operatorname{Tr}(B[A, C])$ to obtain

$$\mathcal{C}_0(\hat{H}') = \frac{i}{2}\operatorname{Tr}\left(-ix[\partial_x, \hat{H}' + \hat{C}]\right) = \frac{i}{2}\operatorname{Tr}\left(-ix[\partial_x, \hat{H}']\right). \tag{56}$$

Next, we insert the identity $\mathbb{1} = (\hat{H}')^2$ to get

$$\mathcal{C}_0(\hat{H}') = \frac{i}{2}\operatorname{Tr}\left(-(\hat{H}')^2 ix[\partial_x, \hat{H}']\right) \tag{57}$$

and use the general anti-commutation relation $\{\hat{H}', [A, \hat{H}']\} = [A, (\hat{H}')^2] = [A, \mathbb{1}] = 0$ for $A = \partial_x$, to obtain

$$\begin{aligned}
\mathcal{C}_0(\hat{H}') &= -\frac{i}{4}\operatorname{Tr}\left(\hat{H}'[-ix, \hat{H}'][\partial_x, \hat{H}']\right) \\
&= -\frac{i}{8}\operatorname{Tr}\left(\hat{H}'\left([-ix, \hat{H}'][\partial_x, \hat{H}'] - [\partial_x, \hat{H}'][-ix, \hat{H}']\right)\right).
\end{aligned} \tag{58}$$

When the semi-classical limit holds, we can then apply the Wigner-Weyl transform to move this expression into the symbol picture, where the commutators of operators are replaced by the Poisson brackets of their symbols, and where the trace over the slow degrees of freedom is replaced by an integral over phase space (see appendix B of [1]). In that limit, we obtain the semi-classical expression

$$\mathcal{C}_0(\hat{H}') = \int dx\,dk\,\frac{-1}{16i\pi}\operatorname{Tr}\left(H'\left(\partial_k H'\partial_x H' - \partial_x H'\partial_k H'\right)\right) \equiv \mathcal{C}_1(H'_{k,x}) \tag{59}$$

which is the desired equality and thus concludes the proof.

The index semiclassical expansion (54) can be used to tackle higher order topological insulators. For that we can apply (54) for each of the $D_\perp$ perpendicular dimensions, increasing the dimension of the mode index by 2 at each iteration. In the end, we obtain that the full semi-classical limit of the mode index is a higher winding/Chern number of dimension $D_M + 2\,D_\perp$. Consistently, this semi-classical mode index also verifies the mode-shell correspondence (51), as depicted by the bottom horizontal double arrow in figure 12. Indeed, since we have reached the full semi-classical limit (i.e. there is no orthogonal direction $x_\perp$ left along which the Hamiltonian acts as a differential operator), the symbol of the cut-off $\theta_\Gamma$ is just a scalar function with vanishing commutators, so the second term of (51) vanishes and only the first term remains. Then, if the transition of the cut-off from 1 to 0 (i.e. the shell) is made sharp in phase space, $d\theta_\Gamma$ becomes a Dirac distribution on the shell so that the integral over the whole phase space is replaced by an integral just on the shell, which gives the semi-classical shell index (52)[6].

Now that we have addressed the general theory of the mode and shell indices in arbitrary dimension, we illustrate how the mode-shell correspondence and its phenomenology are verified in particular examples for $D_M = 2$ and $D_M = 3$.

---

[6]This result also applies for the $D_M = 0$ case that is studied in part I. In particular both appendix 4.1 and 4.3 imply the result proved in appendix F of the part I [1].

# 5   $\mathrm{D_M} = 2$ and $\mathrm{D_M} = 3$ : Models with $2D$-Dirac or $3D$-Weyl cones

In this section 5, we revisit minimal models and lattice models exhibiting Dirac and Weyl cones through the prism of the mode-shell correspondence. It is well accepted that $2D$-Dirac cones and $3D$-Weyl nodes are considered as topological, a characterization respectively given by a winding number and a Chern number. Here, we identify those properties as being given by the shell invariant in our formalism, and provide the expression of their associated and less common mode invariant. The mode-shell correspondence then justifies the use of those shell invariants to count the "topological charge" of the modes and also provides a more general expression of the invariant that does not require a semiclassical limit, or in particular the translation invariance, as in crystals. The interest of this formalism is also that any $2D$-Dirac cone or any $3D$-Weyl node is captured by the same mode invariant $\mathcal{I}_\mathrm{M}$ (respectively for $\mathrm{D_M} = 2$ and $\mathrm{D_M} = 3$) irrespective of whether they emerge as bulk or edge excitations. This contrasts the usual topological description where, e.g., $2D$ Dirac cones are associated with a Berry winding number when they appear as bulk excitations in a $2D$ semimetal, but are associated with a higher-dimensional bulk topological invariant, through the bulk-edge correspondence, when they emerge as surface modes of a $3D$ topological insulator. Those distinctions are captured by the shell invariant that depends in particular of the topology of the shell in phase space, as depicted in figure 13. Such examples are discussed below, for both $2D$ Dirac and $3D$ Weyl nodes. In appendix D we also discuss additionally the case of a continuous model hosting Dirac cones at its interface and discuss its mode-shell topology.

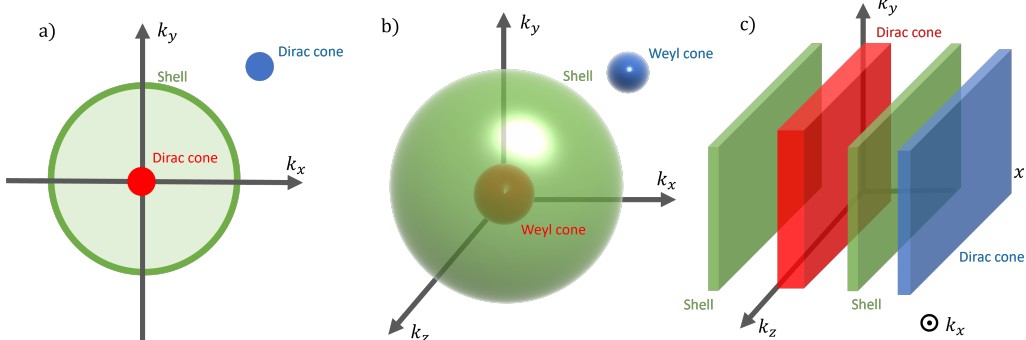

Figure 13: Sketches of different shells (green) that surrounds a) a Dirac point ($\mathrm{D_M} = 2$) confined in wavenumber space ($D = 2$) where the shell forms a circle, b) a Weyl node ($\mathrm{D_M} = 3$) confined in wavenumber space ($D = 3$) where the shell forms a sphere, and c) a Dirac point ($\mathrm{D_M} = 2$) confined at the surface of a 3D insulator, where the shell consists of two $3D$ Brillouin zones located in the two bulks "surrounding" the surfaces of the material in phase space.

## 5.1   Single Dirac/Weyl cones

$2D$ **Dirac cone,** $\mathrm{D_M} = 2$.   The prototypical Hamiltonian of a $2D$-Dirac cone is given by

$$\hat{H}_\mathrm{Dirac} = \begin{pmatrix} 0 & k_x - ik_y \\ k_x + ik_y & 0 \end{pmatrix} \tag{60}$$

whose spectrum $E_\pm = \pm\sqrt{k_x^2 + k_y^2}$ indeed corresponds to a single a Dirac cone centered at $(k_x, k_y) = (0, 0)$. We can use the wavenumbers as parameters $\lambda_1 = k_x$ and $\lambda_2 = k_y$. Since there is no operators

in other dimensions (i.e. $D_\perp = 0$), the model is already in a symbol formulation, so that we could equivalently use the notation $H_{\text{Dirac}}$. Thus, since the Dirac Hamiltonian (60) is chiral symmetric and depends on two parameters, its gapless modes (here the Dirac point) are encoded into the mode index (44) that reads

$$\mathcal{I}_{\text{M}} = \mathcal{C}_1(H') = -\frac{1}{8i\pi} \int \text{Tr}\left(H'(dH')^2 \hat{\theta}_\Gamma\right) \tag{61}$$

with the cut-off $\theta_\Gamma = (1 + \exp(k_x^2 + k_y^2 - \Gamma^2))^{-1}$ and with $H' = -Ce^{-\pi C H_{\text{F, Dirac}}}$. This is a first Chern number for the auxiliary Hamiltonian $H'$. The integral is taken over $k_x$ and $k_y$ and extends over $\mathbb{R}^2$. This is may seem problematic to define a topological index, since $\mathbb{R}^2$ is not a compact. However, the compactness is guaranteed here by the cut-off $\hat{\theta}_\Gamma$ that restrains the plan into a disk of radius $\hat{\theta}_\Gamma$ at the boundary of which $H'$ is single-valued ($H' = C$ if $\hat{H}$ is chiral, and $H' = \sigma_x$ otherwise). Note that this expression only depends on the existence of Dirac points in the dispersion relation, and not on the specific physical situation where those Dirac points are encountered. The only difference comes from the definition of the cut-off $\hat{\theta}_\Gamma$, which will be important when we will move on to the shell index as it changes the topology of the shell.

We shall not evaluate this index directly, but instead use the mode-shell correspondence (50) that tells us that this index also reads as a shell index whose semiclassical expression is given by the formula (52). It corresponds here to the winding number $\mathcal{W}_1$ on the circle $k_x^2 + k_y^2 = \Gamma^2$ that surrounds the Dirac point in phase space, so that $D_S = 1$ (see figure 13.a). This shell index corresponds to the usual topological description of Dirac points. To evaluate this shell index, one notices that the Dirac Hamiltonian (60) is formally identical to the symbol of the Jackiw-Rebbi Hamiltonian of section 3.3 of [1] for which the shell invariant was computed, except that the part of phase space where the mode spreads is now $(k_x, k_y)$ instead of $(x, k)$[7]. The winding number of those two models are therefore identical, so that $\mathcal{W}_1(H_{\text{Dirac}}) = 1$. This means that the simple model (60) can be considered as topological by our shell index, and therefore by our mode index as well. Note that if one flips the sign of one direction (say $k_x \to -k_x$), the shell index of the new Dirac Hamiltonian has opposite sign. In the literature, the first Dirac Hamiltonian is said to have a "positive chirality" while the second one is said to have a "negative chirality" [10]. Such a difference of chirality is captured by the sign of $\mathcal{I}_{\text{M}}$. Therefore, with a slight abuse of notation, we can say that the $2D$ mode index $\mathcal{I}_{\text{M}}$ "counts" the number of $2D$ Dirac cones with chirality.

**3D Weyl cone, $D_M = 3$.** The simplest Hamiltonian that exhibits a $3D$ Weyl cone is the following Weyl Hamiltonian

$$\hat{H}_{\text{Weyl}} = \begin{pmatrix} k_z & k_x - ik_x \\ k_x + ik_y & -k_z \end{pmatrix} \tag{62}$$

whose spectrum $E_\pm = \pm\sqrt{k_x^2 + k_y^2 + k_z^2}$ indeed consists in a single Weyl cone centered at $(k_x, k_y, k_z) = (0, 0, 0)$. The three wavenumbers can be taken as parameters $\lambda_1 = k_x$, $\lambda_2 = k_y$ and $\lambda_3 = k_z$, and there is no operator left in other directions, so that the model is already in a symbol formulation (i.e. $D_\perp = 0$), and we could similarly use the notation $H_{\text{Weyl}}$. Thus, since the Weyl Hamiltonian (62) is not chiral symmetric and depends on three parameters, its gapless modes (here the Weyl point) are encoded into

---

[7]A difference is however that $x$ and $k$ are only commuting variables in the semi-classical picture while $k_x$ and $k_y$ are good quantum numbers which do not necessitate a semi-classical approximation. If the shell invariants are the same, the gapless properties of the operator and the mode indices $\mathcal{I}_{\text{M}}$ are different. The Jackiw-Rebbi model has a single zero mode while the Dirac model has a Dirac cone.

the mode index (44) that reads

$$\mathcal{I}_{\mathrm{M}} = \mathcal{W}_2(H') = -\frac{1}{48\pi^2} \int \mathrm{Tr}\big(\sigma_z H'(dH')^3 \theta_\Gamma\big) \tag{63}$$

where we can choose $\theta_\Gamma = \big(1 + \exp\big(k_x^2 + k_y^2 + k_z^2 - \Gamma^2\big)\big)^{-1}$ for the cut-off. This invariant is a "second" winding number for the auxiliary Hamiltonian, where the integral is taken over the three directions of $\mathbb{R}^3$. Similarly to the Dirac case, this domain becomes compact thanks to the cut-off $\theta_\Gamma$ that limits it to the ball of radius $\Gamma$.

Again, we shall not evaluate this index directly, but instead use the mode-shell correspondence (50) that tells us that this mode index also reads as a shell index whose semiclassical expression, given by the formula (52), corresponds here to the first Chern number $\mathcal{C}_1$ over the shell given by the 2-sphere $k_x^2 + k_y^2 + k_z^2 = \Gamma^2$ that surrounds the Weyl point in phase space, so that $\mathrm{D_S} = 2$ (see figure 13.b). This shell index corresponds to the usual topological description of Weyl nodes. The computation of this Chern number is formally identical to that of modified Jackiw-Rebbi model of Section 3.3 except that the part of phase space is now $(k_x, k_y, k_z)$ instead of $(k_y, x, k_x)$. The Chern numbers of those two models are therefore identical, so that $\mathcal{C}_1(H_{\mathrm{Weyl}}) = 1$. This non-zero value means that this model is considered as topological by our shell index and therefore by our mode index as well. Note that if one flips the sign of one direction (say $k_x \to -k_x$), the shell index of the new Weyl Hamiltonian has opposite sign. In the literature, the first Weyl Hamiltonian is said to have a "positive chirality" while the second one is said to have a "negative chirality" [10]. Such a difference of chirality is captured by the sign of $\mathcal{I}_{\mathrm{M}}$. Therefore, with a slight abuse of notation, we can say that the $3D$ mode index $\mathcal{I}_{\mathrm{M}}$ "counts" the number of $3D$ Weyl nodes with chirality.

## 5.2  $2D$ Dirac and $3D$ Weyl cones in lattice models

Dirac and Weyl cones may emerge either as bulk excitations of a semimetal, or as boundary modes of an insulator. We now want to illustrate the mode-shell correspondence in both cases with simple lattice models that we generate with the additive tensor product construction adapted to this case. The reader who wants to remain focus on the topological invariants for Dirac/Weyl nodes can skip the next paragraph that explains this construction.

### Additive tensor product construction ⊞.

Similarly to part I of this work and to the previous section 3.5, we will use an additive tensor product construction to build simple-to-analyze models. In section 3.5, we rediscussed this construction to generate higher-dimensional (non-chiral symmetric) Hamiltonians with spectral flow. Here we introduce another additive tensor product construction to generate a chiral symmetric Hamiltonian $\hat{H}_{\boxplus}$ from two non-chiral symmetric Hamiltonians $\hat{H}_A$ and $\hat{H}_B$ as follows

$$\hat{H}_{\boxplus} = \hat{H}_A \boxplus \hat{H}_B \equiv \hat{H}_A \otimes \mathbb{1} \otimes \sigma_x + \mathbb{1} \otimes \hat{H}_B \otimes \sigma_y. \tag{64}$$

where $\sigma_x$ and $\sigma_y$ are the usual Pauli matrices and where $\mathbb{1} \otimes \mathbb{1} \otimes \sigma_z$ is a chiral symmetry of $\hat{H}_{\boxplus}$.

In particular, if we take $\hat{H}_B = k_y$, one can generate a model with a $2D$-Dirac cone from a Hamiltonian $\hat{H}_A = \hat{H}(k_x)$ that exhibits a spectral flow like any of those treated in section 2. This construction gives

$$\hat{H}_{\boxplus}(k_x, k_y) = \hat{H}(k_x) \otimes \sigma_x + k_y \mathbb{1} \otimes \sigma_y. \tag{65}$$

Such a model will host a Dirac cones for each gapless mode $|\psi(k_x)\rangle$ of $\hat{H}(k_x)$ with spectral flow. Indeed, because of the additive construction, zero modes of $\hat{H}_\boxplus(k_x, k_y)$ must be of the form $|\psi(k_x)\rangle \otimes |\psi_\pm\rangle$ where $|\psi(k_x)\rangle$ is a zero mode of $\hat{H}(k_x)$. If we project on this subspace, the Hamiltonian then reduces to

$$sopH_\boxplus(k_x, k_y) = E(k_x) \otimes \sigma_x + k_y \mathbb{1} \otimes \sigma_y. \tag{66}$$

which is just a Dirac cones model if we linearize $E(k_x) = k_x \partial_{k_x} E$ with chirality depending on the sign of the group velocity $\partial_{k_x} E$.

When dealing with lattice models, this construction needs to be slightly modified, as in section 3.5, to become

$$\hat{H}_{\tilde{\boxplus}}(k_x, k_y) = \hat{H}(k_x)\,\tilde{\boxplus}\,\sin(k_y) \equiv (\mathbb{1} + (\hat{H} - \mathbb{1})(1 + \cos(k_x))/2) \otimes \sigma_x + \sin(k_y)\mathbb{1} \otimes \sigma_y \tag{67}$$

which avoids unnecessary doubling of the number of Dirac/Weyl cones at $k = \pi/2$. Indeed, with this modification, the system is gapped at $k = \pi$ because $\hat{H}_{\tilde{\boxplus}}(k = \pi) = \mathbb{1} \otimes \sigma_x$.

Similarly, the additive construction of section 3.5 can be used to generate models $\hat{H}_\boxplus(k_x, k_y, k_z)$ hosting Weyl cones from model $\hat{H}(k_x, k_y)$ hosting Dirac cones

$$\hat{H}_\boxplus(k_x, k_y, k_z) = \hat{H}(k_x, k_y)\,\boxplus\,k_z = \hat{H}(k_x, k_y) + k_z.\hat{C}. \tag{68}$$

### Bulk Dirac/Weyl modes in topological semi-metals

A typical situation where Dirac/Weyl cones appear is as bulk excitations of semimetals. In this case, they appear in pairs of opposite chirality. This is known as the Nielsen-Ninomiya theorem [81,82]. This theorem can be easily understood with the mode-shell correspondence. In systems where the set of parameter $\lambda_i$ is bounded (as in the case of a Brillouin zone), the total number of Dirac/Weyl cones is given by the mode index (44) with $\theta = \mathbb{1}$. However, the corresponding shell index is zero because both $d\theta$ and $[\theta, H]$ vanish in this case. Therefore, the total numbers of Dirac/Weyl cones of positive and negative chirality must cancel each other out.

**Example 7: Semimetals with a single pair of Dirac/Weyl points.** There are many models in the literature that host one pair of Dirac/Weyl cones, such as tight-binding systems for graphene [83] in $2D$. Here we use the modified additive construction $\tilde{\boxplus}$ discussed above as a systematic way to generate lattice models with only one pair of $2D$ Dirac or $3D$ Weyl cones. We treat those two cases simultaneously. Following this construction, we have

$$\begin{aligned}\hat{H}_{\text{2-Dirac}}^{\text{lattice}}(k_x, k_y) &= \sin(k_x)\tilde{\boxplus}\sin(k_y) \\ &= (\mathbb{1} + (\sin(k_x) - \mathbb{1})(1 + \cos(k_y))/2)\,\sigma_x + \sin(k_y)\sigma_y\end{aligned} \tag{69}$$

and

$$\begin{aligned}\hat{H}_{\text{2-Weyl}}^{\text{lattice}}(k_x, k_y, k_z) &= \sin(k_x)\tilde{\boxplus}\sin(k_y)\tilde{\boxplus}\sin(k_z) \\ &= \sin(k_z)\sigma_z + \sigma_x + \left(\hat{H}_{\text{2-Dirac}}^{\text{lattice}}(k_x, k_y) - \sigma_x\right)(1 + \cos(k_z))/2\end{aligned} \tag{70}$$

whose spectrum has 2 Dirac/Weyl cones (see Figure 14). Owing to the structure of the modified additive construction, the Dirac/Weyl cones must be centered at $k_y = 0$ (respectively $k_y, k_z = 0$ for the Weyl

cones). For each model, one cone is then centered at $k_x = 0$ and the other at $k_x = \pi$. By linearizing those two models, one can then check that the Dirac/Weyl cones at $k_x = 0$ is associated with a positive mode index $\mathcal{I}_M = 1$ while the one at $k_x = \pi$ is associated with a negative mode index $\mathcal{I}_M = -1$. Similarly the shell index can be computed on a shell which is a small circle/sphere around the Dirac/Weyl point. In this limit the computation of the shell index reduces to the one of the previous section.

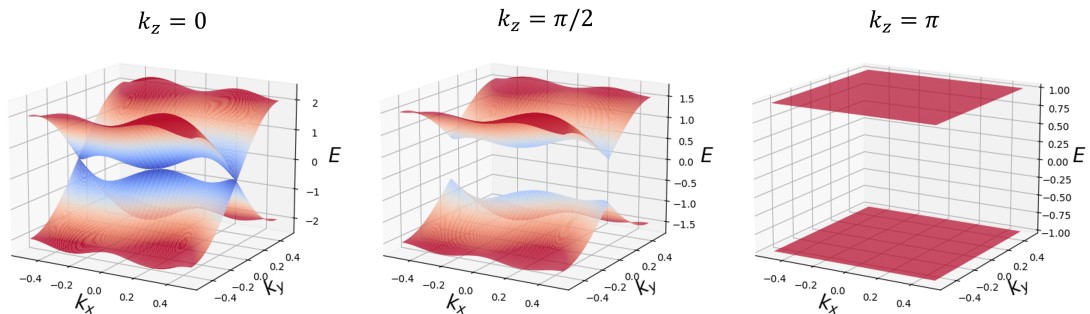

Figure 14: Spectrum of the model (70) hosting 3$D$-Weyl cones for different values of $k_z$. The figure for $k_z = 0$ also coincides with the spectrum of the model (69) hosting 2$D$-Dirac cones.

Those examples constitute other instances where the topological modes are separated from each other in $k$-space, similarly to zero-modes ($D_M = 0$) and spectral flow modes ($D_M = 1$) discussed respectively in sections 3.2 and 3.4 of Part I [1] and in section 3.1 of the present paper. As such, the cones are topologically protected as long as the Hamiltonian is short range in wavenumber. So the Dirac cones are protected against perturbations of the Hamiltonian which are slowly varying in position but not those that vary quickly in space as impurities in the crystal structure. To enhance the protection of the Dirac/Weyl cones to such local perturbations, it would thus be preferable to separate those cones in position space instead. This condition is realized when Weyl/Dirac cones appear as surface states of a topological insulator, as we discuss below.

## Boundary Dirac/Weyl cones of topological insulators

One can similarly construct lattice models of topological insulators in class A and AIII that host 2$D$ Dirac or 3$D$ Weyl modes at their boundaries, by using the modified additive tensor product construction.

**Example 8: 3$D$ chiral topological insulator (class AIII).** Let us start with chiral symmetric topological insulators (class AIII), that host 2$D$-Dirac cones at their surfaces. A lattice model for such an insulator is provided by the modified additive construction on a QWZ model (see section 3.2) which itself can be constructed from an additive construction with the SSH model.

$$
\begin{aligned}
\hat{H}_{\text{3D-TI}}(k_y, k_z) &= \hat{H}_{\text{QWZ}}(k_y) \,\tilde{\boxplus}\, \sin(k_z) \\
&= (\mathbb{1} + (\hat{H}_{\text{QWZ}}(k_y) - \mathbb{1})(1 + \cos(k_z))/2) \otimes \sigma_x + \sin(k_z)\mathbb{1} \otimes \sigma_y \\
&= \hat{H}_{\text{SSH}} \,\tilde{\boxplus}\, \sin(k_y) \,\tilde{\boxplus}\, \sin(k_z).
\end{aligned}
\tag{71}
$$

This operator is indeed chiral symmetric with the chiral operator $\sigma_z$. By construction, gapless modes can only appear at $k_z = 0$ and $k_y = 0$. In the vicinity of those points, the model reduces to

$$
\hat{H}_{\text{3D-TI}}(k_y, k_z) = \hat{H}_{\text{SSH}} \boxplus \begin{pmatrix} 0 & k_y - ik_z \\ k_y + ik_z & 0 \end{pmatrix}.
\tag{72}
$$

Thanks to the structure of the additive construction, we know that the gapless modes of $\hat{H}_{3D\text{-TI}}$ must be of the form $|\psi_{\text{Dirac}}\rangle = |\psi_0\rangle \otimes \begin{pmatrix} |\psi\rangle_+ \\ |\psi\rangle_- \end{pmatrix}$ where $|\psi_0\rangle$ is a zero-mode of the SSH model.

In order to evaluate the number of $2D$ Dirac cones (with chirality) on a given surface (say the left one) of this $3D$ model, one thus takes $|\psi_0\rangle$ as the zero-mode on the left boundary of the SSH chain. As defined in Part I, this mode has a positive chirality. The projection of $\hat{H}_{3D\text{-TI}}$ on the two degrees of freedom $|\psi\rangle_\pm$ then just reduces to the Dirac model of section 5.1 where we showed that $\mathcal{I}_M = 1$, which gives the number of Dirac modes (with chirality) on that surface.

Let us now comment on the shell index for $2D$ surface Dirac cones, that differs from that of bulk Dirac cones in semimetals, although their mode index has the same expression. Indeed, we now have $D = 3$ and $D_M = 2$, which yields a shell of dimension $D_S = 3$. Consistently, the Hamiltonian operator (71) is not in its symbol form here, as one direction is left without parameter, i.e. $D_\perp = 1$. The shell consists here of a $3D$ Brillouin zone (see figure 13 c)), and the corresponding semi-classical index, given by (51), is the second winding number

$$\mathcal{I}_S = \mathcal{W}_2(H_F) = -\frac{1}{12(2\pi)^2} \int_{k_x,k_y,k_z\in[0,2\pi]} dk_x dk_y dk_z \operatorname{Tr}\left(\hat{C}H_F(dH_F)^3\right) . \tag{73}$$

The mode-shell correspondence $\mathcal{I}_M = \mathcal{I}_S$ therefore reduces to the bulk-edge correspondence in that case, where the "bulk" topological invariant is our shell index (73).

**Example 9: $4D$ topological insulator in class A.** We now construct a lattice model for a 4D topological insulator with a single $3D$-Weyl cone confined at each $3D$ boundary. We iterate again the modified additive construction on the model of example 8, so that the evaluation of $\mathcal{I}_M$, i.e. of $3D$ boundary Weyl nodes, simplifies to that of the zero-modes of the SSH model. Following this construction, the operator Hamiltonian reads

$$\begin{aligned} \hat{H}_{4D\text{-TI}}(k_y, k_z, k_u) &= \hat{H}_{3D\text{-TI}}(k_y, k_z) \,\tilde{\boxplus}\, \sin(k_u) \\ &= (\sigma_x + (\hat{H}_{3D\text{-TI}}(k_y, k_z) - \sigma_x)(1 + \cos(k_u))/2) + \sin(k_u)\sigma_z \\ &= \hat{H}_{\text{SSH}} \,\tilde{\boxplus}\, \sin(k_y) \,\tilde{\boxplus}\, \sin(k_z) \,\tilde{\boxplus}\, \sin(k_u) \end{aligned} \tag{74}$$

and the gapless modes must therefore occur near $k_y, k_z, k_y \sim 0$, where the Hamiltonian can be linearized as

$$\hat{H}_{4D\text{-TI}}(k_y, k_z, k_u) = \hat{H}_{\text{SSH}} \boxplus \begin{pmatrix} k_u & k_y - ik_z \\ k_y + ik_z & -k_y \end{pmatrix} . \tag{75}$$

Similarly to the previous example, the gapless modes must be of the form $|\psi_0\rangle \otimes \begin{pmatrix} |\psi\rangle_+ \\ |\psi\rangle_- \end{pmatrix}$ for which the Hamiltonian reduces to that of a single the Weyl node as described in section 5.1. It follows that the number of Weyl cones (with chirality) on the left boundary is given by $\mathcal{I}_M = 1$.

Again, the shell index for $3D$ Weyl boundary modes differ from that of $3D$ bulk Weyl modes in semimetals, although they both have a similar mode index. The system has now $D = 4$ dimensions, and the gapless Weyl modes correspond to $D_M = 3$, so that the shell is of dimension $D_S = 4$. It consists of a $4D$ Brillouin zone (see figure 13 c)), and the corresponding semi-classical index, given by (51), is the second Chern number

$$\mathcal{I}_S = \mathcal{C}_2(H_F) = -\frac{1}{256\pi^2} \int_{k_x,k_y,k_z,k_t\in[0,2\pi]} dk_x dk_y dk_z dk_u \operatorname{Tr}\left(H_F(dH_F)^4\right) . \tag{76}$$

The mode-shell correspondence $\mathcal{I}_M = \mathcal{I}_S$ therefore reduces to the bulk-edge correspondence as a particular case, where the "bulk" topological invariant is our shell index (76).

In the case of $3D$ topological insulators, isolated Dirac cones have been experimentally observed at the boundary of 3D topological insulators [84,85]. Observing $3D$-Weyl cones confined at the edge of $4D$ material is however more difficult due to the fact that we live only in a 3-dimensional world. However, the idea can still be explored as one can replace a physical dimension by either a pumping parameter [86,87] or using internal degrees of freedom such as spin or polarisation as a synthetic dimension [88]. Even if it is not a space dimension, synthetic dimensions can play the same role and as long as the coupling in those dimensions remains short range and the edge modes remain uncoupled in a way which may be stronger than if they where separated in wavenumber.

## 6   Conclusion

In this work, a mode-shell correspondence is developed as a unifying framework for the understanding of the topological nature of certain gapless modes in arbitrary dimension. As such, explicit expressions of $\mathbb{Z}-$valued invariants counting the gapless modes are provided both at the operator level and at the semiclassical level. We showed that this correspondence encompasses the well-known bulk-edge correspondence as a specific instance while extending, in a unifying way, to more general scenarios in phase space. These include models with modes localized in wavenumber rather than position like topological semi-metals, higher-dimensional systems such as weak insulators exhibiting macroscopic spectral flow, and higher-order insulating phases with modes topologically confined in multiple directions. We also examined the topological aspects of $2D$-Dirac and $3D$-Weyl nodes, providing invariants that characterize their topology and that verify the mode-shell correspondence. Through this, we highlighted the similarity and difference in phenomenology between Dirac/Weyl cones confined in wavenumber as in topological semi-metals and those confined in position, like in $3D$ or $4D$ topological insulators. To facilitate the analysis, we used additive tensor product constructions, underscoring their utility in constructing and analyzing topological systems. Together, these results demonstrate the versatility and generality of the mode-shell correspondence as a framework for understanding the topological properties of gapless modes across a wide range of physical contexts. This common description of such a wide phenomenology within the same formalism is made possible by the distinction, in our theory, of the space dimension $D$, the mode dimension $D_M$, and the shell dimension $D_S$ together with its topology.

This theory was developed to account for A and AIII symmetry classes, thus providing explicit expressions of $\mathbb{Z}-$valued indices. Possible extensions of this work for other symmetry classes with $\mathbb{Z}_2$ invariants are left for future investigations.

**Funding information:**   LJ was funded by a PhD grant allocation Contrat doctoral Normalien. This research was supported by the Swedish Research Council (VR) through Grant No. 2020-00214.

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

# A    Proof of the general mode-shell correspondence

In this appendix, the goal is to prove the mode-shell correspondence of the general mode index defined in section 4.1 and, in particular, that it is equal to the shell index $\mathcal{I}_\mathrm{S}$ described in the equation (51). We separate the proof of the chiral and non-chiral case. We start by the chiral case.

## A.1    $\mathrm{D_M}$ even, $\hat{H}$ chiral

We start with the second equation of (44) which reads

$$\mathcal{I}_\mathrm{M} \equiv \mathcal{C}_{2D}(\hat{H}') = b_{2D} \int \mathrm{Tr}\Big( \hat{H}'(d\hat{H}')^{2D}\hat{\theta}_\Gamma \Big) \tag{77}$$

with $2D = \mathrm{D_M}$.

If we introduce the path $\hat{H}'_t = -\hat{C}e^{-t\hat{C}\hat{H}_F}$, we can differentiate such expression and obtain

$$\mathcal{C}_{2D}(\hat{H}')/b_D = \int_0^\pi dt \int \mathrm{Tr}\Big( \partial_t\hat{H}'(d\hat{H}')^{2D}\hat{\theta}_\Gamma \Big) + \sum_{i=0}^{D-1} \mathrm{Tr}\Big( \hat{H}'(d\hat{H}')^i\partial_t(d\hat{H}')(d\hat{H}')^{2D-i-1}\hat{\theta}_\Gamma \Big). \tag{78}$$

If we then integrate by part (in the derivative $d$) the second term, we obtain

$$
\begin{aligned}
\mathcal{C}_{2D}(\hat{H}')/b_D &= \int_0^\pi dt \int \mathrm{Tr}\Big( \partial_t\hat{H}'(d\hat{H}')^{2D}\hat{\theta}_\Gamma \Big) + \sum_{i=0}^{D-1}(-1)^{i+1}\mathrm{Tr}\Big( (d\hat{H}')^{i+1}\partial_t\hat{H}'(d\hat{H}')^{2D-i-1}\hat{\theta}_\Gamma \Big) \\
&\quad + \sum_{i=0}^{D-1}(-1)^i \mathrm{Tr}\Big( \hat{H}'(d\hat{H}')^i\partial_t\hat{H}'(d\hat{H}')^{2D-i-1}(d\hat{\theta}_\Gamma) \Big) \\
&= \int_0^\pi dt \int \sum_{i=0}^{D}(-1)^i \mathrm{Tr}\Big( (d\hat{H}')^i\partial_t\hat{H}'(d\hat{H}')^{2D-i}\hat{\theta}_\Gamma \Big)) \\
&\quad + \sum_{i=0}^{D-1}(-1)^i \mathrm{Tr}\Big( \hat{H}'(d\hat{H}')^i\partial_t\hat{H}'(d\hat{H}')^{2D-i-1}(d\hat{\theta}_\Gamma) \Big)
\end{aligned}
\tag{79}
$$

Because $(\hat{H}')^2 = \mathbb{1}$ we can insert it in the first term between $(d\hat{H}')^i$ and $\partial_t\hat{H}'$. Using the anti-commutation relations $d(\hat{H}')^2 = 0 = \{\hat{H}', d\hat{H}'\}$ as well as $\partial_t(\hat{H}')^2 = 0 = \{\hat{H}', \partial_t\hat{H}'\}$, we can show

that

$$\mathcal{C}_{2D}(\hat{H}')/b_D = \int_0^\pi dt \int \sum_{i=0}^D \frac{1}{2} \mathrm{Tr}\Big( (d\hat{H}')^i \hat{H}' \partial_t \hat{H}' (d\hat{H}')^{2D-i} [\hat{\theta}_\Gamma, \hat{H}'] \Big))$$

$$+ \sum_{i=0}^{D-1} \mathrm{Tr}\Big( (d\hat{H}')^i \hat{H}' \partial_t \hat{H}' (d\hat{H}')^{2D-i-1} (d\hat{\theta}_\Gamma) \Big) \tag{80}$$

Because in this formulation we have terms with either commutator of the cut-off or differential of it, the index is localised on the shell and we can use that $\hat{H}_F^2 = \mathbb{1}$ in this region and so $\hat{H}'_t = \sin(t)\hat{H}_F - \hat{C}\cos t$. Using that therefore $\partial_t \hat{H}' = \cos(t)\hat{H}_F + \hat{C}\sin t$, $d\hat{H}' = \sin(t)d\hat{H}_F$ and $[\hat{\theta}_\Gamma, \hat{H}'] = \sin(t)[\hat{\theta}_\Gamma, \hat{H}_F]$, this lead to

$$\mathcal{C}_{2D}(\hat{H}')/b_D = -\Big( \int_0^\pi dt \sin(t)^{2D+1} \Big) \int \frac{2D+1}{2} \mathrm{Tr}\Big( \hat{C}\hat{H}_F (d\hat{H}_F)^{2D} [\hat{\theta}_\Gamma, \hat{H}'] \Big)$$

$$- \Big( \int_0^\pi dt \sin(t)^{2D-1} \Big) \int 2D \, \mathrm{Tr}\Big( \hat{C}\hat{H}_F (d\hat{H}_F)^{2D-1} (d\hat{\theta}_\Gamma) \Big) \tag{81}$$

If we now use the that $\int_0^\pi dt \sin(t)^{2D-1} = \frac{2^{2D-1}(D-1)!^2}{(2D-1)!}$ and $b_{2D} = \frac{1}{2^{2D+1}D!(-2i\pi)^D}$, we have that $b_{2D} \int_0^\pi dt \sin(t)^{2D+1}(2D+1)/(2D) = a_{2D-1}/2$ and $b_D \int_0^\pi dt \sin(t)^{2D-1} 2D = a_{2D-1}$ so we obtain

$$\mathcal{I}_{\mathrm{M}} \equiv \mathcal{C}_{2D} = -a_{2D-1} \int \Big( \mathrm{Tr}\Big( \hat{C}\hat{H}_F (d\hat{H}_F)^{2D-1} (d\hat{\theta}_\Gamma) \Big) + \frac{1}{2} \mathrm{Tr}\Big( \hat{C}\hat{H}_F (d\hat{H}_F)^{2D} [\hat{\theta}_\Gamma, \hat{H}'] \Big) \Big) \equiv \mathcal{I}_{\mathrm{S}} \tag{82}$$

which is the wanted mode-shell correspondence (51).

## A.2  $\mathrm{D_M}$ odd, $\hat{H}$ non chiral

We start with the first equation of (44) which reads

$$\mathcal{I}_{\mathrm{M}} \equiv \mathcal{W}_{2D-1}(\hat{H}') = a_{2D-1} \int \mathrm{Tr}\Big( \hat{\sigma}_z \hat{H}' (d\hat{H}')^{2D-1} \hat{\theta}_\Gamma \Big) \tag{83}$$

with $\mathrm{D_M} = 2D - 1$

If we introduce the path $\hat{H}'_t = -\sigma_x e^{-it\sigma_z \hat{H}_F}$, we can differentiate such expression and obtain

$$\mathcal{W}_{2D-1}(\hat{H}')/a_{2D-1} = \int_0^\pi dt \int \mathrm{Tr}\Big( \sigma_z \partial_t \hat{H}' (d\hat{H}')^{2D-1} \hat{\theta}_\Gamma \Big)$$

$$+ \sum_{j=0}^{2D-2} \mathrm{Tr}\Big( \sigma_z \hat{H}' (d\hat{H}')^j \partial_t (d\hat{H}') (d\hat{H}')^{2D-j-2} \hat{\theta}_\Gamma \Big) \tag{84}$$

if we then integrate by part (in the derivative $d$) the second term, we obtain

$$
\begin{aligned}
\mathcal{W}_{2D-1}(\hat{H}')/a_{2D-1} = & \int_0^\pi dt \int \mathrm{Tr}\Big(\sigma_z \partial_t \hat{H}'(d\hat{H}')^{2D-1}\hat{\theta}_\Gamma\Big) \\
& + \sum_{j=0}^{2D-2} (-1)^{j+1}\, \mathrm{Tr}\Big(\sigma_z (d\hat{H}')^{j+1}\partial_t \hat{H}'(d\hat{H}')^{2D-j-2}\hat{\theta}_\Gamma\Big) \\
& - \sum_{j=0}^{2D-2} (-1)^j\, \mathrm{Tr}\Big(\sigma_z \hat{H}'(d\hat{H}')^j\partial_t \hat{H}'(d\hat{H}')^{2D-j-2}(d\hat{\theta}_\Gamma)\Big) \\
= & \int_0^\pi dt \int \sum_{j=0}^{2D-1} (-1)^j\, \mathrm{Tr}\Big(\sigma_z (d\hat{H}')^j\partial_t \hat{H}'(d\hat{H}')^{2D-1-j}\hat{\theta}_\Gamma\Big)) \\
& - \sum_{j=0}^{2D-2} (-1)^j\, \mathrm{Tr}\Big(\sigma_z \hat{H}'(d\hat{H}')^j\partial_t \hat{H}'(d\hat{H}')^{2D-j-2}(d\hat{\theta}_\Gamma)\Big).
\end{aligned}
\tag{85}
$$

Because $(\hat{H}')^2 = \mathbb{1}$ we can insert it in the first term between $(d\hat{H}')^j$ and $\partial_t \hat{H}'$. Using the fact the anti-commutation relations $d(\hat{H}')^2 = 0 = \{\hat{H}', d\hat{H}'\}$ as well as $\partial_t(\hat{H}')^2 = 0 = \{\hat{H}', \partial_t \hat{H}'\}$, we can show that

$$
\begin{aligned}
\mathcal{W}_{2D-1}(\hat{H}')/a_D = & \int_0^\pi dt \int \sum_{j=0}^{2D-1} -\frac{1}{2}\mathrm{Tr}\Big(\sigma_z (d\hat{H}')^j\hat{H}'\partial_t \hat{H}'(d\hat{H}')^{2D-1-j}[\hat{\theta}_\Gamma, \hat{H}']\Big)) \\
& - \sum_{j=0}^{2D-2} \mathrm{Tr}\Big(\sigma_z (d\hat{H}')^j\hat{H}'\partial_t \hat{H}'(d\hat{H}')^{2D-j-2}(d\hat{\theta}_\Gamma)\Big)
\end{aligned}
\tag{86}
$$

Because in this formulation we have terms with either commutator of the cut-off or differential of it, the index is localised on the shell and we can use that $\hat{H}_F^2 = \mathbb{1}$ in this region and so $\hat{H}_t' = \sin(t)\sigma_y \hat{H}_F - \sigma_x \cos t$. There for we have $\partial_t \hat{H}' = \cos(t)\sigma_y \hat{H}_F + \sigma_x \sin t$, $d\hat{H}' = \sin(t)\sigma_y(d\hat{H}_F)$ and $[\hat{\theta}_\Gamma, \hat{H}'] = \sin(t)\sigma_y[\hat{\theta}_\Gamma, \hat{H}_F]$ which leads to

$$
\begin{aligned}
\mathcal{W}_{2D-1}(\hat{H}')/a_{2D-1} = & i\Big(\int_0^\pi dt \sin(t)^{2D}\Big)2D \int \mathrm{Tr}\Big(\hat{H}_F(d\hat{H}_F)^{2D-1}[\hat{\theta}_\Gamma, \hat{H}_F]\Big)) \\
& + i\Big(\int_0^\pi dt \sin(t)^{2D-2}\Big)2(2D-1) \int \mathrm{Tr}\Big(\hat{H}_F(d\hat{H}_F)^{2D-2}(d\hat{\theta}_\Gamma)\Big)
\end{aligned}
\tag{87}
$$

If we now use that $\int_0^\pi dt \sin(t)^{2D} = \pi \frac{(2D)!}{2^{2D}D!^2}$ and $a_{2D-1} = \frac{D!}{(2D)!(-2i\pi)^D}$ we obtain $a_{2D}i \int_0^\pi dt \sin(t)^{2D}2D = -b_{2D-2}/2$ and $a_{2D}i \int_0^\pi dt \sin(t)^{2D-2}2(2D-1) = -b_{2D-2}$ which leads to

$$
\mathcal{I}_\mathrm{M} \equiv \mathcal{W}_{2D-1} = -b_{2D-2}\int \left(\mathrm{Tr}\Big(\hat{H}_F(d\hat{H}_F)^{2D-2}(d\hat{\theta}_\Gamma)\Big) + \frac{1}{2}\mathrm{Tr}\Big(\hat{H}_F(d\hat{H}_F)^{2D-1}[\hat{\theta}_\Gamma, \hat{H}']\Big)\right) \equiv \mathcal{I}_\mathrm{S}
\tag{88}
$$

which is the wanted mode-shell correspondence (51).

# B  index semiclassical expansion

In this section we want to prove the equation (54) for the semi-classical expansion of the mode index. To do that we will prove a similar statement, but at the operator level, which reduces to such equation

in the semi-classical limit. The statement we prove in this section is that the mode indices verify the general equality

$$
\begin{aligned}
\mathcal{I}_{\mathrm{M}} &= b_{\mathrm{D_M}} \int \mathrm{Tr}\Big(\hat{H}'(d\hat{H}')^{\mathrm{D_M}}\hat{\theta}_\Gamma\Big) = b_{\mathrm{D'}}(2\pi)^{D_\perp} \int \mathrm{Tr}\Big(\hat{H}'[\hat{\alpha},\hat{H}']^{\mathrm{D'}}\hat{\theta}_\Gamma\Big) \\
&= a_{\mathrm{D_M}} \int \mathrm{Tr}\Big(\hat{C}\hat{H}'(d\hat{H}')^{\mathrm{D_M}}\hat{\theta}_\Gamma\Big) = a_{\mathrm{D'}}(2\pi)^{D_\perp} \int \mathrm{Tr}\Big(\hat{C}\hat{H}'[\hat{\alpha},\hat{H}']^{\mathrm{D'}}\hat{\theta}_\Gamma\Big)
\end{aligned}
\tag{89}
$$

with D' = $\mathrm{D_M}$ +2 $\mathrm{D}_\perp$ and where we use the notation

$$
\begin{aligned}
\int \mathrm{Tr}\Big(\hat{H}'[\hat{\alpha},\hat{H}']^{\mathrm{D'}}\hat{\theta}_\Gamma\Big) &= \sum_{j_1,\ldots,j_{\mathrm{D'}}=1}^{\mathrm{D'}} \epsilon_{j_1,\ldots,j_{\mathrm{D'}}} \int \mathrm{Tr}\Big( \hat{H}' \prod_{m=1}^{\mathrm{D'}} [\hat{a}_{j_m},\hat{H}']\hat{\theta}_\Gamma \Big) \\
\int \mathrm{Tr}\Big(\hat{C}\hat{H}'[\hat{\alpha},\hat{H}']^{\mathrm{D'}}\hat{\theta}_\Gamma\Big) &= \sum_{j_1,\ldots,j_{\mathrm{D'}}=1}^{\mathrm{D'}} \epsilon_{j_1,\ldots,j_{\mathrm{D'}}} \int \mathrm{Tr}\Big( \hat{C}\hat{H}' \prod_{m=1}^{\mathrm{D'}} [\hat{a}_{j_m},\hat{H}']\hat{\theta}_\Gamma \Big)
\end{aligned}
\tag{90}
$$

in which $\epsilon_{j_1,\ldots,j_{\mathrm{D'}}}$ is the totally anti-symmetric Levi-Civita tensor and where the operators $\hat{a}_j$ are defined such that for $j \in [1,\mathrm{D_M}]$, $\hat{a}_j = \partial_{\lambda_j}$ with then $\hat{a}_{\mathrm{D_M}+2j-1} = ix_j$ and $\hat{a}_{\mathrm{D_M}+2j-1} = \partial_{x_j}$ in the continuous case or $\hat{a}_{\mathrm{D_M}+2j-1} = -iT_j^\dagger \hat{n}_j$ and $\hat{a}_{\mathrm{D_M}+2j} = T_j$ in the discrete case. We will use the convention $[\partial_{\lambda_j},A] \equiv (\partial_{\lambda_j}A)$ which is natural as we also have that commutators are mapped to derivative in the semi-classical limit $[\partial_{x_j},\hat{A}] \xrightarrow{S-C} \partial_{x_j}A$, $[i\hat{x}_j,\hat{A}] \xrightarrow{S-C} \partial_{k_j}A$. In all the computations of the proof, the differential acts mostly in the same way as a commutator which allows such an abuse of notation. [8]

In general, in this section, we will use notation $\alpha$ when the object should be understood as an anti-symetrised sum of all possible order of product of $\hat{a}_j$ so it means that for arbitrary operator $A_0,\ldots,A_{\mathrm{D'}}$ we have that

$$
A_0 \prod_{m=1}^{\mathrm{D'}} \alpha A_m = \sum_{j_1,\ldots,j_{\mathrm{D'}}=1}^{\mathrm{D'}} \epsilon_{j_1,\ldots,j_{\mathrm{D'}}} A_0 \prod_{m=1}^{\mathrm{D'}} \hat{a}_{j_m} A_m
\tag{91}
$$

It is relatively simple to see what the equality (90) reduces, in the semi-classical limit to the result (54) described in the main text. Indeed, using the fundamental properties of the Wigner-Weyl transform described in appendix B of [1], we know that, in the semi-classical limit, the commutator $[\hat{a}_{\mathrm{D_M}+2j-1} = ix_j,\hat{H}']$ are replaced by the derivative $\partial_{k_j}\hat{H}'$ and that the commutator $[\hat{a}_{\mathrm{D_M}+2j-2} = [\partial_{x_j},\hat{H}']$ are replaced by the derivative $\partial_{k_j}\hat{H}'$ (and similarly for discrete dimensions). So the commutators are replaced by differential and we simply obtain that

$$
\mathcal{I}_{\mathrm{M}} = \begin{cases} b_{\mathrm{D'}} \int \mathrm{Tr}\Big(\hat{H}'(d\hat{H}')^{\mathrm{D'}}\hat{\theta}_\Gamma\Big) \\ a_{\mathrm{D'}} \int \mathrm{Tr}\Big(\hat{C}\hat{H}'(d\hat{H}')^{\mathrm{D'}}\hat{\theta}_\Gamma\Big) \end{cases}
\tag{92}
$$

with D' = $\mathrm{D_M}$ +2 $\mathrm{D}_\perp$ which is the result (54) for $\mathrm{D}_\perp = 1$.

---

[8]The only difference is that the operators $\hat{a}_j = \partial_{\lambda_j}$ are only well defined when they appears in commutators where we have the convention $[\partial_{\lambda_j},A] \equiv (\partial_{\lambda_j}A)$. This will be the case most of the time in the proof. There is however some intermediary steps where $\hat{\alpha}$ does not appear in commutators which make the notation ill defined. We nevertheless decided to keep them in the proof as they allows to decompose the proof in more elementary steps which are easier to follow individually. In practice, in this proof, there is ways to skip those (ill-defined) intermediary steps, avoid those problems and obtain the same results.

So the difficult part is to prove the equality (89). We will do that by induction. The case $D_\perp = 0$ is obvious so we only need to prove the heredity. For that we will suppose that the formula is true for a $D_\perp$ and prove it for $D_\perp + 1$.

In all this proof, we can use that the cut-off commutes with any operator $[\hat{A}, \hat{\theta}_\Gamma] = 0$ as we are manipulating a mode index, involving $\hat{H}'$ which, by construction, is only non-trivial in the region where either $\hat{\theta}_\Gamma \approx \mathbb{1}$ or $\hat{\theta}_\Gamma \approx 0$.

We will prove the chiral and the non-chiral case separately.

## B.1   $D_M$ even, $\hat{H}$ chiral

We start for the expression of the mode index for $D_\perp$

$$\mathcal{I}_M = g \int \mathrm{Tr}\left( \hat{H}'[\hat{\alpha}, \hat{H}']^{D'} \hat{\theta}_\Gamma \right) \tag{93}$$

with the prefactor coefficient $g = b_{D'} (2\pi)^{D_\perp}$ and where $D' = D_M + 2 D_\perp$.

If we introduce the canonical conjugate operators associated with the new dimension. So, for a continuous dimension, $\hat{a}_{D'+1} = ix_{D_\perp+1}$ and $\hat{a}_{D'+2} = \partial_{x_{D_\perp+1}}$ and, for a discrete dimension, $\hat{a}_{D'+1} = -iT^\dagger_{D_\perp+1} \hat{n}_{D_\perp+1}$ and $\hat{a}_{D'+2} = T_{D_\perp+1}$. In both cases, we have the commutation relation $[\hat{a}_{D'+1}, \hat{a}_{D'+2}] = -i\mathbb{1}$. So we have that

$$\mathcal{I}_M = ig \int \mathrm{Tr}\left( [\hat{a}_{D'+1}, \hat{a}_{D'+2}] \hat{H}'[\hat{\alpha}, \hat{H}']^{D'} \hat{\theta}_\Gamma \right) \tag{94}$$

We then "integrating by part" the commutator using the identity $\mathrm{Tr}([A,B]C) = \mathrm{Tr}([A,BC]) - \mathrm{Tr}(B[A,C]) = -\mathrm{Tr}(B[A,C])$[9] to obtain that

$$\mathcal{I}_M = -ig \int \mathrm{Tr}\left( \hat{a}_{D'+2} \left[ \hat{a}_{D'+1}, \hat{H}'[\hat{\alpha}, \hat{H}']^{D_M + 2D_\perp - 2} \right] \hat{\theta}_\Gamma \right) \tag{95}$$

and then expand the commutator with the product using the identity $[AB, C] = [A, C]B + A[B, C]$[10] which gives us

$$\begin{aligned} \mathcal{I}_M = &-ig \int \mathrm{Tr}\left( \hat{a}_{D'+2} [\hat{a}_{D'+1}, \hat{H}'][\hat{\alpha}, \hat{H}']^{D'} \hat{\theta}_\Gamma \right) \\ &-ig \int \sum_{j=1}^{D'} \mathrm{Tr}\left( \hat{a}_{D'+2} \hat{H}'[\hat{\alpha}, \hat{H}']^{j-1} [\hat{\alpha}, [\hat{a}_{D'+1}, \hat{H}']][\hat{\alpha}, \hat{H}']^{D'-j} \hat{\theta}_\Gamma \right) \end{aligned} \tag{96}$$

To continue we use the property that $\alpha[\alpha, \hat{H}'] + [\alpha, \hat{H}']\alpha = [\alpha^2, \hat{H}'] = 0$[11] as $\alpha^2$ leads to anti-symetrised product $[\hat{a}_m, \hat{a}_n]$ which either vanishes or are proportional to the identity (when the operators are conjugates). therefore it commutes with $\hat{H}'$. This leads to the identity

$$[\alpha, \hat{H}']^j \alpha = (-1)^j \alpha [\alpha, \hat{H}']^j \tag{97}$$

---

[9] Similar to the identity $\int (dB)C = \int (d(BC)) - \int B(dC) = -\int B(dC)$ for differentials
[10] Similar to the identity $d(BC) = (dB)C + B(dC)$ for differentials
[11] Similar to the identity $d^2(B) = 0$ for differentials

that we can use, together with the cyclicity of the trace, the fact that $\epsilon_{j_1,j_2,...,j_{DT}} = -\epsilon_{j_{D'},j_1,...,j_{D'-1}}$ (as D' is even) and $[\hat{\alpha}, \hat{\theta}_\Gamma] = 0 = [\hat{\alpha}, \hat{a}_{D'+2}]$, to show that

$$
\begin{aligned}
\mathcal{I}_M = & - ig \int \text{Tr}\Big(\hat{a}_{D'+2}[\hat{a}_{D'+1}, \hat{H}'][\hat{\alpha}, \hat{H}']^{D'}\hat{\theta}_\Gamma\Big) \\
& - ig \int \sum_{j=1}^{D'} (-1)^j \text{Tr}\Big(\hat{a}_{D'+2}[\hat{\alpha}, \hat{H}'][\hat{\alpha}, \hat{H}']^{j-1}[\hat{a}_{D'+1}, \hat{H}'][\hat{\alpha}, \hat{H}']^{D'-j}\hat{\theta}_\Gamma\Big)
\end{aligned}
\tag{98}
$$

That we can regroup into a single sum

$$
\mathcal{I}_M = -ig \sum_{j=0}^{D'} (-1)^j \int \text{Tr}\Big(\hat{a}_{D'+2}[\hat{\alpha}, \hat{H}']^j[\hat{a}_{D'+1}, \hat{H}'][\hat{\alpha}, \hat{H}']^{D'-j}\hat{\theta}_\Gamma\Big).
\tag{99}
$$

Then we can insert the identity $\hat{H}'^2 = \mathbb{1}$ in such expression to obtain

$$
\mathcal{I}_M = -ig \sum_{j=0}^{D'} (-1)^j \int \text{Tr}\Big(\hat{H}'^2\hat{a}_{D'+2}[\hat{\alpha}, \hat{H}']^j[\hat{a}_{D'+1}, \hat{H}'][\hat{\alpha}, \hat{H}']^{D'-j}\hat{\theta}_\Gamma\Big).
\tag{100}
$$

then if use the identity $\hat{H}'[A, \hat{H}'] + [A, \hat{H}']\hat{H}' = [A, \hat{H}'^2] = 0$ as $\hat{H}'^2 = \mathbb{1}$, we obtain that

$$
\begin{aligned}
\mathcal{I}_M = & -ig/2 \sum_{j=0}^{D'} (-1)^j \int \text{Tr}\Big(\hat{H}'(\hat{H}'\hat{a}_{D'+2} + (-1)^{D'+1}\hat{a}_{D'+2}\hat{H}')[\hat{\alpha}, \hat{H}']^j[\hat{a}_{D'+1}, \hat{H}'][\hat{\alpha}, \hat{H}']^{D'-j}\hat{\theta}_\Gamma\Big) \\
= & ig/2 \sum_{j=0}^{D'} (-1)^j \int \text{Tr}\Big(\hat{H}'[\hat{a}_{D'+2}, \hat{H}'][\hat{\alpha}, \hat{H}']^j[\hat{a}_{D'+1}, \hat{H}'][\hat{\alpha}, \hat{H}']^{D'-j}\hat{\theta}_\Gamma\Big).
\end{aligned}
\tag{101}
$$

as $D'+1 = D_M + 2D_\perp + 1$ is odd as we are in the case where $D_M$ is even.

Using the cyclicity of the trace, and the anti-commutation relation, we can show that

$$
\begin{aligned}
\mathcal{I}_M = & \frac{ig}{2(D'+2)} \sum_{j+k\le D'} (-1)^j \int \text{Tr}\Big(\hat{H}'[\hat{\alpha}, \hat{H}']^k[\hat{a}_{D'+2}, \hat{H}'][\hat{\alpha}, \hat{H}']^j[\hat{a}_{D'+1}, \hat{H}'][\hat{\alpha}, \hat{H}']^{D'-j-k}\hat{\theta}_\Gamma \\
& - \hat{H}'[\hat{\alpha}, \hat{H}']^k[\hat{a}_{D'+1}, \hat{H}'][\hat{\alpha}, \hat{H}']^j[\hat{a}_{D'+2}, \hat{H}'][\hat{\alpha}, \hat{H}']^{D'-j-k}\hat{\theta}_\Gamma\Big)
\end{aligned}
\tag{102}
$$

which is an anti-symetrised sum over all possible position of the operator $\hat{a}_{D'+1}$ and $\hat{a}_{D'+2}$ in the product with a sign depending of the permutation. So in fact we have an anti-symetrised sum over the $D'+2$ components of $\hat{a}_j$

$$
\mathcal{I}_M = \frac{-ig}{2(D'+2)} \text{Tr}\Big(\hat{H}'[\alpha, \hat{H}']^{D'+2}\hat{\theta}_\Gamma\Big)
\tag{103}
$$

In a more detailed way, we can see that by decomposing the above expression using (90) and then depending on which $j_m$ is equal to $D'+1$ or $D'+2$. We denote the $n$ the $m$ for which $j_m = D'+1$ and $n'$ the one for which $j_m = D'+2$. We then have that

$$
\int \text{Tr}\Big(\hat{H}'[\hat{\alpha}, \hat{H}']^{D'+2}\hat{\theta}_\Gamma\Big) = \sum_{j_1,...,j_{D'+2}=1}^{D'+2} \sum_{n,n'} \delta_{j_n,D'+1}\delta_{j_{n'},D'+2}\epsilon_{j_1,...,j_{D'+2}} \int \text{Tr}\Big(\hat{H}' \prod_{m=1}^{D'+2}[a_{j_m}, \hat{H}']\hat{\theta}_\Gamma\Big).
\tag{104}
$$

That we decompose in two cases depending on if we have $n < n'$ or $n' > n$. If $n < n'$, we have that

$$\epsilon_{j_1,\ldots,j_n=\mathrm{D'}+1,\ldots,j_{n'}=\mathrm{D'}+2,\ldots,j_{\mathrm{D'}+2}} = (-1)^{\mathrm{D'}+2-n'}(-1)^{\mathrm{D'}+1-n}\epsilon_{j_1,\ldots,j_{\mathrm{D'}+2},\mathrm{D'}+1,\mathrm{D'}+2}$$

where $j_n$ and $j_{n'}$ are now omitted in the notation $j_1,\ldots,j_{\mathrm{D'}+2}$ in the expression $\epsilon_{j_1,\ldots,j_{\mathrm{D'}+2},\mathrm{D'}+1,\mathrm{D'}+2}$, so up to a relabeling we can denote them as $j_1,\ldots,j_{\mathrm{D'}}$ with $\epsilon_{j_1,\ldots,j_{\mathrm{D'}},\mathrm{D'}+1,\mathrm{D'}+2} = \epsilon_{j_1,\ldots,j_{\mathrm{D'}}}$. Moreover we have that $(-1)^{\mathrm{D'}+2-n'}(-1)^{\mathrm{D'}+1-n} = -(-1)^{-n'-n} = -(-1)^{n'-n}$ so

$$
\begin{aligned}
&\sum_{j_1,\ldots,j_{\mathrm{D'}+2}=1}^{\mathrm{D'}+2} \sum_{n<n'} \epsilon_{j_1,\ldots,j_n=\mathrm{D'}+1,\ldots,j_{n'}=\mathrm{D'}+2,\ldots,j_{\mathrm{D'}+2}} \int \mathrm{Tr}\left( \hat{H}' \prod_{m=1}^{\mathrm{D'}+2} [a_{j_m},\hat{H}']\hat{\theta}_\Gamma \right) \\
&= -\sum_{j_1,\ldots,j_{\mathrm{D'}}=1}^{\mathrm{D'}} \sum_{n<n'} (-1)^{n'-n}\epsilon_{j_1,\ldots,j_{\mathrm{D'}}} \int \mathrm{Tr}\left( \hat{H}' \prod_{m=1}^{n} [a_{j_m},\hat{H}'][\hat{a}_{\mathrm{D'}+1},\hat{H}'] \prod_{m=n+1}^{n'} [a_{j_m},\hat{H}'][\hat{a}_{\mathrm{D'}+2},\hat{H}'] \prod_{m=n'+1}^{\mathrm{D'}} [a_{j_m},\hat{H}']\hat{\theta}_\Gamma \right) \\
&= -\sum_{n<n'} (-1)^{n'-n} \int \mathrm{Tr}\left( \hat{H}'[\alpha,\hat{H}']^n [\hat{a}_{\mathrm{D'}+1},\hat{H}'][\alpha,\hat{H}']^{n'-n}[\hat{a}_{\mathrm{D'}+2},\hat{H}'][\alpha,\hat{H}']^{\mathrm{D'}-n'}\hat{\theta}_\Gamma \right)
\end{aligned}
$$

(105)

which up to the substitution $n = k$ and $n' - n = j$ is just the second term of (102).

Similarly, for $n > n'$ we have that

$$
\begin{aligned}
&\sum_{j_1,\ldots,j_{\mathrm{D'}+2}=1}^{\mathrm{D'}+2} \sum_{n>n'} \epsilon_{j_1,\ldots,j_n=\mathrm{D'}+1,\ldots,j_{n'}=\mathrm{D'}+2,\ldots,j_{\mathrm{D'}+2}} \int \mathrm{Tr}\left( \hat{H}' \prod_{m=1}^{\mathrm{D'}+2} [a_{j_m},\hat{H}']\hat{\theta}_\Gamma \right) \\
&= \sum_{n<n'} (-1)^{n'-n} \int \mathrm{Tr}\left( \hat{H}'[\alpha,\hat{H}']^{n'} [\hat{a}_{\mathrm{D'}+2},\hat{H}'][\alpha,\hat{H}']^{n-n'}[\hat{a}_{\mathrm{D'}+2},\hat{H}'][\alpha,\hat{H}']^{\mathrm{D'}-n'}\hat{\theta}_\Gamma \right)
\end{aligned}
$$

(106)

which is the first term of (102). The sign difference come from the fact that the index $j_{n'} = \mathrm{D'}+2$ now appear before the index $j_n = \mathrm{D'}+1$ in the Levi-Civita tensor $\epsilon_{j_1,\ldots,j_{\mathrm{D'}+2}}$ and so we need to exchange their order to obtain $\epsilon_{j_1,\ldots,j_{\mathrm{D'}},\mathrm{D'}+1,\mathrm{D'}+2}$ creating an additional sign.

Now that we know that

$$\mathcal{I}_\mathrm{M} = \frac{-ig}{2(\mathrm{D'}+2)} \mathrm{Tr}\left( \hat{H}'[\alpha,\hat{H}']^{\mathrm{D'}+2}\hat{\theta}_\Gamma \right) \tag{107}$$

we just need to check the prefactor which is

$$\frac{-ig}{2(\mathrm{D'}+2)} = \frac{-ib_{\mathrm{D_M}+2\,\mathrm{D_\perp}}(2\pi)^{\mathrm{D_\perp}}}{4(\mathrm{D_M}/2 + \mathrm{D_\perp}+1)} = b_{\mathrm{D_M}+2(\mathrm{D_\perp}+1)}(2\pi)^{\mathrm{D_\perp}+1} \tag{108}$$

so we have obtained the wanted form of (89) for $\mathrm{D_\perp}+1$ which complete the proof in the chiral case.

## B.2   $\mathrm{D_M}$ odd, $\hat{H}$ non-chiral

We start for the expression of the non-chiral mode index for $\mathrm{D_\perp}$

$$\mathcal{I}_\mathrm{M} = g \int \mathrm{Tr}\left( \hat{C}\hat{H}'[\hat{\alpha},\hat{H}']^{\mathrm{D'}}\hat{\theta}_\Gamma \right) \tag{109}$$

with the prefactor coefficient $g = a_{\mathrm{D'}}(2\pi)^{\mathrm{D_\perp}}$ and where $\mathrm{D'} = \mathrm{D_M}+2\,\mathrm{D_\perp}$.

If we introduce the canonical conjugate operators associated with the new dimension. So, for a continuous dimension, $\hat{a}_{\mathrm{D'}+1} = ix_{\mathrm{D_\perp}+1}$ and $\hat{a}_{\mathrm{D'}+2} = \partial_{x_{\mathrm{D_\perp}+1}}$ and, for a discrete dimension, $\hat{a}_{\mathrm{D'}+1} =$

$-iT^\dagger_{D_\perp +1}\hat{n}_{D_\perp +1}$ and $\hat{a}_{D'+2} = T_{D_\perp +1}$. In both cases, we have the commutation relation $[\hat{a}_{D'+1}, \hat{a}_{D'+2}] = -i\mathbb{1}$. So we have that

$$\mathcal{I}_M = ig \int \mathrm{Tr}\Big(\hat{C}[\hat{a}_{D'+1}, \hat{a}_{D'+2}]\hat{H}'[\hat{\alpha}, \hat{H}']^{D'}\hat{\theta}_\Gamma\Big) \tag{110}$$

We then "integrating by part" the commutator using the identity $\mathrm{Tr}([A, B]C) = \mathrm{Tr}(\cancel{[A, BC]}) - \mathrm{Tr}(B[A, C]) = -\mathrm{Tr}(B[A, C])$ to obtain that

$$\mathcal{I}_M = -ig \int \mathrm{Tr}\Big(\hat{C}\hat{a}_{D'+2}\Big[\hat{a}_{D'+1}, \hat{H}'[\hat{\alpha}, \hat{H}']^{D_M+2D_\perp -2}\Big]\hat{\theta}_\Gamma\Big) \tag{111}$$

and then expand the commutator with the product using the identity $[AB, C] = [A, C]B + A[B, C]$ which gives us

$$\begin{aligned}\mathcal{I}_M = &-ig \int \mathrm{Tr}\Big(\hat{C}\hat{a}_{D'+2}[\hat{a}_{D'+1}, \hat{H}'][\hat{\alpha}, \hat{H}']^{D'}\hat{\theta}_\Gamma\Big) \\ &-ig \int \sum_{j=1}^{D'} \mathrm{Tr}\Big(\hat{C}\hat{a}_{D'+2}\hat{H}'[\hat{\alpha}, \hat{H}']^{j-1}[\hat{\alpha}, [\hat{a}_{D'+1}, \hat{H}']][\hat{\alpha}, \hat{H}']^{D'-j}\hat{\theta}_\Gamma\Big)\end{aligned} \tag{112}$$

To continue we use the property that $\alpha[\alpha, \hat{H}'] + [\alpha, \hat{H}']\alpha = [\alpha^2, \hat{H}'] = 0$ as $\alpha^2$ leads to anti-symetrised product $[\hat{a}_m, \hat{a}_n]$ which either vanishes or are proportional to the identity (when the variables are conjugates). therefore it commutes with $\hat{H}'$. This leads to the identity

$$[\alpha, \hat{H}']^j\alpha = (-1)^j\alpha[\alpha, \hat{H}']^j \tag{113}$$

that we can use, together with the cyclicity of the trace and that $\epsilon_{j_1, j_2, ..., j_{D'}} = +\epsilon_{j_{D'}, j_1, ..., j_{D'-1})}$ (as D' is odd), to show that

$$\begin{aligned}\mathcal{I}_M = &-ig \int \mathrm{Tr}\Big(\hat{C}\hat{a}_{D'+2}[\hat{a}_{D'+1}, \hat{H}'][\hat{\alpha}, \hat{H}']^{D'}\hat{\theta}_\Gamma\Big) \\ &-ig \int \sum_{j=1}^{D'}(-1)^j \mathrm{Tr}\Big(\hat{C}\hat{a}_{D'+2}[\hat{\alpha}, \hat{H}'][\hat{\alpha}, \hat{H}']^{j-1}[\hat{a}_{D'+1}, \hat{H}'][\hat{\alpha}, \hat{H}']^{D'-j}\hat{\theta}_\Gamma\Big)\end{aligned} \tag{114}$$

That we can regroup into a single sum

$$\mathcal{I}_M = -ig \sum_{j=0}^{D'}(-1)^j \int \mathrm{Tr}\Big(\hat{C}\hat{a}_{D'+2}[\hat{\alpha}, \hat{H}']^j[\hat{a}_{D'+1}, \hat{H}'][\hat{\alpha}, \hat{H}']^{D'-j}\hat{\theta}_\Gamma\Big). \tag{115}$$

Then we can insert the identity $\hat{H}'^2 = \mathbb{1}$ in such expression to obtain

$$\mathcal{I}_M = -ig \sum_{j=0}^{D'}(-1)^j \int \mathrm{Tr}\Big(\hat{C}\hat{H}'^2\hat{a}_{D'+2}[\hat{\alpha}, \hat{H}']^j[\hat{a}_{D'+1}, \hat{H}'][\hat{\alpha}, \hat{H}']^{D'-j}\hat{\theta}_\Gamma\Big). \tag{116}$$

then if use the identity $\hat{H}'[A, \hat{H}'] + [A, \hat{H}']\hat{H}' = [A, \hat{H}'^2] = 0$ as $\hat{H}'^2 = \mathbb{1}$, we obtain that

$$\begin{aligned}\mathcal{I}_M = &-ig/2 \sum_{j=0}^{D'}(-1)^j \int \mathrm{Tr}\Big(\hat{C}\hat{H}'(\hat{H}'\hat{a}_{D'+2} + (-1)^{D'+2}\hat{a}_{D'+2}\hat{H}')[\hat{\alpha}, \hat{H}']^j[\hat{a}_{D'+1}, \hat{H}'][\hat{\alpha}, \hat{H}']^{D'-j}\hat{\theta}_\Gamma\Big) \\ = &ig/2 \sum_{j=0}^{D'}(-1)^j \int \mathrm{Tr}\Big(\hat{C}\hat{H}'[\hat{a}_{D'+2}, \hat{H}'][\hat{\alpha}, \hat{H}']^j[\hat{a}_{D'+1}, \hat{H}'][\hat{\alpha}, \hat{H}']^{D'-j}\hat{\theta}_\Gamma\Big).\end{aligned} \tag{117}$$

as D' $+2 =$ D$_\mathrm{M}$ $+2\,$D$_\perp$ $+2$ is odd as we are in the case where D$_\mathrm{M}$ is odd.

Using the cyclicity of the trace, and the anti-commutation relation, we can show that

$$
\begin{aligned}
\mathcal{I}_\mathrm{M} =& \frac{ig}{2(\mathrm{D'}+2)} \sum_{j+k\leq \mathrm{D'}} (-1)^j \int \mathrm{Tr}\left(\hat{C}\hat{H}'[\hat{\alpha},\hat{H}']^k[\hat{a}_{\mathrm{D'}+2},\hat{H}'][\hat{\alpha},\hat{H}']^j[\hat{a}_{\mathrm{D'}+1},\hat{H}'][\hat{\alpha},\hat{H}']^{\mathrm{D'}-j-k}\hat{\theta}_\Gamma \right. \\
& \left. - \hat{C}\hat{H}'[\hat{\alpha},\hat{H}']^k[\hat{a}_{\mathrm{D'}+1},\hat{H}'][\hat{\alpha},\hat{H}']^j[\hat{a}_{\mathrm{D'}+2},\hat{H}'][\hat{\alpha},\hat{H}']^{\mathrm{D'}-j-k}\hat{\theta}_\Gamma \right)
\end{aligned}
\tag{118}
$$

which is an anti-symetrised sum over all possible position of the operator $\hat{a}_{\mathrm{D'}+1}$ and $\hat{a}_{\mathrm{D'}+2}$ in the product with a sign depending of the permutation. So in fact we have an anti-symetrised sum over the D' $+2$ components of $\hat{a}_j$

$$
\mathcal{I}_\mathrm{M} = \frac{-ig}{2(\mathrm{D'}+2)} \mathrm{Tr}\left(\hat{C}\hat{H}'[\alpha,\hat{H}']^{\mathrm{D'}+2}\hat{\theta}_\Gamma\right)
\tag{119}
$$

The pre-factor coefficient is therefore

$$
\frac{-ig}{2(\mathrm{D'}+2)} = \frac{-ia_{\mathrm{D_M}+2\,\mathrm{D}_\perp}(2\pi)^{\mathrm{D}_\perp}}{2(\mathrm{D_M}+2\,\mathrm{D}_\perp+2)} = a_{\mathrm{D_M}+2(\mathrm{D}_\perp+1)}(2\pi)^{\mathrm{D}_\perp+1}
\tag{120}
$$

so we have obtained the wanted form of (89) for D$_\perp$ $+1$ which complete the proof in the chiral case.

# C   Mode indices without partial semi-classical approximation

In the main text we construct mode topological indices of Hamiltonian $H(\lambda_i)$ depending of D$_\mathrm{M}$ independent parameters $\lambda_i$. In a lot of important cases those parameters are wavenumbers of Hamiltonians in the directions where the mode index is delocalised. For example, we discussed that the mode index describing the existence of unidirectional modes at the edges of a 2D topological insulator can be expressed as

$$
\mathcal{I}_\mathrm{M} = \frac{1}{4\pi i} \int dk_y\, \mathrm{Tr}\left(\hat{H}'\partial_{k_y}\hat{H}'(k_y)\hat{\theta}_\Gamma(x)\right)
\tag{121}
$$

where $y$ is the direction parallel to the edge and the cut-off $\hat{\theta}_\Gamma(x)$ does a selection in $x$ which is the orthogonal direction.

In such expression, $\hat{H}'(k_y)$ is a partial Wigner-Weyl transform of the full operator $\hat{H}'$ in the 2D lattice. A Wigner-Weyl transform is performed only in the $y$ direction (and the $x$ direction left unchanged). Such kind of expression makes sense when the original Hamiltonian $\hat{H}$ is invariant by translation in the $y$ direction $\hat{H}(k_y)$ is then just the Bloch transform) or if it verifies some semi-classical limits in the $y$ direction[12]. This is a problem for studying systems with disorder where semi-classical limit is, most of the time, not achieved.

The goal of this appendix is to provide another formulation of the mode indices defined in the main text, that are defined in the full operator picture without the need of any semi-classical approximation. The general principle is to replace any derivative of $\hat{H}'(k)$ in a wavenumber direction $k_{x_j}$ by a commutator with an operator $\hat{\theta}_{x_j}$ which is diagonal in position and goes from zero to one in $x_j$ (see figure 15) together with a $2\pi i$ prefactor.

$$
\frac{1}{2\pi i}\partial_{k_j}\hat{H}'(k) \rightarrow [\hat{\theta}_{x_j},\hat{H}'(k)].
\tag{122}
$$

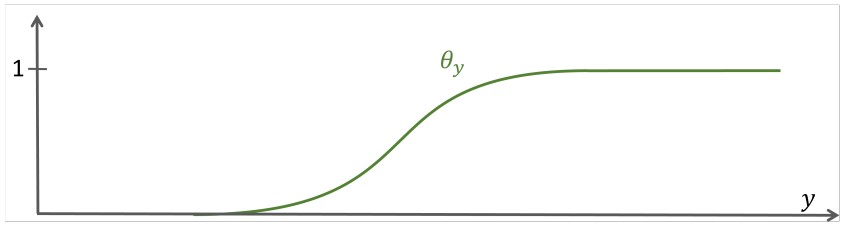

Figure 15: Example of a possible function $\theta_x$ that could be used to generate the operator $\hat{\theta}_x$

For example, if we apply such transformation for the 1D-mode index (121), we obtain an index which reads

$$\mathcal{I}_{\mathrm{M}} = \frac{1}{2} \operatorname{Tr}\left( \hat{H}'[\hat{\theta}_y, \hat{H}']\hat{\theta}_\Gamma(x) \right).\tag{123}$$

We can make a few remarks about such an expression. First such indices are defined at the full operator level without any semi-classical approximation needed and therefore robust to disorder. Secondly if such semi-classical approximation is valid in the $y$ direction, the commutator reduces as $[\hat{\theta}_y, \hat{H}'] \rightarrow i\{\theta_y(y), \hat{H}'(k_y)\} = i\partial_y\theta_y\partial_{k_y}\hat{H}'$ and the full trace is replaced $\operatorname{Tr} \rightarrow \frac{1}{2\pi}\int dy \int dk_y \operatorname{Tr}$ where the second trace only acts on the partial symbol. We can integrate in $y$ using $\int dy\partial_y\theta_y = \theta_y(\infty) - \theta_y(-\infty) = 1$. We therefore obtain than in the semi-classical limits, the above expression reduces to the mode index (121) discussed in the main text.

Similarly for the other modes indices discussed in the main text

$$\mathcal{I}_{\mathrm{M}} = b_{\mathrm{D_M}} \int \operatorname{Tr}\left( \hat{H}'(d\hat{H}')^{\mathrm{D_M}}\hat{\theta}_\Gamma \right)$$
$$\mathcal{I}_{\mathrm{M}} = a_{\mathrm{D_M}} \int \operatorname{Tr}\left( \hat{C}\hat{H}'(d\hat{H}')^{\mathrm{D_M}}\hat{\theta}_\Gamma \right)\tag{124}$$

we can replace them by indices at the operator level

$$\mathcal{I}_{\mathrm{M}} = b_{\mathrm{D_M}}(i2\pi)^{\mathrm{D_M}} \int \operatorname{Tr}\left( \hat{H}'[\hat{\alpha}, \hat{H}']^{\mathrm{D'}}\hat{\theta}_\Gamma \right)$$
$$\mathcal{I}_{\mathrm{M}} = a_{\mathrm{D_M}} \int \operatorname{Tr}\left( \hat{C}\hat{H}'[\hat{\alpha}, \hat{H}']^{\mathrm{D'}}\hat{\theta}_\Gamma \right)\tag{125}$$

where the differentials $d\hat{H}' = \sum_j \partial_{k_j}\hat{H}'dk_j$ are replaced by commutators $[\hat{\alpha}, \hat{H}'] = \sum_j [\hat{\theta}_{x_j}, \hat{H}']dk_j$.

Doing computation with commutators $[\hat{\theta}_{x_j}, \hat{H}']$ is identical than doing them with differentials $\partial_{k_j}\hat{H}'$ as they both verify a leibniz rule ($\partial(AB) = (\partial A)B + A(\partial B)$ and $[C, AB] = [C, A]B + A[C, B]$) and the fact that a total derivative/commutator vanishes when integrated/traced ($\int \operatorname{Tr}(\partial(A)) = 0$ and $\operatorname{Tr}([C, A]) = 0$). So any results obtained for the mode indices (124) are also verified for the mode indices (125). In particular they also verify a mode-shell correspondence where derivative are just replaced by commutator in the expression of the shell index.

This generalisation therefore allow to study topological behavior and the mode-shell correspondence in disordered systems.

---

[12]Note the we only need this to be true in the $y$ direction. Near the edge, the Hamiltonian varies strongly in the $x$ direction which is a typical situation where semi-classical limit is not possible. However this is not a problem as we are in operator formalism in such direction and do not rely on Wigner-Weyl transform in $x$.

# D   Dirac/Weyl cones at interfaces of continuous system

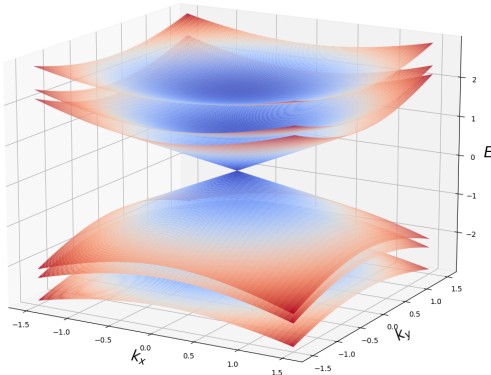

Figure 16: Spectrum of the Hamiltonian $\hat{H}_{\text{Dirac-JR}}$. The model has one Dirac cones while the other bands are gapped.

In this appendix, we study models hosting Dirac cones which are localised in position. For that we consider some continuous model which tends to be simple to analyse than their discrete counterpart wediscuss in the next section. To obtain a model confined in position, we can again, we can use the additive construction to combine a Dirac/Weyl Hamiltonian with a Jackiw-Rebbi model which confines the mode in the new direction.

**3D model with a Dirac cone confined at an interface**  For the Dirac case, such a construction gives the model

$$
\hat{H}_{\text{Dirac-JR}} = \hat{H}_{\text{Dirac}} \boxplus \hat{H}_{\text{JR,z}} = k_x \boxplus k_y \boxplus \hat{H}_{\text{JR,z}}
$$
$$
= \begin{pmatrix}
0 & k_x - ik_y & z - \partial_z & 0 \\
k_x + ik_y & 0 & 0 & z - \partial_z \\
z + \partial_z & 0 & 0 & -k_x + ik_y \\
0 & z + \partial_z & -k_x - ik_y & 0
\end{pmatrix} \tag{126}
$$

Since this model is an additive construction, we know that its eigenvalue are of the form $\sqrt{k_x^2 + k_y^2 + \lambda_n^2}$ where $\lambda_n$ are the eigenvalue of the Jackiw-Rebbi model (see figure 16). Because such a model has a zero modes $\psi_0 = (\exp(-z^2/2), 0)^T$ with $\lambda_0 = 0$, the model $\hat{H}_{\text{Dirac-JR}}$ has a Dirac cone centered in $(k_x, k_y) = (0, 0)$ and near $z \sim 0$. In particular the reduced Hamiltonian for modes of the form $\psi(k_x, k_y) \otimes \psi_0$ is just the Dirac Hamiltonian $\hat{H}_{\text{Dirac}}$ of positive chirality. So the mode index of such a model is $\mathcal{I}_{\text{modes}} = 1$.

The shell index is meanwhile the higher winding number on the 3-sphere $k_x^2 + k_y^2 + z^2 + k_z^2 = \Gamma^2$ of the symbol

$$
H_{\text{Dirac-JR}} = \begin{pmatrix}
0 & k_x - ik_y & z - ik_z & 0 \\
k_x + ik_y & 0 & 0 & z - ik_z \\
z + ik_z & 0 & 0 & -k_x + ik_y \\
0 & z + ik_z & -k_x - ik_y & 0
\end{pmatrix} \tag{127}
$$

Such a symbol is similar to the Jackiw-Rossi model we studied in the first example of section 4.3 of part I [1]. Therefore we know that the shell index is $\mathcal{I}_{\text{S}} = \mathcal{W}_3(H_F) = 1$. So the mode-shell correspondence is verified.

**4D model with a Weyl cone confined at an interface** For the Dirac case, such a construction gives the model

$$\hat{H}_{\text{Weyl-JR}} = \hat{H}_{\text{Weyl}} \boxplus \hat{H}_{\text{JR,u}} = k_x \boxplus k_y \boxplus k_z \boxplus \hat{H}_{\text{JR,u}}$$

$$= \begin{pmatrix} k_z & k_x - ik_y & u - \partial_u & 0 \\ k_x + ik_y & -k_z & 0 & u - \partial_u \\ u + \partial_u & 0 & -k_z & -k_x + ik_y \\ 0 & u + \partial_u & -k_x - ik_y & k_z \end{pmatrix} \tag{128}$$

Since this model is an additive construction, we know that its eigenvalue are of the form $\sqrt{k_x^2 + k_y^2 + k_z^2 + \lambda_n^2}$ where $\lambda_n$ are the eigenvalue of the Jackiw-Rebbi model (see figure 16). Because such a model has a zero modes $\psi_0 = (\exp(-u^2/2), 0)^T$ with $\lambda_0 = 0$, the model $\hat{H}_{\text{Weyl-JR}}$ has a Weyl cone centered in $(k_x, k_y, k_z) = (0, 0)$ and near $u \sim 0$. In particular the reduced Hamiltonian for modes of the form $\psi(k_x, k_y, k_z) \otimes \psi_0$ is just the Weyl Hamiltonian $\hat{H}_{\text{Weyl}}$ of positive chirality. So the mode index of such a model is $\mathcal{I}_{\text{modes}} = 1$.

The shell index is meanwhile the higher winding number on the 4-sphere $k_x^2 + k_y^2 + k_z^2 + u^2 + k_u^2 = \Gamma^2$ of the symbol

$$H_{\text{Weyl-JR}} = \begin{pmatrix} k_z & k_x - ik_y & u - ik_u & 0 \\ k_u + ik_u & -k_z & 0 & u - ik_u \\ u + ik_u & 0 & -k_z & -k_u + ik_u \\ 0 & u + ik_u & -k_x - ik_y & k_z \end{pmatrix} \tag{129}$$

Such a symbol is similar to the higher-order model with spectral flow we studied in section 3.5. Therefore we know that the shell index is $\mathcal{I}_S = \mathcal{C}_4(H_F) = 1$. So the mode-shell correspondence is verified.