# Peer review of "Mode-Shell correspondence, a unifying phase space theory in topological physics -- part II: Higher-dimensional spectral invariants"

_SciPost Physics_

## Round 1 · Referee Report · Anonymous (Referee 2) · 2025-3-10

Report

The authors expand the topological correspondence they had previously developed for the zero-modes of chiral-symmetric Hamiltonians, which they call the "mode-shell correspondence". The latter is now established for systems in class A and AIII, and is applicable to a wide range of cases, from strong topological phases to WTIs, HOTIs, and topolological semimetals.

I greatly enjoyed reading this paper. Despite it being based on previous work, I found it to be self-contained, with a pedagogical and easy to follow introduction that is supplemented by multiple examples of how the correspondence works in practice. I believe these results represent a valuable addition to the field of topological phases: their theory generalizes the well-known bulk-boundary correspondence and provides a unified approach that can be used to treat a variety of different systems. I believe that extensions of this approach will also prove useful in studying the other 8 symmetry classes, as well as the topology of non-Hermitian and Floquet systems.

Except for encouraging the authors to carefully spell-check their paper, I have no other suggestions for improvement. My recommendation is to publish this work in SciPost Physics.

Recommendation

Publish (easily meets expectations and criteria for this Journal; among top 50%)

---

## Round 1 · Referee Report · Isidora Araya Day (Referee 1) · 2025-3-10

Strengths

  • Novel and very interesting approach
  • Applies to Hamiltonians that depend on position and/or momentum
  • Generalization to high mode dimensions
  • Analysis of several examples

Weaknesses

  • Difficult to read
  • Code to reproduce results is not available
  • Only applies to class A and AIII

Report

The manuscript introduces a generalization of the Mode-Shell correspondence to modes of finite dimension in classes A and AIII. The approach is very interesting because it uses momentum and position on equal footing, providing a new perspective on the identification of topological phases. In the case of translational invariant Hamiltonians, the Mode-Shell correspondence agrees with the well-known bulk-edge correspondence. The manuscript provides several toy model examples with their Mode-Shell invariant and claims that the approach works for systems with disorder, although it does not demonstrate how to do so. I recommend to publish after a minor revision.

Questions: 1. Equation (1) says $D_M + D_S = 2D -1$, where does this come from? I suggest the authors to give a concrete example like with the Chern insulator in the paragraph right before. 2. Does selecting the cut-off function $\theta_\Gamma$ require knowing how the modes behave? is there a way to construct $\theta_\Gamma$ systematically? In the case of Fig 7 I can see where the dispersion crosses $E=0$, but how do I approach the problem if this dispersion is expensive to compute? 3. What determines the integral limits for $t$ in the expression for $\mathcal{I}_{\textrm{s-f}}$, Eq. 8 for example? 4. In 3.5 the mode-shell correspondence is applied to HOTIs. If I understand correctly, a spectral flow Hamiltonian is constructed from the HOTI Hamiltonian to construct the mode-shell invariant. It is not clear to me why both Hamiltonians must share the same topological classification. Why is this the case?

Requested changes

  1. I would like to request the authors to provide the code for at least some examples. I believe a clear demonstration of how to apply the mode-shell correspondence would help the readability of the manuscript. This is also point 6 of Scipost's general acceptance criteria.
  2. I believe that $\theta_\Gamma$ is first introduced by referencing the work mode-shell correspondence I, and $\Gamma$ is not introduced. I suggest the authors to introduce these variables and to mention in text that semi-classical limit is for $\Gamma \to \infty$ right before Eq. 11. At the moment it is only included in figure 6.
  3. "The main problem in condensed-matter applications is that defects in the lattice structure (vacancies, impurities, ...) induce perturbations which do not vary slowly compared to the inter-site distance." I would like the authors to make this statement more precise. Is this the main problem of mode shell correspondence or of condensed matter theory? Why?
  4. "Such an approach is particularly useful to tackle topological modes in disordered systems where the semi-classical analysis is not suited, and so are neither the semi-classical shell invariants." I would like the authors to explain in more detail how the approach applies to disordered systems. They could consider including an demonstration. In particular, how does the approach scale in systems with disorder? These tend to be big and $H_F$ would naively be expensive to compute.

Recommendation

Ask for minor revision

  • validity: high
  • significance: high
  • originality: high
  • clarity: good
  • formatting: excellent
  • grammar: excellent

Author:  Lucien Jezequel  on 2025-04-03  [id 5336]

(in reply to Report 1 by Isidora Araya Day on 2025-03-10)

We thank the referee for the valuable feedback. Please find the answers to the questions and requested changes below

Questions:

1) Equation (1) says $D_M+D_S=2D - 1$, where does this come from? I suggest the authors to give a concrete example like with the Chern insulator in the paragraph right before

This is not a result that we explicitly demonstrate as a mathematical theorem, but a relation that appears naturally in the construction of the invariants and that we observe to be always satisfied in all the cases we address, namely: strong topological insulators (TI), weak TIs, higher-order TIs, semimetals, 1D quantum channels. We have rephrased the sentence before the equation (1) to clarify that this is an observation that allows us to classify any gapless modes.

Indeed, the case of the Chern insulator is shown just above ($D=2$, $D_M=1$, $D_S=2$) and obviously satisfies this equation $(1+2 = 2\times 2-1)$. We intentionally provide the values of $D$, $D_S$ and $D_M$ (or equivalently $D_\perp$ introduced later) to stress the validation of equation (1) throughout the manuscript, so that many other examples are given.

2) Does selecting the cut-off function $\theta_\Gamma$ require knowing how the modes behave? is there a way to construct $\theta_\Gamma$ systematically? In the case of Fig 7 I can see where the dispersion crosses E=0, but how do I approach the problem if this dispersion is expensive to compute?

To choose the cut-off function we need to know in which directions the gapless mode is localized (or conversely disperses), in order to enclose the gapless mode. In the end, only the topology of the shell is important. We however do not need the exact dispersion relation to define the cut-off function. The information about where the mode is localized can be obtained through several methods, for instance semi-classical analysis of the symbol (gapless modes of the operator must decays exponentially in regions where the symbol of the operator is gapped), exact diagonalisation (e.g. numerics), perturbation of known model.

We built our theory by starting from the existence of gapless modes in phase space to eventually derive an expression of a shell invariant those are related to. But the theory could be used in the other way: now that this correspondence is established, we could just compute the shell invariant (Chern numbers, winding numbers) to infer the existence of specific gapless modes that disperse along particular directions. So, in the derivation of our theory, we needed to know a bit about the dispersion of the gapless modes, but one does not need it to apply the theory and predict gapless modes by computing the shell invariant.

3) What determines the integral limits for t in the expression for $I_{S-F}$, Eq. 8 for example?

$t$ is an artificial parameter used to smoothly interpolate between the unitary $U = -e^{i\pi H_F}$ used in $I_{S-F}$ and the trivial unitary $U=1$ through the map $U_t = -e^{itH_F}$ so the value of $t$ is bounded between $0$ and $\pi$. This parameter $t$ is not needed to compute $I_{S-F}$ but only introduced as an intermediate step in the computations to transform $I_{S-F}$ into the shell invariant.

4) In 3.5 the mode-shell correspondence is applied to HOTIs. If I understand correctly, a spectral flow Hamiltonian is constructed from the HOTI Hamiltonian to construct the mode-shell invariant. It is not clear to me why both Hamiltonians must share the same topological classification. Why is this the case?

The usual topological classifications of topological insulators are based on the space dimension $D$ (e.g. Kitaev). The mode-shell correspondence provides an alternative classification based on the number of dimensions along which gapless modes disperse $D_M$ and are localized $D_\perp$. In this framework, $D_M$ alone determines the symmetry class (unitary or chiral) and provides a topological index in any dimension. The invariant can be shown to be trivial in the ''bad'' symmetry class. Once this topological index is given, one can then distinguish between different physical situations, namely strong TI, weak TI, HOTI and semi-metals by considering $D_\perp$. This is how the same value of $D_M$ can gather both say Chern insulators and certain HOTIs, and we distinguish then those two phases by the value of $D_\perp$. Conversely, (certain) HOTIs with the same $D_\perp$ are classified according to $D_M$. For instance, for $D_\perp=2$, $D_M=1$ classifies HOTI with hinges mode and no chiral symmetry while $D_M=0$ classifies HOTI with corner modes and a chiral symmetry as discussed in the part I.

Requested changes:

1) I would like to request the authors to provide the code for at least some examples. I believe a clear demonstration of how to apply the mode-shell correspondence would help the readability of the manuscript. This is also point 6 of Scipost's general acceptance criteria.

We have uploaded the code used to generate the figures 8 and 10 on github as requested. An example of code computing the mode and shell indices in higher-order insulators can also be found in the appendix G of Part 1.

2) I believe that $\theta_\Gamma$ is first introduced by referencing the work mode-shell correspondence I, and $\Gamma$ is not introduced. I suggest the authors to introduce these variables and to mention in text that semi-classical limit is for $\Gamma \rightarrow \infty$ right before Eq. 11. At the moment it is only included in figure 6.

The semi-classical limit obtained when $\Gamma \rightarrow \infty$, with $\Gamma$ the radius of the shell is mentionned at the very beginning of the manuscript (3rd sentence). Still, we agree with the referee that a recall must be done in the appropriate paragraph, namely in pages 8 and 11, where the relevant formulas are introduced.

3) "The main problem in condensed-matter applications is that defects in the lattice structure (vacancies, impurities, ...) induce perturbations which do not vary slowly compared to the inter-site distance." I would like the authors to make this statement more precise. Is this the main problem of mode shell correspondence or of condensed matter theory? Why?

We thank the referee for this comment. Indeed our formulation was misleading. What we simply mean is that short range disorder in real space implies long range coupling in reciprocal space. Thus, the topological robustness of gapless modes confined in reciprocal space (e.g. quantum channels, Dirac/Weyl nodes) can be compromised in that case, as different gapless modes can couple because of such perturbations. In contrast, smooth perturbations in real space limit the coupling between gapless modes in reciprocal space and ensure their topological protection. We have rephrased this paragraph and added the following reference https://doi.org/10.1103/PhysRevB.104.235431 to illustrate that point.

4) "Such an approach is particularly useful to tackle topological modes in disordered systems where the semi-classical analysis is not suited, and so are neither the semi-classical shell invariants." I would like the authors to explain in more detail how the approach applies to disordered systems. They could consider including an demonstration. In particular, how does the approach scale in systems with disorder? These tend to be big and HF would naively be expensive to compute.

In its more general (non-commutative) formulation, the mode index only relies on the existence of a gap and does not assume translation invariance. As such, it can be directly employed to compute gapless modes in any dimension (for class A and AIII), whether in the bulk or at the boundaries. Following the recommendation of the referee, we have added the concrete example of the disordered QWZ model in appendix C, where gapless modes correspond to chiral edge states, and provided the code that computes the mode index in that case.

On the numerical aspect, the standard way to compute $H_F$ is through the diagonalisation of the Hamiltonian which scales as $N^3$ and can therefore be relatively expensive for large systems. However, as the index converges exponentially quickly, even small size systems of $L=10\sim 20$ sites can obtain a very good approximation of the invariant and characterize the topology. For example, the topological invariant of our disordered 2D QWZ model can be computed at a high accuracy on a standard computer in a few seconds.

---

## Editorial Decision

resubmitted